# THE UTILITY AND COMPLEXITY OF IN- AND OUT-OF-DISTRIBUTION MACHINE UNLEARNING

**Youssef Allouah**[1][*]**, Joshua Kazdan**[2]**, Rachid Guerraoui**[1]**, Sanmi Koyejo**[2]
[1]EPFL, Switzerland  [2]Stanford University, USA
`youssef.allouah@epfl.ch, sanmi@cs.stanford.edu`

## ABSTRACT

Machine unlearning, the process of selectively removing data from trained models, is increasingly crucial for addressing privacy concerns and knowledge gaps post-deployment. Despite this importance, existing approaches are often heuristic and lack formal guarantees. In this paper, we analyze the fundamental utility, time, and space complexity trade-offs of approximate unlearning, providing rigorous certification analogous to differential privacy. For in-distribution forget data—data similar to the retain set—we show that a surprisingly simple and general procedure, empirical risk minimization with output perturbation, achieves tight unlearning-utility-complexity trade-offs, addressing a previous theoretical gap on the separation from unlearning "for free" via differential privacy, which inherently facilitates the removal of such data. However, such techniques fail with out-of-distribution forget data—data significantly different from the retain set—where unlearning time complexity can exceed that of retraining, even for a single sample. To address this, we propose a new robust and noisy gradient descent variant that provably amortizes unlearning time complexity without compromising utility.

## 1 INTRODUCTION

The ability to selectively remove or "forget" portions of the training data from a model is a crucial challenge in modern machine learning. As deep neural networks are widely deployed across diverse domains like computer vision, natural language processing, and healthcare, there is a growing need to provide individuals with granular control over their data. This need is amplified by stringent regulations like the European Union's General Data Protection Regulation (GDPR) (Voigt and Von dem Bussche, 2017), which enshrines the "right to be forgotten," mandating data erasure upon request. Machine unlearning is designed to address this regulatory necessity by systematically removing the influence of specific training examples without compromising model performance.

Recent years have seen growing interest in developing algorithms that can efficiently "unlearn" data (Bourtoule et al., 2021; Nguyen et al., 2022; Foster et al., 2024). The proposed approaches range from methods like fine-tuning on the retained data while increasing loss on the forget data (Graves et al., 2021; Kurmanji et al., 2024), to more sophisticated techniques that provoke catastrophic forgetting by fine-tuning specific model layers (Goel et al., 2022). Unfortunately, many of these methods lack formal guarantees, rendering it unclear when they comply with regulatory standards.

Unlearning guarantees fall into two main categories: exact and approximate. Exact unlearning guarantees that the unlearned model has never used the forget data, often relying on sharding-based strategies (Bourtoule et al., 2021), which can be space-inefficient and lack error bounds. In contrast, approximate unlearning ensures only that the unlearned model is statistically close to one retrained from scratch without the forget data, akin to the protections offered by differential privacy (Dwork et al., 2014). While approximate unlearning provides a more feasible path in terms of efficiency and practicality, it has yet to fully address certain key challenges.

Although several prior works have explored approximate unlearning (Ginart et al., 2019; Neel et al., 2021; Chourasia and Shah, 2023), the theoretical understanding of the utility-complexity trade-offs remains incomplete. This is particularly true for challenging scenarios, such as when the forget data is out-of-distribution or adversarial. Such cases are highly relevant to practice, given the heterogeneity of user data and the observation that deletion requests are often non-random (Marchant et al., 2022).

---

[*]Work done while at Stanford University.

Addressing these challenges is crucial for making machine unlearning a practical and reliable tool for real-world applications, where models must adapt to diverse data and privacy requirements.

**Contributions.** Our work draws a comprehensive landscape of the utility-complexity trade-offs in approximate unlearning. We tackle two complementary unlearning scenarios: the *in-distribution* case where the forget data is an arbitrary subset of samples from the test distribution, and we initiate the study of the *out-of-distribution* scenario where the forget data may arbitrarily deviate from the test distribution. In particular, our results precisely quantify the number of samples that can be deleted under fixed utility and computation budgets, assuming the empirical loss has a unique global minimum. We analyze the unlearning-training pair consisting of a generic optimization procedure and output perturbation, and show that it can unlearn a constant fraction of the dataset, *independently* of the model dimension, thereby settling a theoretical question by Sekhari et al. (2021) and highlighting a tight separation with differential privacy. In the case of out-of-distribution forget data however, we show that this approach can fail to unlearn a single sample in the worst case. We propose a new algorithm using a robust and efficient variant of gradient descent during training, which ensures a good initialization for unlearning independently of the forget data. We show that this algorithm can certifiably unlearn a constant fraction of the dataset, with near-linear time and space complexities.

## 1.1 RELATED WORK

The concept of machine unlearning, introduced by Cao and Yang (2015), has gained significant attention following the introduction of data-privacy laws, such as GDPR, which mandates that companies must delete user data upon request. Cao and Yang (2015) studied exact unlearning, where the unlearned model behaves as if it had never used the forget data, deterministically mimicking retraining from scratch. However, this strict approach is only feasible for highly structured problems. Bourtoule et al. (2021) tackled this challenge with a partitioning strategy, training ensemble models on different data shards. While this approach reduces the need for full retraining when deleting data, it incurs high space complexity and lacks utility guarantees.

To address the impracticalities of exact unlearning, researchers have shifted focus to approximate unlearning, a relaxation of exact unlearning. Ginart et al. (2019) pioneered this direction, proposing that an unlearned model should be statistically indistinguishable from one retrained without the forget data, similar to the guarantees offered by differential privacy (Dwork et al., 2014). This laid the foundation for a spectrum of algorithms that balance computational efficiency and approximate unlearning guarantees (Guo et al., 2020; Izzo et al., 2021; Golatkar et al., 2021; Gupta et al., 2021; Neel et al., 2021; Chourasia and Shah, 2023).

**Certified unlearning.** A growing body of work focuses on providing guarantees for approximate unlearning, particularly for convex learning problems (Guo et al., 2020; Neel et al., 2021; Sekhari et al., 2021). Methods such as gradient descent with output perturbation (Neel et al., 2021) have proven effective in the convex case, even with sequential deletion requests. However, little is known about the generalization guarantees of such approaches, even in the convex case. Notably, Sekhari et al. (2021) first proved generalization guarantees for unlearning with a Newton step and output perturbation (Guo et al., 2020), uncovering a separation with differential privacy. Specifically, differential privacy without an unlearning mechanism is inherently limited in the number of deletions it can handle, while maintaining fixed utility on the test loss, a result further tightened by Huang and Canonne (2023). Our work improves upon this by showing that gradient descent with output perturbation offers sharper guarantees. Moreover, we demonstrate that this approach cannot be further improved in general, based on lower bounds from robust mean estimation (Diakonikolas et al., 2019).

**Unlearning out-of-distribution data.** Despite advances in certified unlearning, most approaches assume that the forget data is drawn from the same distribution as the training data, leaving out-of-distribution (OOD) data and adversarially corrupted forget data underexplored. Recent works (Goel et al., 2024; Pawelczyk et al., 2024) highlight the complexities that arise when deletion requests target OOD or corrupted samples. These scenarios are especially relevant given that deletion requests are often non-random, originating from diverse and heterogeneous data sources. Moreover, studies like Marchant et al. (2022) have demonstrated that OOD data can be manipulated to slow down certified unlearning algorithms, triggering retraining and causing denial-of-service-like attacks. Our work addresses these issues by introducing a new unlearning algorithm that is robust to OOD and corrupted data. The algorithm achieves near-linear time complexity for unlearning, independently of the nature of the forget data, and thus is provably robust against attacks like Marchant et al. (2022).

## 2 PROBLEM STATEMENT

Consider a training set $\mathcal{S}$ made of $n$ examples independently drawn from distribution $\mathcal{D}$ over data space $\mathcal{Z}$, and a loss function $\ell \colon \mathbb{R}^d \times \mathcal{Z} \to \mathbb{R}$. Our goal is to minimize the population risk:

$$\min_{\boldsymbol{\theta} \in \mathbb{R}^d} \quad \mathcal{L}(\boldsymbol{\theta}) := \mathbb{E}_{\mathbf{z} \sim \mathcal{D}} [\ell(\boldsymbol{\theta}; \mathbf{z})]. \tag{1}$$

During the *training* phase, we aim at solving the empirical risk minimization problem:

$$\min_{\boldsymbol{\theta} \in \mathbb{R}^d} \quad \mathcal{L}(\boldsymbol{\theta}; \mathcal{S}) := \frac{1}{|\mathcal{S}|} \sum_{\mathbf{z} \in \mathcal{S}} \ell(\boldsymbol{\theta}; \mathbf{z}). \tag{2}$$

Let $\mathcal{A}$ denote the training procedure aimed at solving (2) on the full dataset $\mathcal{S}$, producing a model $\mathcal{A}(\mathcal{S}) \in \mathbb{R}^d$. Next, consider a scenario where a subset $\mathcal{S}_f \subset \mathcal{S}$ of size $f := |\mathcal{S}_f|$, referred to as the *forget set*, needs to be removed. The goal is then to update the model based on the *retain set* $\mathcal{S} \setminus \mathcal{S}_f$. Ideally, one would retrain using only the retain set with $\mathcal{A}$. However, due to time and space constraints, an *approximate unlearning* procedure $\mathcal{U}$ is used, which modifies the original model $\mathcal{A}(\mathcal{S})$, knowing the forget data $\mathcal{S}_f$, to be as close as possible to $\mathcal{A}(\mathcal{S} \setminus \mathcal{S}_f)$.

Paraphrasing prior definitions (Ginart et al., 2019; Neel et al., 2021; Sekhari et al., 2021), we formalize approximate unlearning as statistical indistinguishability between the unlearned model and the model trained without the forget data. We denote by $\mathcal{Z}^* := \cup_{k \geq 1} \mathcal{Z}^k$ the space of datasets with elements in $\mathcal{Z}$, i.e., $\mathcal{S} \in \mathcal{Z}^*$ if and only if it is a tuple of data points from $\mathcal{Z}$.

**Definition 1** (($q, \varepsilon$)-approximate unlearning). *Let $q > 1, \varepsilon \geq 0$, and $\mathcal{S}_f \subset \mathcal{S} \in \mathcal{Z}^*$. Let $\mathcal{A} \colon \mathcal{Z}^* \to \mathbb{R}^d$ be a training procedure, and $\mathcal{U} \colon \mathcal{Z}^* \times \mathbb{R}^d \to \mathbb{R}^d$ be a randomized unlearning procedure. The pair $(\mathcal{U}, \mathcal{A})$ achieves $(q, \varepsilon)$-approximate unlearning, on training set $\mathcal{S}$ with forget set $\mathcal{S}_f$, if*

$$\mathrm{D}_q(\mathcal{U}(\mathcal{S}_f, \mathcal{A}(\mathcal{S})) \,\|\, \mathcal{U}(\varnothing, \mathcal{A}(\mathcal{S} \setminus \mathcal{S}_f))) \leq \varepsilon, \tag{3}$$

*where $\mathrm{D}_q(\cdot \,\|\, \cdot)$ is the Rényi divergence of order $q$ between the probability distributions of its arguments, defined for every $P_1, P_2$ as $\mathrm{D}_q(P_1 \,\|\, P_2) := \frac{1}{q-1} \log \mathbb{E}_{X \sim P_2} \left( \frac{P_1(X)}{P_2(X)} \right)^q$.*

Above, $\mathcal{U}(\mathcal{S}_f, \mathcal{A}(\mathcal{S}))$ is the unlearned model and $\mathcal{U}(\varnothing, \mathcal{A}(\mathcal{S} \setminus \mathcal{S}_f))$ is the model trained without the forget data and no unlearning request. The guarantee conveys similar semantics to differential privacy (Dwork et al., 2014); an adversary cannot confidently distinguish the unlearned model from the model trained without the forget data, with any auxiliary information.

In this work, we are interested in the trade-off of approximate unlearning with time and space complexity, for different types of data distributions, including when the forget data is out-of-distribution or corrupt. We define below the utility objectives of the in- and out-of-distribution unlearning scenarios. Throughout, we assume that the loss function $\ell$ is lower bounded, so that all minima are well-defined.

**Definition 2.** *Let $0 \leq f < n$. Consider a data distribution $\mathcal{D}$ over data space $\mathcal{Z}$, unlearning-training pair $(\mathcal{U}, \mathcal{A})$, and recall that $\mathcal{Z}^* := \cup_{k \geq 1} \mathcal{Z}^k$ is the set of training sets. Denote the population risk minimum by $\mathcal{L}_\star := \min_{\boldsymbol{\theta} \in \mathbb{R}^d} \mathcal{L}(\boldsymbol{\theta})$, and independent and identical sampling of $n$ and $n - f$ samples from $\mathcal{D}$ by $\mathcal{S} \sim \mathcal{D}^n$ and $\mathcal{S}_r \sim \mathcal{D}^{n-f}$, respectively.[1] We define the following utility objectives:*

1. *In-distribution:* $\quad \mathcal{L}_{\mathrm{ID}}(\mathcal{U}, \mathcal{A}) := \mathbb{E}_{\mathcal{S} \sim \mathcal{D}^n} \Big[ \max_{\substack{\mathcal{S}_f \subset \mathcal{S} \\ |\mathcal{S}_f| \leq f}} \mathcal{L}(\mathcal{U}(\mathcal{S}_f, \mathcal{A}(\mathcal{S}))) - \mathcal{L}_\star \Big],$

2. *Out-of-distribution:* $\quad \mathcal{L}_{\mathrm{OOD}}(\mathcal{U}, \mathcal{A}) := \mathbb{E}_{\mathcal{S}_r \sim \mathcal{D}^{n-f}} \Big[ \max_{\substack{\mathcal{S}_f \in \mathcal{Z}^* \\ |\mathcal{S}_f| \leq f}} \mathcal{L}(\mathcal{U}(\mathcal{S}_f, \mathcal{A}(\mathcal{S}_r \cup \mathcal{S}_f))) - \mathcal{L}_\star \Big].$

In other words, our in-distribution objective of unlearning quantifies the retained utility when, starting from a training set consisting of $n$ samples from the test distribution $\mathcal{D}$, an adversary can remove up to $f$ samples arbitrarily. This objective has been previously considered by Sekhari et al. (2021). Besides, our out-of-distribution objective considers the worst case where the training set is composed of up to $f$ samples that may not be from $\mathcal{D}$ and are to be removed, while the remainder $\mathcal{S}_r$ of the training set is sampled from $\mathcal{D}$. This objective has not been theoretically studied before in the context of unlearning, and covers practical cases where user data is very heterogeneous, or shifts over time, or is corrupt and needs to be "corrected" (Goel et al., 2024).

---

[1]In the out-of-distribution scenario, we denote the retain data by $\mathcal{S}_r$ for clarity, since it is sampled from the test distribution independently of the forget data.

# 3    DELETION CAPACITY & REDUCTION TO EMPIRICAL RISK MINIMIZATION

We begin by introducing the notion of *deletion capacity* to compare different unlearning-training pairs. Our approach extends the formalism from Sekhari et al. (2021), which defined deletion capacity in terms of utility. In this work, we introduce the concept of *computational deletion capacity*, which accounts for the time complexity incurred during unlearning.

**Definition 3.** *Let $n, \alpha, T > 0$ and consider a pair $(\mathcal{U}, \mathcal{A})$ satisfying approximate unlearning.*

1. *The* utility deletion capacity *is the maximum number $f(\alpha) < n$ of samples that can be removed while ensuring the error remains at most $\alpha$. We refer to it as* in-distribution *if the error is measured as $\mathcal{L}_{\mathrm{ID}}$, and* out-of-distribution *if it is measured as $\mathcal{L}_{\mathrm{OOD}}$.*

2. *The* computational deletion capacity *is the maximum number $f(T) < n$ of samples that can be removed within a given time complexity budget $T$.*

We observe that we aim for unlearning-training pairs which have the largest deletion capacities possible. Besides, introducing a computational aspect to deletion capacity is natural because, without computational constraints, the utility deletion capacity could be maximized by retraining from scratch. Similarly, the computational deletion capacity could be maximized by outputting a data-independent model. Consequently, we are interested in optimizing both capacities simultaneously—that is, maximizing the number of deletions while keeping both the error and time complexity low. An analogous space complexity analysis could also be considered, but for simplicity, we omit it here.

We now show that it is sufficient to analyze the deletion capacity using the empirical loss, via the following generic bounds on the population risk with worst-case deletion, under standard assumptions.

**Proposition 1.** *Assume that, for every $\mathbf{z} \in \mathcal{Z}$, the loss $\ell(\cdot\,; \mathbf{z})$ is $\mu$-strongly convex and $L$-smooth. Consider any unlearning-training pair $(\mathcal{U}, \mathcal{A})$, with output $\hat{\boldsymbol{\theta}} := \mathcal{U}(\mathcal{S}_f, \mathcal{A}(\mathcal{S}))$, and recall the notation of Definition 2. By denoting $\boldsymbol{\theta}^\star := \arg\min_{\boldsymbol{\theta} \in \mathbb{R}^d} \mathcal{L}(\boldsymbol{\theta})$ and $\sigma_\star^2 := \mathbb{E}_{\mathbf{z} \sim \mathcal{D}} \|\nabla \ell(\boldsymbol{\theta}^\star; \mathbf{z})\|^2$, we have*

$$\mathcal{L}_{\mathrm{OOD}}(\mathcal{U}, \mathcal{A}) \leq \frac{L}{\mu} \mathbb{E}_{\mathcal{S}_r \sim \mathcal{D}^{n-f}} \big[ \max_{\substack{\mathcal{S}_f \in \mathcal{Z}^\star \\ |\mathcal{S}_f| \leq f}} \mathcal{L}(\hat{\boldsymbol{\theta}}; \mathcal{S}_r) - \mathcal{L}_{\star, \mathcal{S}_r} \big] + \frac{L}{2\mu^2} \frac{\sigma_\star^2}{n-f}. \tag{4}$$

*Moreover, if $\nabla \ell(\boldsymbol{\theta}^\star; \mathbf{z}), \mathbf{z} \sim \mathcal{D}$, is sub-Gaussian with variance proxy $\sigma^2$, we have*

$$\mathcal{L}_{\mathrm{ID}}(\mathcal{U}, \mathcal{A}) \leq \frac{L}{\mu} \mathbb{E}_{\mathcal{S} \sim \mathcal{D}^n} \big[ \max_{\substack{\mathcal{S}_f \subset \mathcal{S} \\ |\mathcal{S}_f| \leq f}} \mathcal{L}(\mathcal{U}(\mathcal{S}_f, \mathcal{A}(\mathcal{S})); \mathcal{S} \setminus \mathcal{S}_f) - \mathcal{L}_{\star, \mathcal{S} \setminus \mathcal{S}_f} \big] + \frac{8L\sigma^2}{\mu^2} \frac{1 + f \ln(n)}{n - f}. \tag{5}$$

*Finally, assuming that for every $\mathbf{z} \in \mathcal{Z}$, the loss $\ell(\cdot\,; \mathbf{z})$ is $R$-Lipschitz, we have*

$$\mathcal{L}_{\mathrm{ID}}(\mathcal{U}, \mathcal{A}) \leq \frac{2L}{\mu} \mathbb{E}_{\mathcal{S} \sim \mathcal{D}^n} \big[ \max_{\substack{\mathcal{S}_f \subset \mathcal{S} \\ |\mathcal{S}_f| \leq f}} \mathcal{L}(\hat{\boldsymbol{\theta}}; \mathcal{S} \setminus \mathcal{S}_f) - \mathcal{L}_{\star, \mathcal{S} \setminus \mathcal{S}_f} \big] + \frac{4LR^2}{\mu^2} \left( \frac{1}{n} + \left( \frac{f}{n} \right)^2 \right). \tag{6}$$

This proposition provides a general strategy for analyzing the utility deletion capacity: if the worst-case empirical loss on the retain data is bounded by $\alpha_{\mathrm{emp}}$, this directly implies a bound on the number $f$ of deletions that can be made, with a guaranteed bound on the population risk. Importantly, we do not need to directly analyze the generalization error of the unlearned model.

For example, for a target population risk bound $\alpha \geq \alpha_{\mathrm{emp}}$ in the Lipschitz in-distribution case with $n = \Omega(\frac{1}{\alpha})$, we deduce that the utility deletion capacity is at least of the order $\Omega(n\sqrt{\alpha})$, i.e., a constant fraction of the full dataset when assuming a constant error. Another interesting example is 'lazy' differential privacy, that is ignoring removal requests after training with differential privacy, which satisfies approximate unlearning. Standard empirical risk minimization bounds of DP-SGD (Bassily et al., 2014), with the group differential privacy property adapted for Rényi differential privacy (Bun and Steinke, 2016), yield the empirical risk error $\widetilde{\mathcal{O}}(\frac{f^2 d}{n^2 \varepsilon})$. Plugging this bound in Proposition 1 guarantees a utility deletion capacity of $\Omega(n\sqrt{\frac{\alpha \varepsilon}{d}})$. This has recently been shown to be tight for 'lazy' differential privacy (Huang and Canonne, 2023), after converting from Rényi to approximate differential privacy (Mironov, 2017). We defer the full proofs related to this section to Appendix B.

---

**Algorithm 1** Unlearning via Noisy Minimizer Approximation

---

**Input:** Target empirical loss $\alpha_{\text{emp}}$, smoothness constant $L$, model dimension $d$, unlearning budget $\varepsilon$.

*Training:* get $\boldsymbol{\theta}_{\mathcal{S}}^{\mathcal{A}}$ by approximating the risk minimizer on $\mathcal{S}$ up to squared distance $\frac{\alpha_{\text{emp}}\varepsilon}{4Ld}$

*Unlearning:* get $\boldsymbol{\theta}^{\mathcal{U}}$ by approximating the risk minimizer on $\mathcal{S} \setminus \mathcal{S}_f$ up to squared distance $\frac{\alpha_{\text{emp}}\varepsilon}{4Ld}$

after initializing at $\boldsymbol{\theta}_{\mathcal{S}}^{\mathcal{A}}$

**return** $\boldsymbol{\theta}^{\mathcal{U}} + \mathcal{N}(0, \frac{\alpha_{\text{emp}}}{2Ld}\mathbf{I}_d)$

---

## 4 IN-DISTRIBUTION UNLEARNING VIA NOISY RISK MINIMIZATION

In this section, we tackle in-distribution unlearning. Specifically, we analyze Algorithm 1, a generic unlearning framework assuming access to an approximate empirical risk minimization oracle. We show that any instance of this framework can minimize the in-distribution empirical utility objective to an arbitrary precision, subject to standard assumptions.

**Notation.** For any dataset $\mathcal{S}$, we denote by $\boldsymbol{\theta}_{\mathcal{S}}^{\star}$ the global minimizer of the empirical loss $\mathcal{L}(\cdot\,; \mathcal{S})$, which we assume to be unique. In particular, for any forget set $\mathcal{S}_f \subset \mathcal{S}$, we denote by $\boldsymbol{\theta}_{\mathcal{S} \setminus \mathcal{S}_f}^{\star}$ the unique global minimizer of the empirical loss $\mathcal{L}(\cdot\,; \mathcal{S} \setminus \mathcal{S}_f)$. Consider an arbitrary optimization procedure $\mathcal{A}$ (for training or unlearning) which outputs $\boldsymbol{\theta}_{\mathcal{S}}^{\mathcal{A}} \in \mathbb{R}^d$ when given dataset $\mathcal{S}$ and initial model $\boldsymbol{\theta}_0 \in \mathbb{R}^d$. For every $\alpha_{\text{precision}}, \Delta_{\text{initial}} > 0$, we denote by $T_{\mathcal{A}}(\alpha_{\text{precision}}, \Delta_{\text{initial}})$ the computational complexity required by $\mathcal{A}$ to guarantee *on any dataset* $\mathcal{S}$ that, given the initialization error $\|\boldsymbol{\theta}_0 - \boldsymbol{\theta}_{\mathcal{S}}^{\star}\|^2 \leq \Delta_{\text{initial}}$, its output $\boldsymbol{\theta}_{\mathcal{S}}^{\mathcal{A}}$ satisfies $\|\boldsymbol{\theta}_{\mathcal{S}}^{\mathcal{A}} - \boldsymbol{\theta}_{\mathcal{S}}^{\star}\|^2 \leq \alpha_{\text{precision}}$.

**Theorem 1.** *Let $\varepsilon, \alpha_{\text{emp}}, \Delta > 0, 0 \leq f < n$, and $q > 1$. Assume that, for every $\mathbf{z} \in \mathcal{S}$, the loss $\ell(\cdot\,; \mathbf{z})$ is $L$-smooth, and $\varepsilon \leq d$. Assume that for every $\mathcal{S}_f \subset \mathcal{S}, |\mathcal{S}_f| \leq f$, the empirical loss over $\mathcal{S} \setminus \mathcal{S}_f$ has a unique minimizer. Recall the notation above and consider the unlearning-training pair $(\mathcal{U}, \mathcal{A})$ in Algorithm 1, with the initialization error of $\mathcal{A}$ on set $\mathcal{S}$ being at most $\Delta$.*

*Then, $(\mathcal{U}, \mathcal{A})$ satisfies $(q, q\varepsilon)$-approximate unlearning with empirical loss, over worst-case $\mathcal{S} \setminus \mathcal{S}_f$, at most $\alpha_{\text{emp}}$ in expectation over the randomness of the algorithm, with time complexity:*

$$\text{Training: } T_{\mathcal{A}}\left(\frac{\alpha_{\text{emp}}\varepsilon}{4Ld}, \Delta\right), \qquad \text{Unlearning: } T_{\mathcal{U}}\left(\frac{\alpha_{\text{emp}}\varepsilon}{2Ld}, \frac{\alpha_{\text{emp}}\varepsilon}{4Ld} + 2 \max_{\substack{\mathcal{S}_f \subset \mathcal{S} \\ |\mathcal{S}_f| \leq f}} \left\|\boldsymbol{\theta}_{\mathcal{S}}^{\star} - \boldsymbol{\theta}_{\mathcal{S} \setminus \mathcal{S}_f}^{\star}\right\|^2\right).$$

Theorem 1 covers most optimization methods that have been studied in certified unlearning, such as gradient descent (Neel et al., 2021; Chourasia and Shah, 2023) and the Newton method variants (Guo et al., 2020; Sekhari et al., 2021), and also covers unexplored methods with known gradient complexity bounds, e.g., those using stochastic gradients, projection, or acceleration (Bubeck et al., 2015). The proof crucially leverages the existence of a unique minimizer as an anchor point to guarantee statistical indistinguishability. In fact, the optimization oracle needs to reach the aforementioned minimizer up to precision proportional to the unlearning budget $\varepsilon$, to compensate for the output perturbation in Algorithm 1. Our approach generalizes previous analyzes (Neel et al., 2021), especially since it does not require convexity; it applies to several non-convex problems with a unique global minimizer, such as principal component analysis and matrix completion (Zhu et al., 2018).

Thanks to Theorem 1 and Proposition 1, we get Corollary 2 using gradient descent as an approximate risk minimizer, which has worst-case time complexity $\mathcal{O}(nd \log(\frac{\Delta}{\alpha_{\text{emp}}}))$ for strongly convex problems, for precision $\alpha_{\text{emp}}$ and initialization error $\Delta$, with space complexity $\mathcal{O}(d)$ (Nesterov et al., 2018).

**Corollary 2.** *Let $\varepsilon, \alpha, \alpha_{\text{emp}} > 0$, and $q > 1$. Assume that, for every $\mathbf{z} \in \mathcal{Z}$, the loss $\ell(\cdot\,; \mathbf{z})$ is $\mu$-strongly convex and $L$-smooth, and that $\varepsilon \leq d$. Consider the unlearning-training pair $(\mathcal{U}, \mathcal{A})$ in Algorithm 1, where the approximate minimizers are obtained via* gradient descent[2], *and denote $\boldsymbol{\theta}_0 \in \mathbb{R}^d$ the initial model for training.*

---

[2]i.e., the sequence $\boldsymbol{\theta}_{t+1} = \boldsymbol{\theta}_t - \frac{2}{L+\mu}\nabla\mathcal{L}(\boldsymbol{\theta}_t; \mathcal{S}), t \geq 0$, and we replace $\mathcal{S}$ with $\mathcal{S} \setminus \mathcal{S}_f$ for unlearning. We explain how to compute the number of iterations in Remark 6.

| Algorithm | In-Distribution Deletion Capacity | |
| --- | --- | --- |
| | Utility | Computational |
| Algorithm 1 with Gradient Descent | $\Omega(n\sqrt{\alpha})$ | $\Omega\left(n\frac{\alpha\varepsilon\exp(T/2nd)}{Rd\|\boldsymbol{\theta}_0-\boldsymbol{\theta}_{\mathcal{S}}^{\star}\|}\right)$ |
| Newton step (Sekhari et al., 2021) | $\widetilde{\Omega}\left(n\min\left\{\alpha,\frac{\sqrt{\alpha}\varepsilon^{1/4}}{d^{1/4}}\right\}\right)$ | $(n-1)\mathbf{1}_{T=\Omega(nd^2+d^{2.38})}$ |
| Differential Privacy (Huang and Canonne, 2023) | $\widetilde{\Theta}\left(n\sqrt{\frac{\alpha\varepsilon}{d}}\right)$ | $(n-1)\mathbf{1}_{T=\Omega(n^2d)}$ |
| Lower bound (Lai et al., 2016) | $\mathcal{O}(n\sqrt{\alpha})$ | — |

Table 1: Summary of the in-distribution deletion capacities (the larger, the better), for error bound $\alpha > 0$ and computation budget $T > 0$, under approximate unlearning for strongly convex tasks, with smoothness and Lipschitz assumptions. We adapt the unlearning guarantees of prior works to align with our Rényi divergence-based definition. The last two reported computational capacities mean that no sample can be unlearned unless $T$ exceeds the proven time complexity of these algorithms.

*Then, $(\mathcal{U}, \mathcal{A})$ satisfies $(q, q\varepsilon)$-approximate unlearning with empirical loss, over the worst-case $\mathcal{S} \setminus \mathcal{S}_f$, at most $\alpha_{\mathrm{emp}}$ in expectation over the randomness of the algorithm with time complexity:*

$$\textit{Training: } \mathcal{O}\left(nd\log\left(\frac{d}{\alpha_{\mathrm{emp}}\varepsilon}\|\boldsymbol{\theta}_0-\boldsymbol{\theta}_{\mathcal{S}}^{\star}\|^2\right)\right), \textit{Unlearning: } \mathcal{O}\left(nd\log\left(1+\frac{d}{\alpha_{\mathrm{emp}}\varepsilon}\max_{\substack{\mathcal{S}_f\subset\mathcal{S}\\|\mathcal{S}_f|\leq f}}\left\|\boldsymbol{\theta}_{\mathcal{S}}^{\star}-\boldsymbol{\theta}_{\mathcal{S}\setminus\mathcal{S}_f}^{\star}\right\|^2\right)\right),$$

*ignoring dependencies on $L, \mu$. Also, the space complexity is $\mathcal{O}(d)$ during training and unlearning.*

*For $\alpha_{\mathrm{emp}} \leq \alpha$, assuming that for every $\mathbf{z} \in \mathcal{Z}$ the loss $\ell(\cdot\,;\mathbf{z})$ is $R$-Lipschitz, the in-distribution population risk $\mathcal{L}_{\mathrm{ID}}(\mathcal{U}, \mathcal{A})$ is at most $\alpha$, if $n = \Omega(\frac{1}{\alpha})$ and $f = \mathcal{O}(n\sqrt{\alpha})$, with time complexity:*

$$\textit{Training: } \mathcal{O}\left(nd\log\left(\frac{d}{\alpha\varepsilon}\mathbb{E}_{\mathcal{S}}\|\boldsymbol{\theta}_0-\boldsymbol{\theta}_{\mathcal{S}}^{\star}\|^2\right)\right), \textit{Unlearning: } \mathcal{O}\left(nd\log\left(1+\frac{d}{\alpha\varepsilon}\left(\frac{Rf}{n}\right)^2\right)\right).$$

From the result above, we deduce that Algorithm 1, when using gradient descent, achieves an in-distribution utility deletion capacity of at least $\Omega(n\sqrt{\alpha})$. This implies that a constant fraction of the dataset can be deleted while maintaining a fixed error $\alpha$. Our analysis establishes a *tight separation* from 'lazy' differential privacy methods, where deletion capacity degrades polynomially with the model dimension $d$ (Huang and Canonne, 2023). Furthermore, this result answers a previously open theoretical question by Sekhari et al. (2021), demonstrating that dimension-independent utility deletion capacity is indeed possible. Notably, the best previously known utility deletion capacity decayed with dimension as $\Omega(1/d^{1/4})$(Sekhari et al., 2021). We experimentally validate this separation in Figure 1a on a simple least-squares regression task. Finally, we observe that the deletion capacity of $\Omega(n\sqrt{\alpha})$, as established in Corollary 2, is *tight* in terms of its dependence on both $\alpha$ and $n$, following lower bounds for robust mean estimation (Lai et al., 2016; Diakonikolas et al., 2019).

Meanwhile, the time complexity bound from Corollary 2 increases logarithmically with the fraction of unlearned samples. In fact, for a time budget $T$, the computational deletion capacity given in Corollary 2 is $\Omega(n\frac{\alpha\varepsilon\exp(T/2nd)}{Rd\|\boldsymbol{\theta}_0-\boldsymbol{\theta}_{\mathcal{S}}^{\star}\|})$ in the Lipschitz case. This capacity scales linearly with $n$ and exponentially with the time budget $T$, effectively counterbalancing the linear dependence on the unlearning budget $\varepsilon$, the error $\alpha$, and the inverse of the dimension $d$. The exponential dependence on $T$ is highly favorable, though it stems from the logarithmic gradient complexity of gradient descent in strongly convex tasks which degrades to quadratic for non-strongly convex tasks (Nesterov et al., 2018). In contrast, the algorithm by Sekhari et al. (2021), which achieved the previously best known utility deletion capacity, has a time complexity of $\mathcal{O}(nd^2 + d^{2.38})$ and space complexity of $\mathcal{O}(d^2)$. Consequently, its computational deletion capacity is zero unless the time budget $T$ is at least $\Omega(nd^2 + d^{2.38})$. This comparison shows that gradient descent with output perturbation possesses the largest known in-distribution deletion capacities. A summary comparison of various unlearning-training approaches, in terms of in-distribution deletion capacity, is provided in Table 1. We defer the full proofs related to this section to Appendix C.

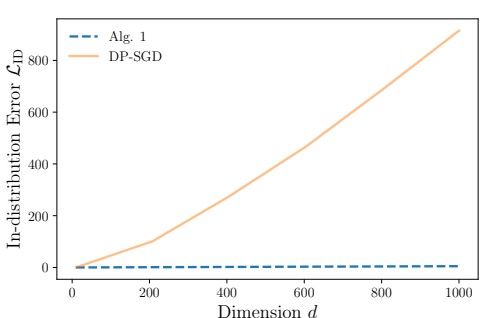
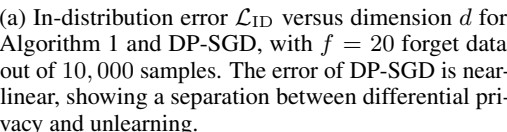
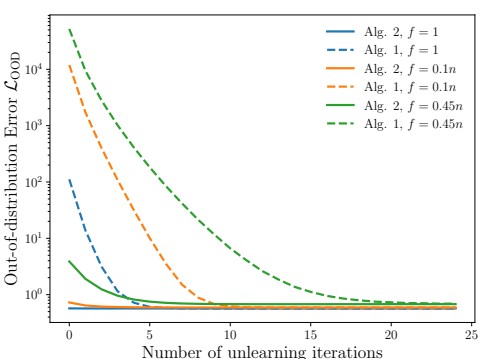

(a) In-distribution error $\mathcal{L}_{\text{ID}}$ versus dimension $d$ for Algorithm 1 and DP-SGD, with $f = 20$ forget data out of $10{,}000$ samples. The error of DP-SGD is near-linear, showing a separation between differential privacy and unlearning.

(b) Out-of-distribution error $\mathcal{L}_{\text{OOD}}$ versus number of *unlearning* iterations for Algorithms 1 and 2, using gradient descent, with $f \in \{1, 0.1n, 0.45n\}$ forget data out of $1{,}000$ samples. The per-iteration cost is the same for both algorithms. The unlearning time of Alg. 1 (non-robust) can be $10\times$ slower than Alg. 2.

Figure 1: Numerical validation on a linear regression task with synthetic data for the same unlearning budget, with in-distribution *(left)* and out-of-distribution *(right)* data. The in-distribution forget set is sampled at random, while the out-of-distribution data is obtained by shifting labels with a fixed offset. Additional details and results on real data can be found in Appendix F.

Thanks to our analysis, we also establish that the certified unlearning algorithms of Neel et al. (2021) and Chourasia and Shah (2023), whose generalization bounds were unknown prior to our work, also achieve a tight in-distribution utility deletion capacity, and a similar computational deletion capacity as Algorithm 1 with gradient descent. We recall that there are a few algorithmic differences with the latter, since Neel et al. (2021) additionally project models and Chourasia and Shah (2023) add noise at each iteration and assume a Gaussian model initialization.

## 5 OUT-OF-DISTRIBUTION UNLEARNING VIA ROBUST TRAINING

While Corollary 2 offers significant improvements over existing results, extending the same analysis to the out-of-distribution utility objective poses new challenges. In this case (Definition 2), the forget data can deviate arbitrarily from the test distribution. Unfortunately, the time complexity of the unlearning procedure in Algorithm 1 grows with the distance between the risk minimizers on the retain and full data, which becomes unbounded for the out-of-distribution objective, defeating the purpose of approximate unlearning in the worst case.

This is formalized in Proposition 2 below, where a *single* forget sample can make the initialization error of the unlearning phase of Algorithm 1 arbitrarily large. This naturally implies that the unlearning phase can be slower than retraining from an arbitrary initialization in the worst case, following standard gradient complexity lower bounds (Nesterov et al., 2018, Theorem 2.1.13).

**Proposition 2.** *Let $f = 1, n > 1$, and $\mathcal{Z} = \mathbb{R}^d$. There exists a 1-strongly convex and 1-smooth loss function, such that for any retain set $\mathcal{S}_r \in \mathcal{Z}^{n-1}$, any (unlearning-time) initialization error $\Delta > 0$, there exists a forget sample $\mathbf{z}_f \in \mathcal{Z}$ achieving $\left\| \boldsymbol{\theta}^\star_{\mathcal{S}_r \cup \{\mathbf{z}_f\}} - \boldsymbol{\theta}^\star_{\mathcal{S}_r} \right\|^2 = \Delta$, where $\boldsymbol{\theta}^\star_{\mathcal{S}_r}$ and $\boldsymbol{\theta}^\star_{\mathcal{S}_r \cup \{\mathbf{z}_f\}}$ denote the empirical minimizers on the retain and full data respectively.*

To address this, we introduce a new strategy where the goal is to train on the full dataset in a manner that minimizes sensitivity to the forget data. Since the forget data is unknown in advance and could potentially consist of outliers, we employ a robust variant of gradient descent. Specifically, we use the coordinate-wise trimmed mean of the gradient batch, as described in Algorithm 2. The trimmed mean, with trimming parameter $\tau$, is a classical robust statistics method that computes the average of all inputs along each coordinate, excluding the $\tau$ smallest and largest values (Lugosi and Mendelson,

---

**Algorithm 2** Unlearning via Robust Training and Noisy Minimizer Approximation

---

**Input:** Target empirical loss $\alpha_{\text{emp}}$, smoothness $L$ and strong convexity constant $\mu$, model dimension $d$, initial model $\boldsymbol{\theta}_0$, trimming parameter $f$, unlearning budget $\varepsilon$, initialization error $\Delta$.

*Training:* get $\boldsymbol{\theta}_{\mathcal{S}}^{\mathcal{A}_f}$ by robust training (shown below) for $K \geq \frac{2L}{\mu} \log(\frac{Ld\Delta}{\alpha_{\text{emp}}\varepsilon})$ iterations:

**for** $t = 0 \dots K-1$ **do**

    Compute the trimmed mean gradient: $\mathbf{r}_t = \text{TM}_f(\nabla\ell(\boldsymbol{\theta}_t; \mathbf{z}_1), \dots, \nabla\ell(\boldsymbol{\theta}_t; \mathbf{z}_n))$

    `/* average all but f largest and smallest inputs coordinate-wise */`

    Update the model: $\boldsymbol{\theta}_{t+1} = \boldsymbol{\theta}_t - \frac{1}{L}\mathbf{r}_t$

**end**

*Unlearning:* get $\boldsymbol{\theta}^{\mathcal{U}}$ by approximating the risk minimizer on $\mathcal{S} \setminus \mathcal{S}_f$ up to squared distance $\frac{\alpha_{\text{emp}}\varepsilon}{4Ld}$ by initializing at $\boldsymbol{\theta}_{\mathcal{S}}^{\mathcal{A}_f}$

**return** $\boldsymbol{\theta}^{\mathcal{U}} + \mathcal{N}(0, \frac{\alpha_{\text{emp}}}{2Ld}\mathbf{I}_d)$

---

2019). This approach allows mitigating the influence of outliers in the forget data, thereby enhancing the efficiency and robustness of the unlearning phase, especially in out-of-distribution settings.

In this section, we denote the retain set as $\mathcal{S}_r$ for clarity since, in the out-of-distribution scenario, it is sampled from the test distribution and independent of the forget data. Recall also that $\boldsymbol{\theta}_{\star,\mathcal{S}_r}$ is the minimizer of the empirical loss over the retain data. In order to analyze Algorithm 2, we introduce the *interpolation error* constant on the retain set $\mathcal{S}_r$, and its counterpart on the data distribution $\mathcal{D}$ given $k \geq 1$ samples, respectively:

$$\mathcal{E}(\mathcal{S}_r) := \frac{1}{|\mathcal{S}_r|} \sum_{\mathbf{z} \in \mathcal{S}_r} \left\| \nabla\ell(\boldsymbol{\theta}_{\mathcal{S}_r}^\star; \mathbf{z}) \right\|^2, \quad \mathcal{E}_k(\mathcal{D}) := \mathbb{E}_{\mathcal{S}_r \sim \mathcal{D}^k} \mathcal{E}(\mathcal{S}_r). \tag{7}$$

The smaller the interpolation error, the easier it is to fit the retain data, and the underlying data distribution, respectively. Theorem 3 below states the unlearning and utility guarantees of Algorithm 2.

**Theorem 3.** *Let* $\varepsilon, \alpha, \alpha_{\text{emp}}, \mu, L > 0, q > 1, \boldsymbol{\theta}_0 \in \mathbb{R}^d$, *and* $f \leq n \min\left\{\frac{1}{3}, \frac{12\mu}{5(L-\mu)}\right\}$. *Assume that, for every* $\mathbf{z} \in \mathcal{Z}$, *the loss* $\ell(\cdot\,; \mathbf{z})$ *is* $\mu$-*strongly convex and* $L$-*smooth. Consider the unlearning-training pair* $(\mathcal{U}, \mathcal{A})$ *in Algorithm 2 using gradient descent during unlearning.*

*Then,* $(\mathcal{U}, \mathcal{A})$ *satisfies* $(q, q\varepsilon)$-*approximate unlearning with empirical loss, over* $\mathcal{S}_r$ *with worst-case* $\mathcal{S}_f$, *at most* $\alpha_{\text{emp}}$ *in expectation over the randomness of the algorithm with time complexity:*

*Training:* $\mathcal{O}\left( nd \log\left( \frac{d}{\alpha_{\text{emp}}\varepsilon} \left\| \boldsymbol{\theta}_0 - \boldsymbol{\theta}_{\mathcal{S}_r}^\star \right\|^2 \right) \right)$, *Unlearning:* $\mathcal{O}\left( nd \log\left( 1 + \frac{d}{\alpha_{\text{emp}}\varepsilon} \frac{f}{n} \mathcal{E}(\mathcal{S}_r) \right) \right)$,

*ignoring dependencies on* $L, \mu$. *The space complexity is* $\mathcal{O}(d)$ *during training and unlearning. For* $\alpha_{\text{emp}} \leq \alpha$, *the out-of-distribution risk* $\mathcal{L}_{\text{OOD}}(\mathcal{U}, \mathcal{A})$ *is at most* $\alpha$, *if* $n - f = \Omega(\frac{1}{\alpha})$, *with time complexity:*

*Training:* $\mathcal{O}\left( nd \log\left( \frac{d}{\alpha\varepsilon} \mathbb{E}_{\mathcal{S}_r} \left\| \boldsymbol{\theta}_0 - \boldsymbol{\theta}_{\mathcal{S}_r}^\star \right\|^2 \right) \right)$, *Unlearning:* $\mathcal{O}\left( nd \log\left( 1 + \frac{d}{\alpha\varepsilon} \frac{f}{n} \mathcal{E}_{n-f}(\mathcal{D}) \right) \right)$.

Theorem 3 addresses the primary limitation of the analysis in Corollary 2: the unlearning time complexity is now independent of the out-of-distribution forget data. Instead, the time complexity is driven by the interpolation error on the retain set, rather than the difference between the empirical risk minimizers of the retain and full datasets. The interpolation error is often much smaller for well-behaved data or sufficiently large models, making this bound much tighter. In contrast, Corollary 2 could only achieve such a strong bound under the restrictive assumption that the loss is Lipschitz, which either results in an excessively large Lipschitz constant, e.g., scaling with model dimension for bounded domains, or excludes fundamental tasks, such as unconstrained least-squares regression. We numerically validate the robustness of the unlearning time complexity of Algorithm 2, compared to Algorithm 1, in Figure 1b on a least-squares regression task, and defer additional validation on real data to Appendix F. Finally, this result is the first theoretical guarantee against the so-called slow-down attacks in machine unlearning (Marchant et al., 2022), which not only seek to undermine utility but also unlearning efficiency, denial-of-service attacks.

| Algorithm | Out-of-Distribution Deletion Capacity | | |
| --- | --- | --- | --- |
| | Utility | Computational | |
| | | Lipschitz | Non-Lipschitz |
| Algorithm 1 with Gradient Descent | $n-1$ | $\Omega\left(n\frac{\alpha\varepsilon\exp(T/2nd)}{Rd\|\boldsymbol{\theta}_0-\boldsymbol{\theta}_{\mathcal{S}}^\star\|}\right)$ | $0$ |
| Algorithm 2 with Gradient Descent | $\Omega(n)$ | $\Omega\left(n\frac{\alpha^2\varepsilon^2\exp(T/nd)}{\mathcal{E}(\mathcal{S}_r)d^2\|\boldsymbol{\theta}_0-\boldsymbol{\theta}_{\mathcal{S}_r}^\star\|^2}\right)$ | $\Omega\left(n\frac{\alpha^2\varepsilon^2\exp(T/nd)}{\mathcal{E}(\mathcal{S}_r)d^2\|\boldsymbol{\theta}_0-\boldsymbol{\theta}_{\mathcal{S}_r}^\star\|^2}\right)$ |

Table 2: Summary of the out-of-distribution utility and computational deletion capacities (the larger, the better) due to Theorem 3, for error bound $\alpha > 0$ and computation budget $T > 0$, under approximate unlearning for strongly convex tasks, with smoothness and Lipschitz assumptions. The out-of-distribution deletion capacities of previous certified unlearning methods are not known.

The proof of Theorem 3 demonstrates that the robust training procedure converges to the empirical risk minimizer on the retain data, up to a small error proportional to the interpolation error and the fraction of forget data. This provides a solid initialization for the unlearning process, with limited sensitivity to the forget data. Additionally, although Algorithm 2 sets the trimming parameter $\tau$ equal to the size $f$ of the forget set for simplicity, the result of Theorem 3 only requires $\tau = \mathcal{O}(f)$, and can be straightforwardly extended to any trimming parameter by replacing $f$ by $\tau$ in the theorem. We defer the full proofs related to this section to Appendix E.

The most significant aspect of deletion capacity here is computational. Since we can achieve arbitrarily small empirical risk, Proposition 1 implies that the out-of-distribution utility deletion capacity of Algorithm 2 is only constrained by the fact that the trimming parameter can be at most half of the full data size, and is thus a constant fraction $\Omega(n)$ of the dataset. On the other hand, manipulating the time complexity bound from Theorem 3 gives a computational deletion capacity of $\Omega\left(n\frac{\alpha^2\varepsilon^2\exp(T/nd)}{\mathcal{E}(\mathcal{S}_r)d^2\|\boldsymbol{\theta}_0-\boldsymbol{\theta}_{\mathcal{S}_r}^\star\|^2}\right)$. This bound is favorable due to its exponential dependence on the time budget $T$, linear dependence on $n$, and the typically small interpolation error, which effectively mitigates the quadratic dependence on other parameters. In contrast, the unlearning time complexity of any unlearning-training pair covered by Theorem 1 maybe unbounded, as explained earlier in the section, and thus the corresponding deletion capacity is zero. Still, with the restrictive assumption that the initialization error is bounded on the full dataset and that the loss is $R$-Lipschitz everywhere, the computational deletion capacity from Corollary 2 is $\Omega\left(n\frac{\alpha\varepsilon\exp(T/2nd)}{Rd\|\boldsymbol{\theta}_0-\boldsymbol{\theta}_{\mathcal{S}}^\star\|}\right)$, which may be hindered by a large Lipschitz constant $R$ and does not benefit from a small interpolation error. A summary comparison of our unlearning-training approaches, in terms of out-of-distribution deletion capacity, is provided in Table 2.

## 6 CONCLUSION

This paper presents a theoretical analysis of the utility and complexity trade-offs in approximate machine unlearning. By focusing on both in-distribution and out-of-distribution unlearning scenarios, we offer new insights into how much data can be unlearnt under fixed computational budgets while maintaining utility. For the in-distribution case, we showed that a simple optimization procedure with output perturbation can unlearn a constant fraction of the dataset, independent of model dimension, thereby resolving a key theoretical question and highlighting the clear distinction from differential privacy-based unlearning approaches. For the more challenging out-of-distribution case, we introduced a robust gradient descent variant, ensuring a good initialization for unlearning and certifiably unlearning a constant fraction of the data with near-linear time and space complexity.

An intriguing open research direction is the analysis of unified upper bounds on the deletion capacities, specifically what is the maximum number of samples that can be deleted for a fixed computation, utility, and unlearning budgets? So far, only an upper bound on the utility deletion capacity is known. Other open research directions include extending our results to more complex models, and improving utility and time complexity guarantees in real-world applications.

ACKNOWLEDGMENTS

YA acknowledges support by SNSF doctoral mobility and 200021_200477 grants. SK acknowledges support by NSF 2046795 and 2205329, IES R305C240046, the MacArthur Foundation, Stanford HAI, OpenAI, and Google. YA thanks Anastasia Koloskova for feedback on the manuscript, and Berivan Isik, Ken Liu, Mehryar Mohri, and members of the Stanford STAIR lab for earlier discussions. The authors are thankful to the anonymous reviewers for their constructive comments.

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

## APPENDIX ORGANIZATION

The appendix is organized as follows. Appendix A recalls standard definitions used in the main paper. Appendix B contains the proof of Proposition 1. Appendix C contains the proofs of Theorem 1 and Corollary 2. Appendix D contains the proof of Proposition 2. Appendix E contains the proof of Theorem 3. Finally, Appendix F contains additional details on Tables 1 and 2 and Figure 1.

## A    STANDARD DEFINITIONS

We recall that we assume the loss function to be differentiable everywhere, throughout the paper.

**Definition 4** (L-smoothness). *A function $\mathcal{L}\colon \mathbb{R}^d \to \mathbb{R}$ is L-smooth if, for all $\boldsymbol{\theta}, \boldsymbol{\theta}' \in \mathbb{R}^d$, we have*

$$\mathcal{L}(\boldsymbol{\theta}') - \mathcal{L}(\boldsymbol{\theta}) - \langle \nabla\mathcal{L}(\boldsymbol{\theta}), \boldsymbol{\theta}' - \boldsymbol{\theta} \rangle \leq \frac{L}{2} \left\| \boldsymbol{\theta}' - \boldsymbol{\theta} \right\|^2 .$$

The above is equivalent to, for all $\boldsymbol{\theta}, \boldsymbol{\theta}' \in \mathbb{R}^d$, having $\left\| \nabla\mathcal{L}(\boldsymbol{\theta}') - \nabla\mathcal{L}(\boldsymbol{\theta}) \right\| \leq L \left\| \boldsymbol{\theta}' - \boldsymbol{\theta} \right\|$ (see, e.g., (Nesterov et al., 2018)).

**Definition 5** ($\mu$-strong convexity). *A function $\mathcal{L}\colon \mathbb{R}^d \to \mathbb{R}$ is $\mu$-stongly convex if, for all $\boldsymbol{\theta}, \boldsymbol{\theta}' \in \mathbb{R}^d$, we have*

$$\mathcal{L}(\boldsymbol{\theta}') - \mathcal{L}(\boldsymbol{\theta}) - \langle \nabla\mathcal{L}(\boldsymbol{\theta}), \boldsymbol{\theta}' - \boldsymbol{\theta} \rangle \geq \frac{\mu}{2} \left\| \boldsymbol{\theta}' - \boldsymbol{\theta} \right\|^2 .$$

Moreover, we recall that strong convexity implies the Polyak-Łojasiewicz (PL) inequality (Karimi et al., 2016) $2\mu \left( \mathcal{L}(\boldsymbol{\theta}) - \mathcal{L}_\star \right) \leq \left\| \nabla\mathcal{L}(\boldsymbol{\theta}) \right\|^2$. Note that a function satisfies $L$-smoothness and $\mu$-strong convexity inequality simultaneously only if $\mu \leq L$.

**Definition 6** (R-Lipschitz). *A function $\mathcal{L}\colon \mathbb{R}^d \to \mathbb{R}$ is R-Lipschitz if, for all $\boldsymbol{\theta}, \boldsymbol{\theta}' \in \mathbb{R}^d$, we have*

$$\left| \mathcal{L}(\boldsymbol{\theta}') - \mathcal{L}(\boldsymbol{\theta}) \right| \leq R \left\| \boldsymbol{\theta}' - \boldsymbol{\theta} \right\| .$$

The above is also equivalent to all the gradients being bounded by $R$ in norm.

## B    PROOF OF PROPOSITION 1

**Lemma 4.** *Let $0 \leq f < n$. Assume that, for every $\mathbf{z} \in \mathcal{Z}$, the loss $\ell(\,\cdot\,; \mathbf{z})$ is $\mu$-strongly convex and L-smooth. Then, we have*

$$\mathbb{E}_{\mathcal{S} \sim \mathcal{D}^n} \left[ \mathcal{L}(\boldsymbol{\theta}_{\mathcal{S}}^\star) - \mathcal{L}_\star \right] \leq \frac{L}{2\mu^2} \frac{\mathbb{E}_{\mathbf{z} \sim \mathcal{D}} \left\| \nabla\ell(\boldsymbol{\theta}^\star; \mathbf{z}) \right\|^2}{n}. \tag{8}$$

*Moreover, if $\nabla\ell(\boldsymbol{\theta}^\star; \mathbf{z}), \mathbf{z} \sim \mathcal{D}$, is sub-Gaussian with variance proxy $\sigma^2$, then*

$$\mathbb{E}_{\mathcal{S} \sim \mathcal{D}^n} \left[ \max_{\substack{\mathcal{S}_f \subset \mathcal{S} \\ |\mathcal{S}_f| \leq f}} \mathcal{L}(\boldsymbol{\theta}_{\mathcal{S} \setminus \mathcal{S}_f}^\star) - \mathcal{L}_\star \right] \leq \frac{8L\sigma^2}{\mu^2} \frac{f\ln(n) + 1}{n - f}. \tag{9}$$

*Proof.* Assume that, for every $\mathbf{z} \in \mathcal{Z}$, the loss $\ell(\,\cdot\,; \mathbf{z})$ is $\mu$-strongly convex and $L$-smooth. Then, successively smoothness and then strong convexity, we have

$$\mathbb{E}_{\mathcal{S} \sim \mathcal{D}^n} \left[ \mathcal{L}(\boldsymbol{\theta}_{\mathcal{S}}^\star) - \mathcal{L}_\star \right] \leq \frac{L}{2} \mathbb{E}_{\mathcal{S} \sim \mathcal{D}^n} \left\| \boldsymbol{\theta}_{\mathcal{S}}^\star - \boldsymbol{\theta}^\star \right\|^2 \leq \frac{L}{2\mu^2} \mathbb{E}_{\mathcal{S} \sim \mathcal{D}^n} \left\| \nabla\mathcal{L}(\boldsymbol{\theta}^\star; \mathcal{S}) \right\|^2$$

$$= \frac{L}{2\mu^2} \frac{\mathbb{E}_{\mathbf{z} \sim \mathcal{D}} \left\| \nabla\ell(\boldsymbol{\theta}^\star; \mathbf{z}) \right\|^2}{n}.$$

The last equality is simply due to $\mathcal{S}$ consisting of $n$ i.i.d. samples from $\mathcal{D}$. This proves the first statement.

Now, assume that $\nabla\ell(\boldsymbol{\theta}^\star; \mathbf{z}), \mathbf{z} \sim \mathcal{D}$, is sub-Gaussian with variance proxy $\sigma^2$. This implies that, for every $\mathcal{S}_f, |\mathcal{S}_f| \leq f$, $\nabla\mathcal{L}(\boldsymbol{\theta}^\star; \mathcal{S} \setminus \mathcal{S}_f)$ is sub-Gaussian with variance proxy $\frac{\sigma^2}{|\mathcal{S} \setminus \mathcal{S}_f|} \leq \frac{\sigma^2}{n-f}$, as the

average of sub-Gaussian of independent sub-Gaussian random variables (see (Rigollet and Hütter, 2015, Section 1.2)). As a standard property of sub-Gaussian variables (Pauwels, 2020, Theorem 2.1.1), we thus have

$$\mathbb{E}_{\mathcal{S}\sim\mathcal{D}^n} \exp\left(\frac{n-f}{8\sigma^2} \|\nabla\mathcal{L}(\boldsymbol{\theta}^\star; \mathcal{S}\setminus\mathcal{S}_f)\|^2\right) \leq 2.$$

Using the above, and Jensen's inequality, we have

$$\mathbb{E}_{\mathcal{S}\sim\mathcal{D}^n} \max_{\substack{\mathcal{S}_f\subset\mathcal{S} \\ |\mathcal{S}_f|\leq f}} \|\nabla\mathcal{L}(\boldsymbol{\theta}^\star; \mathcal{S}\setminus\mathcal{S}_f)\|^2 = \frac{8\sigma^2}{n-f} \mathbb{E}_{\mathcal{S}} \ln\left(\exp\left(\frac{n-f}{8\sigma^2} \max_{\substack{\mathcal{S}_f\subset\mathcal{S} \\ |\mathcal{S}_f|\leq f}} \|\nabla\mathcal{L}(\boldsymbol{\theta}^\star; \mathcal{S}\setminus\mathcal{S}_f)\|^2\right)\right)$$

$$\leq \frac{8\sigma^2}{n-f} \ln\left(\mathbb{E}_{\mathcal{S}} \exp\left(\frac{n-f}{8\sigma^2} \max_{\substack{\mathcal{S}_f\subset\mathcal{S} \\ |\mathcal{S}_f|\leq f}} \|\nabla\mathcal{L}(\boldsymbol{\theta}^\star; \mathcal{S}\setminus\mathcal{S}_f)\|^2\right)\right)$$

$$= \frac{8\sigma^2}{n-f} \ln\left(\mathbb{E}_{\mathcal{S}} \max_{\substack{\mathcal{S}_f\subset\mathcal{S} \\ |\mathcal{S}_f|\leq f}} \exp\left(\frac{n-f}{8\sigma^2} \|\nabla\mathcal{L}(\boldsymbol{\theta}^\star; \mathcal{S}\setminus\mathcal{S}_f)\|^2\right)\right)$$

$$\leq \frac{8\sigma^2}{n-f} \ln\left(\sum_{\substack{\mathcal{S}_f\subset\mathcal{S} \\ |\mathcal{S}_f|\leq f}} \mathbb{E}_{\mathcal{S}} \exp\left(\frac{n-f}{8\sigma^2} \|\nabla\mathcal{L}(\boldsymbol{\theta}^\star; \mathcal{S}\setminus\mathcal{S}_f)\|^2\right)\right) \leq \frac{8\sigma^2}{n-f} \ln\left(2\sum_{k=0}^{f}\binom{n}{k}\right).$$

We now use the following consequence of the binomial theorem: $\sum_{k=0}^{f}\binom{n}{k} \leq \sum_{k=0}^{f} n^k 1^{f-k} \leq (n+1)^f$. We obtain

$$\mathbb{E}_{\mathcal{S}\sim\mathcal{D}^n} \max_{\substack{\mathcal{S}_f\subset\mathcal{S} \\ |\mathcal{S}_f|\leq f}} \|\nabla\mathcal{L}(\boldsymbol{\theta}^\star; \mathcal{S}\setminus\mathcal{S}_f)\|^2 \leq \frac{8\sigma^2}{n-f} \ln\left(2\sum_{k=0}^{f}\binom{n}{k}\right) \leq \frac{8\sigma^2}{n-f} \ln\left(2(n+1)^f\right)$$

$$= \frac{8\sigma^2}{n-f}\left(\ln(2) + f\ln(n+1)\right).$$

Now, we use successively smoothness and strong convexity of the loss function, then the inequality above:

$$\mathbb{E}_{\mathcal{S}\sim\mathcal{D}^n}\left[\max_{\substack{\mathcal{S}_f\subset\mathcal{S} \\ |\mathcal{S}_f|\leq f}} \mathcal{L}(\boldsymbol{\theta}^\star_{\mathcal{S}\setminus\mathcal{S}_f}) - \mathcal{L}_\star\right] \leq \frac{L}{2} \mathbb{E}_{\mathcal{S}\sim\mathcal{D}^n} \max_{\substack{\mathcal{S}_f\subset\mathcal{S} \\ |\mathcal{S}_f|\leq f}} \left\|\boldsymbol{\theta}^\star_{\mathcal{S}\setminus\mathcal{S}_f} - \boldsymbol{\theta}^\star\right\|^2$$

$$\leq \frac{L}{2\mu^2} \mathbb{E}_{\mathcal{S}\sim\mathcal{D}^n} \max_{\substack{\mathcal{S}_f\subset\mathcal{S} \\ |\mathcal{S}_f|\leq f}} \|\nabla\mathcal{L}(\boldsymbol{\theta}^\star; \mathcal{S}\setminus\mathcal{S}_f)\|^2$$

$$= \frac{L}{2\mu^2}\frac{8\sigma^2}{n-f}\left(\ln(2) + f\ln(n+1)\right).$$

Simplifying the above upper bound concludes the proof. $\qquad\square$

**Lemma 5** ((Neel et al., 2021; Sekhari et al., 2021)). *Let $0 \leq f < n$ and $\mathcal{S} \in \mathcal{Z}^n$. Assume that, for every $\mathbf{z} \in \mathcal{Z}$, the loss $\ell(\cdot\,; \mathbf{z})$ is $\mu$-strongly convex and $R$-Lipschitz. We have*

$$\max_{\substack{\mathcal{S}_f\subset\mathcal{S} \\ |\mathcal{S}_f|\leq f}} \left\|\boldsymbol{\theta}^\star_{\mathcal{S}} - \boldsymbol{\theta}^\star_{\mathcal{S}\setminus\mathcal{S}_f}\right\| \leq \frac{2Rf}{\mu n}.$$

**Proposition 1.** *Assume that, for every $\mathbf{z} \in \mathcal{Z}$, the loss $\ell(\cdot\,; \mathbf{z})$ is $\mu$-strongly convex and $L$-smooth. Consider any unlearning-training pair $(\mathcal{U}, \mathcal{A})$, with output $\hat{\boldsymbol{\theta}} := \mathcal{U}(\mathcal{S}_f, \mathcal{A}(\mathcal{S}))$, and recall the notation of Definition 2. By denoting $\boldsymbol{\theta}^\star := \arg\min_{\boldsymbol{\theta}\in\mathbb{R}^d} \mathcal{L}(\boldsymbol{\theta})$ and $\sigma_\star^2 := \mathbb{E}_{\mathbf{z}\sim\mathcal{D}} \|\nabla\ell(\boldsymbol{\theta}^\star; \mathbf{z})\|^2$, we have*

$$\mathcal{L}_{\mathrm{OOD}}(\mathcal{U}, \mathcal{A}) \leq \frac{L}{\mu} \mathbb{E}_{\mathcal{S}_r\sim\mathcal{D}^{n-f}}\left[\max_{\substack{\mathcal{S}_f\in\mathcal{Z}^\star \\ |\mathcal{S}_f|\leq f}} \mathcal{L}(\hat{\boldsymbol{\theta}}; \mathcal{S}_r) - \mathcal{L}_{\star,\mathcal{S}_r}\right] + \frac{L}{2\mu^2}\frac{\sigma_\star^2}{n-f}. \qquad (4)$$

*Moreover, if $\nabla\ell(\boldsymbol{\theta}^\star;\mathbf{z}), \mathbf{z}\sim\mathcal{D}$, is sub-Gaussian with variance proxy $\sigma^2$, we have*

$$\mathcal{L}_{\mathrm{ID}}(\mathcal{U},\mathcal{A}) \leq \frac{L}{\mu}\,\mathbb{E}_{\mathcal{S}\sim\mathcal{D}^n}\big[\max_{\substack{\mathcal{S}_f\subset\mathcal{S}\\|\mathcal{S}_f|\leq f}}\mathcal{L}(\mathcal{U}(\mathcal{S}_f,\mathcal{A}(\mathcal{S}));\mathcal{S}\setminus\mathcal{S}_f) - \mathcal{L}_{\star,\mathcal{S}\setminus\mathcal{S}_f}\big] + \frac{8L\sigma^2}{\mu^2}\frac{1+f\ln(n)}{n-f}. \quad (5)$$

*Finally, assuming that for every $\mathbf{z}\in\mathcal{Z}$, the loss $\ell(\,\cdot\,;\mathbf{z})$ is R-Lipschitz, we have*

$$\mathcal{L}_{\mathrm{ID}}(\mathcal{U},\mathcal{A}) \leq \frac{2L}{\mu}\,\mathbb{E}_{\mathcal{S}\sim\mathcal{D}^n}\big[\max_{\substack{\mathcal{S}_f\subset\mathcal{S}\\|\mathcal{S}_f|\leq f}}\mathcal{L}(\hat{\boldsymbol{\theta}};\mathcal{S}\setminus\mathcal{S}_f) - \mathcal{L}_{\star,\mathcal{S}\setminus\mathcal{S}_f}\big] + \frac{4LR^2}{\mu^2}\left(\frac{1}{n}+\left(\frac{f}{n}\right)^2\right). \quad (6)$$

*Proof.* Assume that, for every $\mathbf{z}\in\mathcal{Z}$, the loss $\ell(\,\cdot\,;\mathbf{z})$ is $\mu$-strongly convex and $L$-smooth. Let $\varepsilon,\delta,\alpha > 0$. Moreover, denote by $\boldsymbol{\theta}^\star_{\mathcal{S}\setminus\mathcal{S}_f}$ and $\boldsymbol{\theta}^\star_{\mathcal{S}}$ the minimizers of the empirical loss functions $\mathcal{L}(\,\cdot\,;\mathcal{S}\setminus\mathcal{S}_f)$ and $\mathcal{L}(\,\cdot\,;\mathcal{S})$, respectively. For the out-of-distribution case, we denote by $\boldsymbol{\theta}^\star_{\mathcal{S}_r}$ the minimizer of the empirical loss function $\mathcal{L}(\,\cdot\,;\mathcal{S}_r)$. These exist and are well-defined by strong convexity of the loss function.

**Lipschitz in-distribution case.** Assume in addition that, for every $\mathbf{z}\in\mathcal{Z}$, the loss $\ell(\,\cdot\,;\mathbf{z})$ is $R$-Lipschitz. To analyze the population loss, we recall that the loss function is $R$-Lipschitz, $L$-smooth, and $\mu$-strongly convex by assumption, which allows using Lemma 4 as follows:

$$\mathbb{E}_{\mathcal{S}\sim\mathcal{D}^n}[\mathcal{L}(\boldsymbol{\theta}^\star_{\mathcal{S}}) - \mathcal{L}_\star] \leq \frac{L}{2\mu^2}\frac{\mathbb{E}_{\mathbf{z}\sim\mathcal{D}}\|\nabla\ell(\boldsymbol{\theta}^\star;\mathbf{z})\|^2}{n} \leq \frac{LR^2}{2\mu^2 n}. \quad (10)$$

Therefore, we have

$$\mathcal{L}_{\mathrm{ID}}(\mathcal{U},\mathcal{A}) := \mathbb{E}_{\mathcal{S}\sim\mathcal{D}^n}\big[\max_{\substack{\mathcal{S}_f\subset\mathcal{S}\\|\mathcal{S}_f|\leq f}}\mathcal{L}(\mathcal{U}(\mathcal{S}_f,\mathcal{A}(\mathcal{S}))) - \mathcal{L}_\star\big]$$

$$= \mathbb{E}_{\mathcal{S}\sim\mathcal{D}^n}\big[\max_{\substack{\mathcal{S}_f\subset\mathcal{S}\\|\mathcal{S}_f|\leq f}}\mathcal{L}(\mathcal{U}(\mathcal{S}_f,\mathcal{A}(\mathcal{S}))) - \mathcal{L}(\boldsymbol{\theta}^\star_{\mathcal{S}})\big] + \mathbb{E}_{\mathcal{S}\sim\mathcal{D}^n}[\mathcal{L}(\boldsymbol{\theta}^\star_{\mathcal{S}}) - \mathcal{L}_\star]$$

$$\leq \mathbb{E}_{\mathcal{S}\sim\mathcal{D}^n}\big[\max_{\substack{\mathcal{S}_f\subset\mathcal{S}\\|\mathcal{S}_f|\leq f}}\mathcal{L}(\mathcal{U}(\mathcal{S}_f,\mathcal{A}(\mathcal{S}))) - \mathcal{L}(\boldsymbol{\theta}^\star_{\mathcal{S}})\big] + \frac{4R^2}{\mu n}.$$

Now, we use that the loss function is $L$-smooth by assumption, followed by Jensen's inequality, which yields

$$\mathcal{L}_{\mathrm{ID}}(\mathcal{U},\mathcal{A}) \leq \frac{L}{2}\,\mathbb{E}_{\mathcal{S}\sim\mathcal{D}^n}\max_{\substack{\mathcal{S}_f\subset\mathcal{S}\\|\mathcal{S}_f|\leq f}}\|\mathcal{U}(\mathcal{S}_f,\mathcal{A}(\mathcal{S})) - \boldsymbol{\theta}^\star_{\mathcal{S}}\|^2 + \frac{LR^2}{2\mu^2 n}$$

$$\leq L\,\mathbb{E}_{\mathcal{S}\sim\mathcal{D}^n}\max_{\substack{\mathcal{S}_f\subset\mathcal{S}\\|\mathcal{S}_f|\leq f}}\left\|\mathcal{U}(\mathcal{S}_f,\mathcal{A}(\mathcal{S})) - \boldsymbol{\theta}^\star_{\mathcal{S}\setminus\mathcal{S}_f}\right\|^2 + L\,\mathbb{E}_{\mathcal{S}\sim\mathcal{D}^n}\max_{\substack{\mathcal{S}_f\subset\mathcal{S}\\|\mathcal{S}_f|\leq f}}\left\|\boldsymbol{\theta}^\star_{\mathcal{S}} - \boldsymbol{\theta}^\star_{\mathcal{S}\setminus\mathcal{S}_f}\right\|^2 + \frac{LR^2}{2\mu^2 n}.$$

We recall from (Sekhari et al., 2021, Lemma 6) that, by the Lipschitzness of the loss, we have

$$\max_{\substack{\mathcal{S}_f\subset\mathcal{S}\\|\mathcal{S}_f|\leq f}}\left\|\boldsymbol{\theta}^\star_{\mathcal{S}} - \boldsymbol{\theta}^\star_{\mathcal{S}\setminus\mathcal{S}_f}\right\| \leq \frac{2Rf}{\mu n}.$$

Taking squares and expectations and then plugging the above inequality in the previous one yields

$$\mathcal{L}_{\mathrm{ID}}(\mathcal{U},\mathcal{A}) \leq L\,\mathbb{E}_{\mathcal{S}\sim\mathcal{D}^n}\max_{\substack{\mathcal{S}_f\subset\mathcal{S}\\|\mathcal{S}_f|\leq f}}\left\|\mathcal{U}(\mathcal{S}_f,\mathcal{A}(\mathcal{S})) - \boldsymbol{\theta}^\star_{\mathcal{S}\setminus\mathcal{S}_f}\right\|^2 + 4L\left(\frac{Rf}{\mu n}\right)^2 + \frac{LR^2}{2\mu^2 n}.$$

Because the loss function is $\mu$-strongly convex by assumption, we have

$$\frac{\mu}{2}\max_{\substack{\mathcal{S}_f\subset\mathcal{S}\\|\mathcal{S}_f|\leq f}}\left\|\mathcal{U}(\mathcal{S}_f,\mathcal{A}(\mathcal{S})) - \boldsymbol{\theta}^\star_{\mathcal{S}\setminus\mathcal{S}_f}\right\|^2 \leq \max_{\substack{\mathcal{S}_f\subset\mathcal{S}\\|\mathcal{S}_f|\leq f}}\mathcal{L}(\mathcal{U}(\mathcal{S}_f,\mathcal{A}(\mathcal{S}));\mathcal{S}\setminus\mathcal{S}_f) - \mathcal{L}_{\star,\mathcal{S}\setminus\mathcal{S}_f}.$$

By taking expectations on the bound above and plugging it into the previous bound, we obtain

$$\mathcal{L}_{\mathrm{ID}}(\mathcal{U},\mathcal{A}) \leq \frac{2L}{\mu}\,\mathbb{E}_{\mathcal{S}\sim\mathcal{D}^n}\big[\max_{\substack{\mathcal{S}_f\subset\mathcal{S}\\|\mathcal{S}_f|\leq f}}\mathcal{L}(\mathcal{U}(\mathcal{S}_f,\mathcal{A}(\mathcal{S}));\mathcal{S}\setminus\mathcal{S}_f) - \mathcal{L}_{\star,\mathcal{S}\setminus\mathcal{S}_f}\big] + \frac{4LR^2}{\mu^2}\left(\frac{1}{n}+\left(\frac{f}{n}\right)^2\right).$$

Simplifying and rearranging terms concludes the proof of the Lipschitz in-distribution case.

**Sub-Gaussian in-distribution case.** Assume now that $\nabla\ell(\boldsymbol{\theta}^\star;\mathbf{z}), \mathbf{z}\sim\mathcal{D}$, is sub-Gaussian with variance proxy $\sigma^2$. Together with the strong convexity and smoothness assumptions, we can use the second statement of Lemma 4 as follows:

$$
\mathcal{L}_{\mathrm{ID}}(\mathcal{U},\mathcal{A}) \coloneqq \mathbb{E}_{\mathcal{S}\sim\mathcal{D}^n}\big[\max_{\substack{\mathcal{S}_f\subset\mathcal{S}\\|\mathcal{S}_f|\leq f}} \mathcal{L}(\mathcal{U}(\mathcal{S}_f,\mathcal{A}(\mathcal{S}))) - \mathcal{L}_\star\big]
$$

$$
= \mathbb{E}_{\mathcal{S}\sim\mathcal{D}^n}\big[\max_{\substack{\mathcal{S}_f\subset\mathcal{S}\\|\mathcal{S}_f|\leq f}} \mathcal{L}(\mathcal{U}(\mathcal{S}_f,\mathcal{A}(\mathcal{S}))) - \mathcal{L}(\boldsymbol{\theta}^\star_{\mathcal{S}\setminus\mathcal{S}_f})\big] + \mathbb{E}_{\mathcal{S}\sim\mathcal{D}^n}\big[\max_{\substack{\mathcal{S}_f\subset\mathcal{S}\\|\mathcal{S}_f|\leq f}} \mathcal{L}(\boldsymbol{\theta}^\star_{\mathcal{S}\setminus\mathcal{S}_f}) - \mathcal{L}_\star\big]
$$

$$
\leq \mathbb{E}_{\mathcal{S}\sim\mathcal{D}^n}\big[\max_{\substack{\mathcal{S}_f\subset\mathcal{S}\\|\mathcal{S}_f|\leq f}} \mathcal{L}(\mathcal{U}(\mathcal{S}_f,\mathcal{A}(\mathcal{S}))) - \mathcal{L}(\boldsymbol{\theta}^\star_{\mathcal{S}\setminus\mathcal{S}_f})\big] + \frac{8L\sigma^2}{\mu^2}\frac{f\ln(n)+1}{n-f}.
$$

Now, we use successively that loss is smooth then strongly convex, to show the following for any $\mathcal{S}$:

$$
\mathcal{L}(\mathcal{U}(\mathcal{S}_f,\mathcal{A}(\mathcal{S}))) - \mathcal{L}(\boldsymbol{\theta}^\star_{\mathcal{S}\setminus\mathcal{S}_f}) \leq \frac{L}{2}\left\|\mathcal{U}(\mathcal{S}_f,\mathcal{A}(\mathcal{S})) - \boldsymbol{\theta}^\star_{\mathcal{S}\setminus\mathcal{S}_f}\right\|^2
$$

$$
\leq \frac{L}{\mu}\left(\mathcal{L}(\mathcal{U}(\mathcal{S}_f,\mathcal{A}(\mathcal{S}));\mathcal{S}\setminus\mathcal{S}_f) - \mathcal{L}(\boldsymbol{\theta}^\star_{\mathcal{S}\setminus\mathcal{S}_f};\mathcal{S}\setminus\mathcal{S}_f)\right).
$$

Plugging the above back in the previous inequality, we obtain

$$
\mathcal{L}_{\mathrm{ID}}(\mathcal{U},\mathcal{A}) \coloneqq \mathbb{E}_{\mathcal{S}\sim\mathcal{D}^n}\big[\max_{\substack{\mathcal{S}_f\subset\mathcal{S}\\|\mathcal{S}_f|\leq f}} \mathcal{L}(\mathcal{U}(\mathcal{S}_f,\mathcal{A}(\mathcal{S}))) - \mathcal{L}_\star\big]
$$

$$
\leq \frac{L}{\mu}\mathbb{E}_{\mathcal{S}\sim\mathcal{D}^n}\big[\max_{\substack{\mathcal{S}_f\subset\mathcal{S}\\|\mathcal{S}_f|\leq f}} \mathcal{L}(\mathcal{U}(\mathcal{S}_f,\mathcal{A}(\mathcal{S}));\mathcal{S}\setminus\mathcal{S}_f) - \mathcal{L}_{\star,\mathcal{S}\setminus\mathcal{S}_f}\big] + \frac{8L\sigma^2}{\mu^2}\frac{f\ln(n)+1}{n-f}.
$$

This concludes the proof of the sub-Gaussian in-distribution case.

**Out-of-distribution case.** Using Lemma 4, the smoothness and strong convexity assumptions imply

$$
\mathbb{E}_{\mathcal{S}_r\sim\mathcal{D}^{n-f}}[\mathcal{L}(\boldsymbol{\theta}^\star_{\mathcal{S}_r}) - \mathcal{L}_\star] \leq \frac{L}{2\mu^2}\frac{\mathbb{E}_{\mathbf{z}\sim\mathcal{D}}\left\|\nabla\ell(\boldsymbol{\theta}^\star;\mathbf{z})\right\|^2}{n-f}. \tag{11}
$$

Therefore, we have

$$
\mathcal{L}_{\mathrm{OOD}}(\mathcal{U},\mathcal{A}) \coloneqq \mathbb{E}_{\mathcal{S}_r\sim\mathcal{D}^{n-f}}\big[\max_{\substack{\mathcal{S}_f\in\mathcal{Z}^*\\|\mathcal{S}_f|\leq f}} \mathcal{L}(\mathcal{U}(\mathcal{S}_f,\mathcal{A}(\mathcal{S}_r\cup\mathcal{S}_f))) - \mathcal{L}_\star\big]
$$

$$
= \mathbb{E}_{\mathcal{S}_r\sim\mathcal{D}^{n-f}}\big[\max_{\substack{\mathcal{S}_f\in\mathcal{Z}^*\\|\mathcal{S}_f|\leq f}} \mathcal{L}(\mathcal{U}(\mathcal{S}_f,\mathcal{A}(\mathcal{S}))) - \mathcal{L}(\boldsymbol{\theta}^\star_{\mathcal{S}_r})\big] + \mathbb{E}_{\mathcal{S}_r\sim\mathcal{D}^{n-f}}[\mathcal{L}(\boldsymbol{\theta}^\star_{\mathcal{S}_r}) - \mathcal{L}_\star]
$$

$$
\leq \mathbb{E}_{\mathcal{S}_r\sim\mathcal{D}^{n-f}}\big[\max_{\substack{\mathcal{S}_f\in\mathcal{Z}^*\\|\mathcal{S}_f|\leq f}} \mathcal{L}(\mathcal{U}(\mathcal{S}_f,\mathcal{A}(\mathcal{S}))) - \mathcal{L}(\boldsymbol{\theta}^\star_{\mathcal{S}_r})\big] + \frac{L}{2\mu^2}\frac{\mathbb{E}_{\mathbf{z}\sim\mathcal{D}}\left\|\nabla\ell(\boldsymbol{\theta}^\star;\mathbf{z})\right\|^2}{n-f}.
$$

Successively using smoothness and strong convexity, and recalling the notation $\mathcal{L}_{\star,\mathcal{S}_r} \coloneqq \mathcal{L}(\boldsymbol{\theta}^\star_{\mathcal{S}_r};\mathcal{S}_r) = \min_{\boldsymbol{\theta}\in\mathbb{R}^d}\mathcal{L}(\boldsymbol{\theta};\mathcal{S}_r)$, we have

$$
\mathcal{L}(\mathcal{U}(\mathcal{S}_f,\mathcal{A}(\mathcal{S}))) - \mathcal{L}(\boldsymbol{\theta}^\star_{\mathcal{S}_r}) \leq \frac{L}{2}\left\|\mathcal{U}(\mathcal{S}_f,\mathcal{A}(\mathcal{S})) - \boldsymbol{\theta}^\star_{\mathcal{S}_r}\right\|^2 \leq \frac{L}{\mu}\left(\mathcal{L}(\mathcal{U}(\mathcal{S}_f,\mathcal{A}(\mathcal{S}));\mathcal{S}_r) - \mathcal{L}_{\star,\mathcal{S}_r}\right).
$$

After taking a maximum over $\mathcal{S}_f$ and expectations and plugging this last bound in the one before, we get

$$
\mathcal{L}_{\mathrm{OOD}}(\mathcal{U},\mathcal{A}) \leq \frac{L}{\mu}\mathbb{E}_{\mathcal{S}_r\sim\mathcal{D}^{n-f}}\big[\max_{\substack{\mathcal{S}_f\in\mathcal{Z}^*\\|\mathcal{S}_f|\leq f}} \mathcal{L}(\mathcal{U}(\mathcal{S}_f,\mathcal{A}(\mathcal{S}));\mathcal{S}_r) - \mathcal{L}_{\star,\mathcal{S}_r}\big] + \frac{L}{2\mu^2}\frac{\mathbb{E}_{\mathbf{z}\sim\mathcal{D}}\left\|\nabla\ell(\boldsymbol{\theta}^\star;\mathbf{z})\right\|^2}{n-f}.
$$

This concludes the proof. $\qquad\square$

## C   PROOFS OF THEOREM 1 AND COROLLARY 2

**Theorem 1.** *Let $\varepsilon, \alpha_{\mathrm{emp}}, \Delta > 0, 0 \leq f < n$, and $q > 1$. Assume that, for every $\mathbf{z} \in \mathcal{S}$, the loss $\ell(\cdot\,; \mathbf{z})$ is $L$-smooth, and $\varepsilon \leq d$. Assume that for every $\mathcal{S}_f \subset \mathcal{S}, |\mathcal{S}_f| \leq f$, the empirical loss over $\mathcal{S} \setminus \mathcal{S}_f$ has a unique minimizer. Recall the notation above and consider the unlearning-training pair $(\mathcal{U}, \mathcal{A})$ in Algorithm 1, with the initialization error of $\mathcal{A}$ on set $\mathcal{S}$ being at most $\Delta$.*

*Then, $(\mathcal{U}, \mathcal{A})$ satisfies $(q, q\varepsilon)$-approximate unlearning with empirical loss, over worst-case $\mathcal{S} \setminus \mathcal{S}_f$, at most $\alpha_{\mathrm{emp}}$ in expectation over the randomness of the algorithm, with time complexity:*

$$\textit{Training: } T_{\mathcal{A}}\Big(\frac{\alpha_{\mathrm{emp}}\varepsilon}{4Ld}, \Delta\Big), \qquad \textit{Unlearning: } T_{\mathcal{U}}\Big(\frac{\alpha_{\mathrm{emp}}\varepsilon}{2Ld}, \frac{\alpha_{\mathrm{emp}}\varepsilon}{4Ld} + 2\max_{\substack{\mathcal{S}_f \subset \mathcal{S}\\|\mathcal{S}_f|\leq f}}\Big\|\boldsymbol{\theta}_{\mathcal{S}}^{\star} - \boldsymbol{\theta}_{\mathcal{S}\setminus\mathcal{S}_f}^{\star}\Big\|^2\Big).$$

*Proof.* Let $q > 1, \varepsilon, \alpha_{\mathrm{emp}} > 0, 0 \leq f < n$, and $\mathcal{S} \in \mathcal{Z}^n$ a given training set. Assume that the loss function is $L$-smooth at any data point, and that $\varepsilon \leq d$. Denote by $\boldsymbol{\theta}_{\mathcal{S}\setminus\mathcal{S}_f}^{\star}$ and $\boldsymbol{\theta}_{\mathcal{S}}^{\star}$ the minimizers of the empirical loss functions $\mathcal{L}(\cdot\,; \mathcal{S}\setminus\mathcal{S}_f)$ and $\mathcal{L}(\cdot\,; \mathcal{S})$, respectively. These exist and are well-defined by assumption. Also, following Algorithm 1, denote by $\boldsymbol{\theta}_{\mathcal{S}\setminus\mathcal{S}_f}^{\mathcal{A}}$ and $\boldsymbol{\theta}_{\mathcal{S}}^{\mathcal{A}}$ the model obtained using $\mathcal{A}$ over the training sets $\mathcal{S}\setminus\mathcal{S}_f$ and $\mathcal{S}$, respectively. Moreover, denote by $\boldsymbol{\theta}^{\mathcal{U}}$ the model obtained after using $\mathcal{U}$ over the training set $\mathcal{S}\setminus\mathcal{S}_f$ before Gaussian noise addition.

For any precision $\alpha_{\mathrm{emp}} > 0$, initialization error $\Delta > 0$, we denote the worst-case computational complexity of the training procedure to approximate the empirical risk minimizer up to squared distance $\alpha_{\mathrm{emp}}$ by $T_{\mathcal{A}}(\alpha_{\mathrm{emp}}, \Delta)$, and by $T_{\mathcal{U}}(\alpha_{\mathrm{emp}}, \Delta)$ during unlearning. Therefore, at the computational cost of $T_{\mathcal{A}}(\frac{\alpha_{\mathrm{emp}}\varepsilon}{4Ld}, \Delta)$ during training and $\max_{\substack{\mathcal{S}_f \subset \mathcal{S}\\|\mathcal{S}_f|\leq f}} T_{\mathcal{U}}(\frac{\alpha_{\mathrm{emp}}\varepsilon}{4Ld}, \|\boldsymbol{\theta}_{\mathcal{S}}^{\mathcal{A}} - \boldsymbol{\theta}_{\mathcal{S}\setminus\mathcal{S}_f}^{\star}\|^2)$ during unlearning since we initialize at $\boldsymbol{\theta}_{\mathcal{S}}^{\mathcal{A}}$, we have by definition

$$\left\|\boldsymbol{\theta}_{\mathcal{S}}^{\mathcal{A}} - \boldsymbol{\theta}_{\mathcal{S}}^{\star}\right\|^2 \leq \frac{\alpha_{\mathrm{emp}}\varepsilon}{4Ld}, \; \max_{\substack{\mathcal{S}_f \subset \mathcal{S}\\|\mathcal{S}_f|\leq f}}\left\|\boldsymbol{\theta}^{\mathcal{U}} - \boldsymbol{\theta}_{\mathcal{S}\setminus\mathcal{S}_f}^{\star}\right\|^2 \leq \frac{\alpha_{\mathrm{emp}}\varepsilon}{4Ld}, \; \left\|\boldsymbol{\theta}_{\mathcal{S}\setminus\mathcal{S}_f}^{\mathcal{A}} - \boldsymbol{\theta}_{\mathcal{S}\setminus\mathcal{S}_f}^{\star}\right\|^2 \leq \frac{\alpha_{\mathrm{emp}}\varepsilon}{4Ld}. \quad (12)$$

Moreover, using Jensen's inequality we have

$$\left\|\boldsymbol{\theta}_{\mathcal{S}}^{\mathcal{A}} - \boldsymbol{\theta}_{\mathcal{S}\setminus\mathcal{S}_f}^{\star}\right\|^2 \leq 2\left\|\boldsymbol{\theta}_{\mathcal{S}}^{\mathcal{A}} - \boldsymbol{\theta}_{\mathcal{S}}^{\star}\right\|^2 + 2\left\|\boldsymbol{\theta}_{\mathcal{S}}^{\star} - \boldsymbol{\theta}_{\mathcal{S}\setminus\mathcal{S}_f}^{\star}\right\|^2 \leq \frac{\alpha_{\mathrm{emp}}\varepsilon}{2Ld} + 2\left\|\boldsymbol{\theta}_{\mathcal{S}}^{\star} - \boldsymbol{\theta}_{\mathcal{S}\setminus\mathcal{S}_f}^{\star}\right\|^2. \quad (13)$$

Thus, the computational complexity of unlearning is upper bounded by $T_{\mathcal{U}}(\frac{\alpha_{\mathrm{emp}}\varepsilon}{4Ld}, \frac{\alpha_{\mathrm{emp}}\varepsilon}{2Ld} + 2\max_{\substack{\mathcal{S}_f \subset \mathcal{S}\\|\mathcal{S}_f|\leq f}}\left\|\boldsymbol{\theta}_{\mathcal{S}}^{\star} - \boldsymbol{\theta}_{\mathcal{S}\setminus\mathcal{S}_f}^{\star}\right\|^2)$.

**Unlearning analysis.**   Our goal here is to show that $\mathcal{U}(\mathcal{S}_f, \mathcal{A}(\mathcal{S}))$ and $\mathcal{U}(\varnothing, \mathcal{A}(\mathcal{S}\setminus\mathcal{S}_f))$ are near-indistinguishable in the sense of Definition 1. To do so, we bound the distance between $\boldsymbol{\theta}_{\mathcal{S}\setminus\mathcal{S}_f}^{\mathcal{A}}$ and $\boldsymbol{\theta}^{\mathcal{U}}$, and infer the unlearning guarantee via the Rényi divergence bound of the Gaussian mechanism.

Now, using inequalities (12) and Jensen's inequality, we obtain

$$\left\|\boldsymbol{\theta}^{\mathcal{U}} - \boldsymbol{\theta}_{\mathcal{S}\setminus\mathcal{S}_f}^{\mathcal{A}}\right\|^2 \leq 2\left\|\boldsymbol{\theta}^{\mathcal{U}} - \boldsymbol{\theta}_{\mathcal{S}\setminus\mathcal{S}_f}^{\star}\right\|^2 + 2\left\|\boldsymbol{\theta}_{\mathcal{S}\setminus\mathcal{S}_f}^{\mathcal{A}} - \boldsymbol{\theta}_{\mathcal{S}\setminus\mathcal{S}_f}^{\star}\right\|^2 \leq \frac{\alpha_{\mathrm{emp}}\varepsilon}{Ld}. \quad (14)$$

Recall that $\mathcal{U}(\mathcal{S}_f, \mathcal{A}(\mathcal{S})) := \boldsymbol{\theta}^{\mathcal{U}} + \mathcal{N}(0, \frac{\alpha_{\mathrm{emp}}}{2Ld}\mathbf{I}_d)$ and $\mathcal{U}(\varnothing, \mathcal{A}(\mathcal{S}\setminus\mathcal{S}_f)) := \boldsymbol{\theta}_{\mathcal{S}\setminus\mathcal{S}_f}^{\mathcal{A}} + \mathcal{N}(0, \frac{\alpha_{\mathrm{emp}}}{2Ld}\mathbf{I}_d)$. We recall that the formula of Rényi divergences (Gil et al., 2013) of order $q$ for Gaussians $\mathcal{N}(\mu, \boldsymbol{\Sigma}), \mathcal{N}(\mu', \boldsymbol{\Sigma})$ for arbitrary vectors $\mu, \mu' \in \mathbb{R}^d$ and symmetric positive definite matrix $\boldsymbol{\Sigma} \in \mathbb{R}^{d\times d}$ is $\frac{q}{2}(\mu - \mu')^{\top}\boldsymbol{\Sigma}^{-1}(\mu - \mu')$. Therefore, we conclude that $(\mathcal{U}, \mathcal{A})$ satisfies $(q, q\varepsilon)$-approximate unlearning:

$$\mathrm{D}_q(\mathcal{U}(\mathcal{S}_f, \mathcal{A}(\mathcal{S})) \,\|\, \mathcal{U}(\varnothing, \mathcal{A}(\mathcal{S}\setminus\mathcal{S}_f))) = \frac{q}{2 \cdot \frac{\alpha_{\mathrm{emp}}}{2Ld}}\left\|\boldsymbol{\theta}^{\mathcal{U}} - \boldsymbol{\theta}_{\mathcal{S}\setminus\mathcal{S}_f}^{\mathcal{A}}\right\|^2 \leq q\varepsilon. \quad (15)$$

**Utility analysis.** We now analyze the empirical and population loss of the model $\mathcal{U}(\mathcal{S}_f, \mathcal{A}(\mathcal{S}))$.

Recall that the loss function is $L$-smooth. Therefore, using inequalities (12), Jensen's inequality and taking expectations over the randomness of the additive Gaussian noise $\mathcal{N}(0, \frac{\alpha_{\text{emp}}}{2Ld}\mathbf{I}_d)$, we can bound the empirical loss:

$$
\mathbb{E}[\max_{\substack{\mathcal{S}_f \subset \mathcal{S} \\ |\mathcal{S}_f| \leq f}} \mathcal{L}(\mathcal{U}(\mathcal{S}_f, \mathcal{A}(\mathcal{S})); \mathcal{S} \setminus \mathcal{S}_f)] - \mathcal{L}^\star_{\mathcal{S} \setminus \mathcal{S}_f} \leq \frac{L}{2} \mathbb{E}[\max_{\substack{\mathcal{S}_f \subset \mathcal{S} \\ |\mathcal{S}_f| \leq f}} \left\| \mathcal{U}(\mathcal{S}_f, \mathcal{A}(\mathcal{S})) - \boldsymbol{\theta}^\star_{\mathcal{S} \setminus \mathcal{S}_f} \right\|^2]
$$

$$
= \frac{L}{2} \mathbb{E}_{\mathbf{X} \sim \mathcal{N}(0, \frac{\alpha_{\text{emp}}}{2Ld}\mathbf{I}_d)}[\max_{\substack{\mathcal{S}_f \subset \mathcal{S} \\ |\mathcal{S}_f| \leq f}} \left\| \boldsymbol{\theta}_{\mathcal{U}} - \boldsymbol{\theta}^\star_{\mathcal{S} \setminus \mathcal{S}_f} + \mathbf{X} \right\|^2]
$$

$$
\leq L \max_{\substack{\mathcal{S}_f \subset \mathcal{S} \\ |\mathcal{S}_f| \leq f}} \left\| \boldsymbol{\theta}^{\mathcal{U}} - \boldsymbol{\theta}^\star_{\mathcal{S} \setminus \mathcal{S}_f} \right\|^2 + L \mathbb{E}_{\mathbf{X} \sim \mathcal{N}(0, \frac{\alpha_{\text{emp}}}{2Ld}\mathbf{I}_d)} \left\| \mathbf{X} \right\|^2
$$

$$
= L \max_{\substack{\mathcal{S}_f \subset \mathcal{S} \\ |\mathcal{S}_f| \leq f}} \left\| \boldsymbol{\theta}^{\mathcal{U}} - \boldsymbol{\theta}^\star_{\mathcal{S} \setminus \mathcal{S}_f} \right\|^2 + Ld\frac{\alpha_{\text{emp}}}{2Ld} \leq L\frac{\alpha_{\text{emp}}\varepsilon}{4Ld} + \frac{\alpha_{\text{emp}}}{2} \leq \alpha_{\text{emp}}, \tag{16}
$$

after using the assumption that $\varepsilon \leq d$ for the last inequality. Therefore, the expected empirical risk error is at most $\alpha_{\text{emp}}$ with the following computational complexities before and during unlearning respectively:

$$
T_{\mathcal{A}}(\frac{\alpha_{\text{emp}}\varepsilon}{4Ld}, \Delta), \quad T_{\mathcal{U}}(\frac{\alpha_{\text{emp}}\varepsilon}{2Ld}, \frac{\alpha_{\text{emp}}\varepsilon}{2Ld} + 2 \max_{\substack{\mathcal{S}_f \subset \mathcal{S} \\ |\mathcal{S}_f| \leq f}} \left\| \boldsymbol{\theta}^\star_{\mathcal{S}} - \boldsymbol{\theta}^\star_{\mathcal{S} \setminus \mathcal{S}_f} \right\|^2). \tag{17}
$$

This concludes the proof. $\qquad\square$

**Corollary 2.** *Let $\varepsilon, \alpha, \alpha_{\text{emp}} > 0$, and $q > 1$. Assume that, for every $\mathbf{z} \in \mathcal{Z}$, the loss $\ell(\cdot\,;\mathbf{z})$ is $\mu$-strongly convex and $L$-smooth, and that $\varepsilon \leq d$. Consider the unlearning-training pair $(\mathcal{U}, \mathcal{A})$ in Algorithm 1, where the approximate minimizers are obtained via* gradient descent[3], *and denote $\boldsymbol{\theta}_0 \in \mathbb{R}^d$ the initial model for training.*

*Then, $(\mathcal{U}, \mathcal{A})$ satisfies $(q, q\varepsilon)$-approximate unlearning with empirical loss, over the worst-case $\mathcal{S} \setminus \mathcal{S}_f$, at most $\alpha_{\text{emp}}$ in expectation over the randomness of the algorithm with time complexity:*

*Training:* $\mathcal{O}\left(nd \log \left( \frac{d}{\alpha_{\text{emp}}\varepsilon} \left\| \boldsymbol{\theta}_0 - \boldsymbol{\theta}^\star_{\mathcal{S}} \right\|^2 \right)\right)$, *Unlearning:* $\mathcal{O}\left(nd \log \left( 1 + \frac{d}{\alpha_{\text{emp}}\varepsilon} \max_{\substack{\mathcal{S}_f \subset \mathcal{S} \\ |\mathcal{S}_f| \leq f}} \left\| \boldsymbol{\theta}^\star_{\mathcal{S}} - \boldsymbol{\theta}^\star_{\mathcal{S} \setminus \mathcal{S}_f} \right\|^2 \right)\right)$,

*ignoring dependencies on $L, \mu$. Also, the space complexity is $\mathcal{O}(d)$ during training and unlearning.*

*For $\alpha_{\text{emp}} \leq \alpha$, assuming that for every $\mathbf{z} \in \mathcal{Z}$ the loss $\ell(\cdot\,;\mathbf{z})$ is $R$-Lipschitz, the in-distribution population risk $\mathcal{L}_{\text{ID}}(\mathcal{U}, \mathcal{A})$ is at most $\alpha$, if $n = \Omega(\frac{1}{\alpha})$ and $f = \mathcal{O}(n\sqrt{\alpha})$, with time complexity:*

*Training:* $\mathcal{O}\left(nd \log \left( \frac{d}{\alpha\varepsilon} \mathbb{E}_{\mathcal{S}} \left\| \boldsymbol{\theta}_0 - \boldsymbol{\theta}^\star_{\mathcal{S}} \right\|^2 \right)\right)$, *Unlearning:* $\mathcal{O}\left(nd \log \left( 1 + \frac{d}{\alpha\varepsilon} \left( \frac{Rf}{n} \right)^2 \right)\right)$.

*Proof.* Let $\varepsilon, \alpha, \alpha_{\text{emp}} > 0$, and $q > 1$. Assume that, for every $\mathbf{z} \in \mathcal{Z}$, the loss $\ell(\cdot\,;\mathbf{z})$ is $\mu$-strongly convex and $L$-smooth, and that $\varepsilon \leq d$. Consider the training-unlearning pair $(\mathcal{U}, \mathcal{A})$ in Algorithm 1, where the approximate minimizer is obtained via gradient descent, with initialization $\boldsymbol{\theta}_0 \in \mathbb{R}^d$ during training.

For gradient descent, the worst-case computational complexity to reach precision (squared distance to empirical risk minimizer) $\alpha_{\text{emp}} > 0$ with initialization error (squared distance to empirical risk minimizer) $\Delta > 0$ is $\mathcal{O}(nd \log \frac{\Delta}{\alpha_{\text{emp}}})$ ignoring dependencies on $L, \mu$ (see, e.g., Nesterov et al. (2018)). Therefore, by applying Theorem 1, we have that $(\mathcal{U}, \mathcal{A})$ satisfies $(q, q\varepsilon)$-approximate unlearning. Moreover, in terms of utility, we have

$$
\mathbb{E}_{\mathcal{U}} \max_{\substack{\mathcal{S}_f \subset \mathcal{S} \\ |\mathcal{S}_f| \leq f}} \mathcal{L}(\mathcal{U}(\mathcal{S}_f, \mathcal{A}(\mathcal{S})); \mathcal{S} \setminus \mathcal{S}_f) - \mathcal{L}_{\star, \mathcal{S} \setminus \mathcal{S}_f} \leq \alpha_{\text{emp}}, \tag{18}
$$

---

[3]i.e., the sequence $\boldsymbol{\theta}_{t+1} = \boldsymbol{\theta}_t - \frac{2}{L+\mu} \nabla \mathcal{L}(\boldsymbol{\theta}_t; \mathcal{S})$, $t \geq 0$, and we replace $\mathcal{S}$ with $\mathcal{S} \setminus \mathcal{S}_f$ for unlearning. We explain how to compute the number of iterations in Remark 6.

with the following training and unlearning time respectively:

$$\mathcal{O}\left(nd\log\left(\frac{d}{\alpha_{\text{emp}}\varepsilon}\|\boldsymbol{\theta}_0 - \boldsymbol{\theta}_{\mathcal{S}}^\star\|^2\right)\right), \mathcal{O}\left(nd\log\left(1 + \frac{d}{\alpha_{\text{emp}}\varepsilon}\max_{\substack{\mathcal{S}_f \subset \mathcal{S} \\ |\mathcal{S}_f| \leq f}}\left\|\boldsymbol{\theta}_{\mathcal{S}}^\star - \boldsymbol{\theta}_{\mathcal{S}\setminus\mathcal{S}_f}^\star\right\|^2\right)\right).$$

In fact, we can obtain the guarantee (18) in expectation over $\mathcal{S} \sim \mathcal{D}^n$, at the cost of the expectation of the runtimes above. Indeed, taking expectations over the training set in the standard convergence guarantee of gradient descent for smooth strongly convex problems (e.g., (Nesterov et al., 2018, Theorem 2.1.15) implies that expected error $\alpha$ with expected initialization error $\mathbb{E}_{\mathcal{S}}[\Delta]$ can be achieved in $\mathcal{O}(nd\frac{L}{\mu}\log\frac{\mathbb{E}_{\mathcal{S}}[\Delta]}{\alpha})$ time. That is, we have

$$\mathbb{E}_{\mathcal{S}\sim\mathcal{D}^n}\left[\max_{\substack{\mathcal{S}_f \subset \mathcal{S} \\ |\mathcal{S}_f| \leq f}}\mathcal{L}(\mathcal{U}(\mathcal{S}_f, \mathcal{A}(\mathcal{S})); \mathcal{S} \setminus \mathcal{S}_f) - \mathcal{L}_{\star, \mathcal{S}\setminus\mathcal{S}_f}\right] \leq \alpha, \tag{19}$$

with the following training and unlearning time respectively:

$$\mathcal{O}\left(nd\log\left(\frac{d}{\alpha\varepsilon}\mathbb{E}_{\mathcal{S}\sim\mathcal{D}^n}\|\boldsymbol{\theta}_0 - \boldsymbol{\theta}_{\mathcal{S}}^\star\|^2\right)\right), \mathcal{O}\left(nd\log\left(1 + \frac{d}{\alpha\varepsilon}\mathbb{E}_{\mathcal{S}\sim\mathcal{D}^n}\max_{\substack{\mathcal{S}_f \subset \mathcal{S} \\ |\mathcal{S}_f| \leq f}}\left\|\boldsymbol{\theta}_{\mathcal{S}}^\star - \boldsymbol{\theta}_{\mathcal{S}\setminus\mathcal{S}_f}^\star\right\|^2\right)\right).$$

In turn, assuming that the loss is $R$-Lipschitz at any data point, we can plug the bound (38) in the generalization bound of Proposition 1. As a result, we have

$$\mathcal{L}_{\text{ID}}(\mathcal{U}, \mathcal{A}) \leq \frac{2L}{\mu}\mathbb{E}_{\mathcal{S}\sim\mathcal{D}^n}\left[\max_{\substack{\mathcal{S}_f \subset \mathcal{S} \\ |\mathcal{S}_f| \leq f}}\mathcal{L}(\hat{\boldsymbol{\theta}}; \mathcal{S} \setminus \mathcal{S}_f) - \mathcal{L}_{\star, \mathcal{S}\setminus\mathcal{S}_f}\right] + \frac{4R^2}{\mu}\left(\frac{1}{n} + \frac{L}{\mu}\left(\frac{f}{n}\right)^2\right)$$

$$\leq \frac{2L}{\mu}\alpha + \frac{4R^2}{\mu}\left(\frac{1}{n} + \frac{L}{\mu}\left(\frac{f}{n}\right)^2\right).$$

Therefore, ignoring dependencies in $L, \mu$, the in-distribution population risk is at most $\alpha$ (the factor $\frac{2L}{\mu}$ in the first term above can be removed at the cost of a logarithmic overhead in the time complexity) when $n = \Omega(\frac{1}{\alpha})$ and $f = \mathcal{O}(n\sqrt{\alpha})$.

Finally, we note that the unlearning time complexity can be further bounded, using the Lipschitz assumption, since we have from (Sekhari et al., 2021, Lemma 6) that

$$\max_{\substack{\mathcal{S}_f \subset \mathcal{S} \\ |\mathcal{S}_f| \leq f}}\left\|\boldsymbol{\theta}_{\mathcal{S}}^\star - \boldsymbol{\theta}_{\mathcal{S}\setminus\mathcal{S}_f}^\star\right\| \leq \frac{2Rf}{\mu n}.$$

Thus, ignoring dependencies on $L, \mu$, the unlearning time complexity is

$$\mathcal{O}\left(nd\log\left(1 + \frac{d}{\alpha\varepsilon}\left(\frac{Rf}{n}\right)^2\right)\right)$$

This concludes the proof. $\qquad\square$

**Remark 6** (Practical implementation of Algorithm 1 with gradient descent). *There are two practical scenarios where we can compute an upper bound on the number of optimization iterations needed to reach a predefined precision (in terms of squared distance to the empirical risk minimizer). Consider the optimizer to be gradient descent here for clarity, and denote the empirical loss $\mathcal{L}_{\text{emp}}$ , and the corresponding empirical risk minimizer $\boldsymbol{\theta}_{\star,\text{emp}}$.*
*The first scenario is when the loss function is non-negative (or some global lower bound is known); this is quite common in machine learning, e.g., quadratic loss, cross-entropy loss, hinge loss, etc... In this case, we know that for any initial model $\boldsymbol{\theta}_0 \in \mathbb{R}^d$ , we have $\|\boldsymbol{\theta}_0 - \boldsymbol{\theta}_{\star,\text{emp}}\|^2 \leq \frac{2}{\mu}(\mathcal{L}_{\text{emp}}(\boldsymbol{\theta}_0) - \mathcal{L}_{\text{emp}}(\boldsymbol{\theta}_{\star,\text{emp}})) \leq \frac{2}{\mu}\mathcal{L}_{\text{emp}}(\boldsymbol{\theta}_0)$, where the first inequality is due to $\mu$-strong convexity, and the second to the loss being non-negative. Therefore, knowing only the loss at the initial model, and (a lower*

*bound on) the strong convexity parameter, e.g., $\ell_2$-regularization factor, we have a computable upper bound on the initialization error. The upper bound on the number of iterations follows directly from standard first-order convergence analyses, e.g., see Theorem 2.1.15 of Nesterov et al. (2018), and is computable knowing the aforementioned bound on the initialization error, and the smoothness and strong convexity constants. The second scenario is when the parameter space is bounded, and in which case we use the projected variant of gradient descent, analyzed in Neel et al. (2021) for empirical risk minimization. There, we know that the initialization error is bounded by the diameter of the parameter space, which is computable. A computable upper bound on the number of iterations needed follows with a similar argument as the first scenario above.*

## D  PROOF OF PROPOSITION 2

**Proposition 2.** *Let $f = 1, n > 1$, and $\mathcal{Z} = \mathbb{R}^d$. There exists a 1-strongly convex and 1-smooth loss function, such that for any retain set $\mathcal{S}_r \in \mathcal{Z}^{n-1}$, any (unlearning-time) initialization error $\Delta > 0$, there exists a forget sample $\mathbf{z}_f \in \mathcal{Z}$ achieving $\left\| \boldsymbol{\theta}^\star_{\mathcal{S}_r \cup \{\mathbf{z}_f\}} - \boldsymbol{\theta}^\star_{\mathcal{S}_r} \right\|^2 = \Delta$, where $\boldsymbol{\theta}^\star_{\mathcal{S}_r}$ and $\boldsymbol{\theta}^\star_{\mathcal{S}_r \cup \{\mathbf{z}_f\}}$ denote the empirical minimizers on the retain and full data respectively.*

*Proof.* Consider the data space $\mathcal{Z} = \mathbb{R}^d$ and the quadratic loss $\ell(\boldsymbol{\theta}; \mathbf{z}) = \frac{1}{2} \|\boldsymbol{\theta} - \mathbf{z}\|^2, \forall \boldsymbol{\theta}, \mathbf{z} \in \mathbb{R}^d$. This loss function is 1-strongly convex and 1-smooth at any data point. Fix $\mathbf{z}_f := \boldsymbol{\theta}_{\star, \mathcal{S}_r} + n\sqrt{\Delta} \cdot \mathbf{u}$ for some unit vector $\mathbf{u} \in \mathbb{R}^d, \|\mathbf{u}\| = 1$. First, observe that

$$\boldsymbol{\theta}_{\star, \mathcal{S}_r} = \arg\min_{\boldsymbol{\theta} \in \mathbb{R}^d} \left\{ \mathcal{L}(\boldsymbol{\theta}; \mathcal{S}_r) = \frac{1}{2|\mathcal{S}_r|} \sum_{\mathbf{z} \in \mathcal{S}_r} \|\boldsymbol{\theta} - \mathbf{z}\|^2 \right\} = \frac{1}{|\mathcal{S}_r|} \sum_{\mathbf{z} \in \mathcal{S}_r} \mathbf{z},$$

and similarly $\boldsymbol{\theta}_{\star, \mathcal{S}_r \cup \{\mathbf{z}_f\}} = \arg\min_{\boldsymbol{\theta} \in \mathbb{R}^d} \mathcal{L}(\boldsymbol{\theta}; \mathcal{S}_r \cup \{\mathbf{z}_f\}) = \frac{1}{|\mathcal{S}_r|+1} \sum_{\mathbf{z} \in \mathcal{S}_r \cup \{\mathbf{z}_f\}} \mathbf{z}$. Therefore, we have

$$\boldsymbol{\theta}_{\star, \mathcal{S}_r \cup \{\mathbf{z}_f\}} = \frac{1}{|\mathcal{S}_r| + 1} \sum_{\mathbf{z} \in \mathcal{S}_r \cup \{\mathbf{z}_f\}} \mathbf{z} = \frac{1}{|\mathcal{S}_r| + 1} \left( \mathbf{z}_f + \sum_{\mathbf{z} \in \mathcal{S}_r} \mathbf{z} \right)$$

$$= \frac{1}{|\mathcal{S}_r| + 1} \left( \mathbf{z}_f + |\mathcal{S}_r| \boldsymbol{\theta}_{\star, \mathcal{S}_r} \right) = \frac{1}{n} \left( \mathbf{z}_f + (n-1)\boldsymbol{\theta}_{\star, \mathcal{S}_r} \right).$$

Finally, thanks to the choice $\mathbf{z}_f := \boldsymbol{\theta}_{\star, \mathcal{S}_r} + n\sqrt{\Delta} \cdot \mathbf{u}, \|\mathbf{u}\| = 1$, we conclude

$$\left\| \boldsymbol{\theta}_{\star, \mathcal{S}_r \cup \{\mathbf{z}_f\}} - \boldsymbol{\theta}_{\star, \mathcal{S}_r} \right\|^2 = \frac{1}{n^2} \|\mathbf{z}_f - \boldsymbol{\theta}_{\star, \mathcal{S}_r}\|^2 = \Delta.$$

$\square$

## E  PROOF OF THEOREM 3

**Lemma 7.** *Let $n \in \mathbb{N}^*$ and $f < n/2$. For any $\mathbf{g}_1, \ldots, \mathbf{g}_n \in \mathbb{R}^d$ and any $\mathcal{I} \subseteq [n]$ of size $|\mathcal{I}| \geq n - f$, we have*

$$\|\mathrm{TM}_f(\mathbf{g}_1, \ldots, \mathbf{g}_n) - \overline{\mathbf{g}}_{\mathcal{I}}\|^2 \leq \frac{6f}{n - 2f} \left( 1 + \frac{f}{n - 2f} \right) \frac{1}{|\mathcal{I}|} \sum_{i \in \mathcal{I}} \|\mathbf{g}_i - \overline{\mathbf{g}}_{\mathcal{I}}\|^2, \tag{20}$$

*where we denote the average $\overline{\mathbf{g}}_{\mathcal{I}} := \frac{1}{|\mathcal{I}|} \sum_{i \in \mathcal{I}} \mathbf{g}_i$.*

*Proof.* Let $n \in \mathbb{N}^*$ and $f < n/2$. Fix vectors $\mathbf{g}_1, \ldots, \mathbf{g}_n \in \mathbb{R}^d$ and subset $\mathcal{I} \subseteq [n]$ of size $|\mathcal{I}| = n - f$. For any set $\mathcal{T} \subseteq [n]$, we denote $\overline{\mathbf{g}}_{\mathcal{I}} := \frac{1}{|\mathcal{T}|} \sum_{i \in \mathcal{T}} \mathbf{g}_i := \mathbb{E}_{i \sim \mathcal{T}}[\mathbf{g}_i]$; the last notation is handy and refers to the expectation over the uniform sampling of $i$ from the set $\mathcal{T}$. First, observe that since each element in $\mathcal{I}$ has equal probability of belonging to a uniformly random subset $\mathcal{T} \subseteq \mathcal{I}, |\mathcal{T}| = n - f$, we have

$$\overline{\mathbf{g}}_{\mathcal{I}} := \frac{1}{|\mathcal{I}|} \sum_{i \in \mathcal{I}} \mathbf{g}_i = \mathbb{E}_{i \sim \mathcal{I}}[\mathbf{g}_i] = \mathbb{E}_{\substack{\mathcal{T} \sim \mathcal{I} \\ |\mathcal{T}| = n - f}}[\overline{\mathbf{g}}_{\mathcal{T}}], \tag{21}$$

where we recall that the last notation is the expectation over the uniform sampling over subsets of $\mathcal{I}$ of size $n - f$. Therefore, using Jensen's inequality, we have

$$\|\text{TM}_f(\mathbf{g}_1, \ldots, \mathbf{g}_n) - \overline{\mathbf{g}}_{\mathcal{I}}\|^2 = \left\| \text{TM}_f(\mathbf{g}_1, \ldots, \mathbf{g}_n) - \mathbb{E}_{\substack{\mathcal{T} \sim \mathcal{I} \\ |\mathcal{T}| = n - f}}[\overline{\mathbf{g}}_{\mathcal{T}}] \right\|^2$$

$$\leq \mathbb{E}_{\substack{\mathcal{T} \sim \mathcal{I} \\ |\mathcal{T}| = n - f}} \|\text{TM}_f(\mathbf{g}_1, \ldots, \mathbf{g}_n) - \overline{\mathbf{g}}_{\mathcal{T}}\|^2.$$

Now, recall from (Allouah et al., 2023, Proposition 2) that for every $\mathcal{T} \subseteq [n], |\mathcal{T}| = n - f$, we have

$$\|\text{TM}_f(\mathbf{g}_1, \ldots, \mathbf{g}_n) - \overline{\mathbf{g}}_{\mathcal{T}}\|^2 \leq \frac{6f}{n - 2f} \left( 1 + \frac{f}{n - 2f} \right) \frac{1}{|\mathcal{T}|} \sum_{i \in \mathcal{T}} \|\mathbf{g}_i - \overline{\mathbf{g}}_{\mathcal{T}}\|^2. \tag{22}$$

Plugging the above in the previous inequality and taking expectations yields

$$\|\text{TM}_f(\mathbf{g}_1, \ldots, \mathbf{g}_n) - \overline{\mathbf{g}}_{\mathcal{I}}\|^2 \leq \mathbb{E}_{\substack{\mathcal{T} \sim \mathcal{I} \\ |\mathcal{T}| = n - f}} \|\text{TM}_f(\mathbf{g}_1, \ldots, \mathbf{g}_n) - \overline{\mathbf{g}}_{\mathcal{T}}\|^2$$

$$\leq \frac{6f}{n - 2f} \left( 1 + \frac{f}{n - 2f} \right) \mathbb{E}_{\substack{\mathcal{T} \sim \mathcal{I} \\ |\mathcal{T}| = n - f}} \frac{1}{|\mathcal{T}|} \sum_{i \in \mathcal{T}} \|\mathbf{g}_i - \overline{\mathbf{g}}_{\mathcal{T}}\|^2$$

$$= \frac{6f}{n - 2f} \left( 1 + \frac{f}{n - 2f} \right) \mathbb{E}_{\substack{i \sim \mathcal{T} \\ \mathcal{T} \sim \mathcal{I} \\ |\mathcal{T}| = n - f}} \|\mathbf{g}_i - \overline{\mathbf{g}}_{\mathcal{T}}\|^2$$

$$\leq \frac{6f}{n - 2f} \left( 1 + \frac{f}{n - 2f} \right) \mathbb{E}_{i \sim \mathcal{I}} \|\mathbf{g}_i - \overline{\mathbf{g}}_{\mathcal{I}}\|^2$$

$$= \frac{6f}{n - 2f} \left( 1 + \frac{f}{n - 2f} \right) \frac{1}{|\mathcal{I}|} \sum_{i \in \mathcal{I}} \|\mathbf{g}_i - \overline{\mathbf{g}}_{\mathcal{I}}\|^2.$$

The third equality above is due to the same argument of (21), bias-variance decomposition, and Jensen's inequality. This concludes the proof. $\qquad\square$

**Lemma 8.** *Assume that, for every $\mathbf{z} \in \mathcal{Z}$, the loss $\ell(\cdot\,; \mathbf{z})$ is $\mu$-strongly convex and $L$-smooth. Let $\boldsymbol{\theta}_0 \in \mathbb{R}^d$, $f \leq n \min \left\{ \frac{1}{3}, \frac{12\mu}{5(L - \mu)} \right\}$, and consider the training Algorithm 2. Then, for any $T \geq 1$ and any $\mathcal{S}_f \in \mathcal{Z}^*, |\mathcal{S}_f| \leq f$, we have*

$$\mathcal{L}(\boldsymbol{\theta}_T; \mathcal{S}_r) - \mathcal{L}_{\star, \mathcal{S}_r} \leq \frac{45f}{\mu n} \frac{1}{|\mathcal{S}_r|} \sum_{\mathbf{z} \in \mathcal{S}_r} \|\nabla \ell(\boldsymbol{\theta}_{\mathcal{S}_r}^\star; \mathbf{z})\|^2 + \exp \left( -\frac{\mu}{2L} T \right) (\mathcal{L}(\boldsymbol{\theta}_0; \mathcal{S}_r) - \mathcal{L}_{\star, \mathcal{S}_r}), \tag{23}$$

*where we denoted $\boldsymbol{\theta}_{\mathcal{S}_r}^\star := \arg\min_{\boldsymbol{\theta} \in \mathbb{R}^d} \mathcal{L}(\boldsymbol{\theta}; \mathcal{S}_r)$.*

*Proof.* Let $t \geq 0$, $f \leq n \min \left\{ \frac{1}{3}, \frac{12\mu}{5(L - \mu)} \right\}$, and $\mathcal{S}_r, \mathcal{S}_f \in \mathcal{Z}^*$ such that $|\mathcal{S}_f| = n - |\mathcal{S}_r| \leq f$. Recall that $\mathcal{L}(\cdot\,; \mathcal{S} \setminus \mathcal{S}_f)$ is $L$-smooth by assumption. From Algorithm 2, recall that $\boldsymbol{\theta}_{t+1} = \boldsymbol{\theta}_t - \gamma \mathbf{r}_t$ with $\mathbf{r}_t := \text{TM}_f(\nabla \ell(\boldsymbol{\theta}_t; \mathbf{z}_1), \ldots, \nabla \ell(\boldsymbol{\theta}_t; \mathbf{z}_n))$. Hence, by the smoothness assumption, we have

$$\mathcal{L}(\boldsymbol{\theta}_{t+1}; \mathcal{S} \setminus \mathcal{S}_f) - \mathcal{L}(\boldsymbol{\theta}_t; \mathcal{S} \setminus \mathcal{S}_f) \leq -\gamma \langle \nabla \mathcal{L}(\boldsymbol{\theta}_t; \mathcal{S} \setminus \mathcal{S}_f), \mathbf{r}_t \rangle + \frac{1}{2} \gamma^2 L \|\mathbf{r}_t\|^2. \tag{24}$$

Moreover, we recall the identity

$$\langle \nabla \mathcal{L}(\boldsymbol{\theta}_t; \mathcal{S} \setminus \mathcal{S}_f), \mathbf{r}_t \rangle = \frac{1}{2} \left( \|\nabla \mathcal{L}(\boldsymbol{\theta}_t; \mathcal{S} \setminus \mathcal{S}_f)\|^2 + \|\mathbf{r}_t\|^2 - \|\nabla \mathcal{L}(\boldsymbol{\theta}_t; \mathcal{S} \setminus \mathcal{S}_f) - \mathbf{r}_t\|^2 \right).$$

Substituting the above in (24) we obtain that

$$\mathcal{L}(\boldsymbol{\theta}_{t+1}; \mathcal{S} \setminus \mathcal{S}_f) - \mathcal{L}(\boldsymbol{\theta}_t; \mathcal{S} \setminus \mathcal{S}_f)$$

$$\leq -\frac{\gamma}{2} \left( \|\nabla \mathcal{L}(\boldsymbol{\theta}_t; \mathcal{S} \setminus \mathcal{S}_f)\|^2 + \|\mathbf{r}_t\|^2 - \|\nabla \mathcal{L}(\boldsymbol{\theta}_t; \mathcal{S} \setminus \mathcal{S}_f) - \mathbf{r}_t\|^2 \right) + \frac{1}{2} \gamma^2 L \|\mathbf{r}_t\|^2$$

$$= -\frac{\gamma}{2} \|\nabla \mathcal{L}(\boldsymbol{\theta}_t; \mathcal{S} \setminus \mathcal{S}_f)\|^2 - \frac{\gamma}{2} (1 - \gamma L) \|\mathbf{r}_t\|^2 + \frac{\gamma}{2} \|\nabla \mathcal{L}(\boldsymbol{\theta}_t; \mathcal{S} \setminus \mathcal{S}_f) - \mathbf{r}_t\|^2.$$

Substituting $\gamma = \frac{1}{L}$ in the above we obtain that

$$\mathcal{L}(\boldsymbol{\theta}_{t+1}; \mathcal{S} \setminus \mathcal{S}_f) - \mathcal{L}(\boldsymbol{\theta}_t; \mathcal{S} \setminus \mathcal{S}_f) \leq -\frac{1}{2L} \|\nabla \mathcal{L}(\boldsymbol{\theta}_t)\|^2 + \frac{1}{2L} \|\mathbf{r}_t - \nabla \mathcal{L}(\boldsymbol{\theta}_t; \mathcal{S} \setminus \mathcal{S}_f)\|^2 . \quad (25)$$

By applying Lemma 7 to the vectors $\nabla \ell(\boldsymbol{\theta}_t; \mathbf{z}_1), \ldots, \nabla \ell(\boldsymbol{\theta}_t; \mathbf{z}_n)$ and the indices set $\mathcal{I} := \{i \in [n]: \mathbf{z}_i \in \mathcal{S}_r\}$, and denoting $\kappa := \frac{6f}{n-2f} \left(1 + \frac{f}{n-2f}\right)$, we obtain

$$\|\mathbf{r}_t - \nabla \mathcal{L}(\boldsymbol{\theta}_t; \mathcal{S} \setminus \mathcal{S}_f)\|^2 = \left\| \mathrm{TM}_f(\nabla \ell(\boldsymbol{\theta}_t; \mathbf{z}_1), \ldots, \nabla \ell(\boldsymbol{\theta}_t; \mathbf{z}_n)) - \frac{1}{|\mathcal{I}|} \sum_{j \in \mathcal{I}} \nabla \ell(\boldsymbol{\theta}_t; \mathbf{z}_j) \right\|^2$$

$$\leq \frac{\kappa}{|\mathcal{I}|} \sum_{i \in \mathcal{I}} \left\| \nabla \ell(\boldsymbol{\theta}_t; \mathbf{z}_i) - \frac{1}{|\mathcal{I}|} \sum_{j \in \mathcal{I}} \nabla \ell(\boldsymbol{\theta}_t; \mathbf{z}_j) \right\|^2 \quad (26)$$

$$= \frac{\kappa}{|\mathcal{I}|} \sum_{i \in \mathcal{I}} \|\nabla \ell(\boldsymbol{\theta}_t; \mathbf{z}_i) - \nabla \mathcal{L}(\boldsymbol{\theta}_t; \mathcal{S} \setminus \mathcal{S}_f)\|^2 . \quad (27)$$

Besides, by denoting $\zeta_\star^2 := \frac{2}{|\mathcal{S}_r|} \sum_{\mathbf{z} \in \mathcal{S}_r} \|\nabla \ell(\boldsymbol{\theta}_{\mathcal{S}_r}^\star; \mathbf{z})\|^2$ and $P := \frac{2L}{\mu}$, we have thanks to strong convexity and smoothness (see, e.g., (Allouah et al., 2024, Proposition 1)) that

$$\frac{1}{|\mathcal{I}|} \sum_{i \in \mathcal{I}} \|\nabla \ell(\boldsymbol{\theta}_t; \mathbf{z}_i)\|^2 - \|\nabla \mathcal{L}(\boldsymbol{\theta}_t; \mathcal{S} \setminus \mathcal{S}_f)\|^2 \leq \zeta_\star^2 + (P-1) \|\nabla \mathcal{L}(\boldsymbol{\theta}_t; \mathcal{S} \setminus \mathcal{S}_f)\|^2 . \quad (28)$$

$$\frac{1}{|\mathcal{I}|} \sum_{i \in \mathcal{I}} \|\nabla \ell(\boldsymbol{\theta}_t; \mathbf{z}_i) - \nabla \mathcal{L}(\boldsymbol{\theta}_t; \mathcal{S} \setminus \mathcal{S}_f)\|^2 = \frac{1}{|\mathcal{I}|} \sum_{i \in \mathcal{I}} \|\nabla \ell(\boldsymbol{\theta}_t; \mathbf{z}_i)\|^2 - \|\nabla \mathcal{L}(\boldsymbol{\theta}_t; \mathcal{S} \setminus \mathcal{S}_f)\|^2$$

$$\leq \zeta_\star^2 + (P-1) \|\nabla \mathcal{L}(\boldsymbol{\theta}_t; \mathcal{S} \setminus \mathcal{S}_f)\|^2 .$$

Using the above in (27) yields

$$\|\mathbf{r}_t - \nabla \mathcal{L}(\boldsymbol{\theta}_t; \mathcal{S} \setminus \mathcal{S}_f)\|^2 \leq \kappa \zeta_\star^2 + \kappa(P-1) \|\nabla \mathcal{L}(\boldsymbol{\theta}_t; \mathcal{S} \setminus \mathcal{S}_f)\|^2 .$$

Substituting the above in (25) yields

$$\mathcal{L}(\boldsymbol{\theta}_{t+1}; \mathcal{S} \setminus \mathcal{S}_f) - \mathcal{L}(\boldsymbol{\theta}_t; \mathcal{S} \setminus \mathcal{S}_f) \leq -\frac{1}{2L} \|\nabla \mathcal{L}(\boldsymbol{\theta}_t; \mathcal{S} \setminus \mathcal{S}_f)\|^2 + \frac{1}{2L} \left( \kappa \zeta_\star^2 + \kappa(P-1) \|\nabla \mathcal{L}(\boldsymbol{\theta}_t); \mathcal{S} \setminus \mathcal{S}_f\|^2 \right) . \quad (29)$$

Multiplying both sides in (29) by $2L$ and rearranging terms, we get

$$(1 - \kappa(P-1)) \|\nabla \mathcal{L}(\boldsymbol{\theta}_t; \mathcal{S} \setminus \mathcal{S}_f)\|^2 \leq \kappa \zeta_\star^2 + 2L \left( \mathcal{L}(\boldsymbol{\theta}_t; \mathcal{S} \setminus \mathcal{S}_f) - \mathcal{L}(\boldsymbol{\theta}_{t+1}; \mathcal{S} \setminus \mathcal{S}_f) \right)$$

$$= \kappa \zeta_\star^2 + 2L \left( \mathcal{L}(\boldsymbol{\theta}_t) - \mathcal{L}_{\star, \mathcal{S} \setminus \mathcal{S}_f} + \mathcal{L}_{\star, \mathcal{S} \setminus \mathcal{S}_f} - \mathcal{L}(\boldsymbol{\theta}_{t+1}) \right) .$$

After rearranging terms, and using strong convexity to lower bound the norm of the gradient with the optimality gap, e.g., see (Karimi et al., 2016), we obtain

$$2L \left( \mathcal{L}(\boldsymbol{\theta}_{t+1}; \mathcal{S} \setminus \mathcal{S}_f) - \mathcal{L}_{\star, \mathcal{S} \setminus \mathcal{S}_f} \right) \leq \kappa \zeta_\star^2 - 2\mu \left( 1 - \kappa(P-1) \right) \left( \mathcal{L}(\boldsymbol{\theta}_t; \mathcal{S} \setminus \mathcal{S}_f) - \mathcal{L}_{\star, \mathcal{S} \setminus \mathcal{S}_f} \right)$$

$$+ 2L \left( \mathcal{L}(\boldsymbol{\theta}_t; \mathcal{S} \setminus \mathcal{S}_f) - \mathcal{L}_{\star, \mathcal{S} \setminus \mathcal{S}_f} \right)$$

$$= \kappa \zeta_\star^2 + \left( 2L - 2\mu \left( 1 - \kappa(P-1) \right) \right) \left( \mathcal{L}(\boldsymbol{\theta}_t; \mathcal{S} \setminus \mathcal{S}_f) - \mathcal{L}_{\star, \mathcal{S} \setminus \mathcal{S}_f} \right) .$$

Dividing both sides by $2L$, we get

$$\mathcal{L}(\boldsymbol{\theta}_{t+1}; \mathcal{S} \setminus \mathcal{S}_f) - \mathcal{L}_{\star, \mathcal{S} \setminus \mathcal{S}_f} \leq \frac{\kappa \zeta_\star^2}{2L} + \left( 1 - \frac{\mu}{L} \left( 1 - \kappa(P-1) \right) \right) \left( \mathcal{L}(\boldsymbol{\theta}_t; \mathcal{S} \setminus \mathcal{S}_f) - \mathcal{L}_{\star, \mathcal{S} \setminus \mathcal{S}_f} \right) . \quad (30)$$

Then, applying (30) recursively for time indices in $k \in \{0, \ldots, t-1\}$ yields

$$\mathcal{L}(\boldsymbol{\theta}_{t+1}; \mathcal{S} \setminus \mathcal{S}_f) - \mathcal{L}_{\star, \mathcal{S} \setminus \mathcal{S}_f} \leq \frac{\kappa \zeta_\star^2}{2L} \sum_{k=0}^{t} \left(1 - \frac{\mu}{L}\left(1 - \kappa(P-1)\right)\right)^k$$

$$+ \left(1 - \frac{\mu}{L}\left(1 - \kappa(P-1)\right)\right)^{t+1} \left(\mathcal{L}(\boldsymbol{\theta}_0; \mathcal{S} \setminus \mathcal{S}_f) - \mathcal{L}_{\star, \mathcal{S} \setminus \mathcal{S}_f}\right)$$

$$\leq \frac{\kappa \zeta_\star^2}{2L} \frac{1}{1 - \left(1 - \frac{\mu}{L}\left(1 - \kappa(P-1)\right)\right)} + \left(1 - \frac{\mu}{L}\left(1 - \kappa(P-1)\right)\right)^{t+1} \left(\mathcal{L}(\boldsymbol{\theta}_0; \mathcal{S} \setminus \mathcal{S}_f) - \mathcal{L}_{\star, \mathcal{S} \setminus \mathcal{S}_f}\right)$$

$$= \frac{\kappa \zeta_\star^2}{2\mu\left(1 - \kappa(P-1)\right)} + \left(1 - \frac{\mu}{L}\left(1 - \kappa(P-1)\right)\right)^{t+1} \left(\mathcal{L}(\boldsymbol{\theta}_0; \mathcal{S} \setminus \mathcal{S}_f) - \mathcal{L}_{\star, \mathcal{S} \setminus \mathcal{S}_f}\right).$$

We now take $t = T - 1 \geq 0$. Moreover, using the fact that $(1+x)^n \leq e^{nx}$ for all $x \in \mathbb{R}$ and remarking that $\frac{\kappa}{1-\kappa(P-1)} \leq 45\frac{f}{n}$ and $1 - \kappa(P-1) \geq \frac{1}{2}$ when $\frac{f}{n} \leq \min\left\{\frac{1}{3}, \frac{12}{5(P-1)}\right\}$ yields that

$$\mathcal{L}(\boldsymbol{\theta}_T; \mathcal{S} \setminus \mathcal{S}_f) - \mathcal{L}_{\star, \mathcal{S} \setminus \mathcal{S}_f} \leq \frac{45}{2\mu} \frac{f}{n} \zeta_\star^2 + \exp\left(-\frac{\mu}{2L}T\right) \left(\mathcal{L}(\boldsymbol{\theta}_0; \mathcal{S} \setminus \mathcal{S}_f) - \mathcal{L}_{\star, \mathcal{S} \setminus \mathcal{S}_f}\right).$$

This concludes the proof. $\qquad\square$

**Theorem 3.** *Let $\varepsilon, \alpha, \alpha_{\mathrm{emp}}, \mu, L > 0, q > 1, \boldsymbol{\theta}_0 \in \mathbb{R}^d$, and $f \leq n \min\left\{\frac{1}{3}, \frac{12\mu}{5(L-\mu)}\right\}$. Assume that, for every $\mathbf{z} \in \mathcal{Z}$, the loss $\ell(\cdot\,; \mathbf{z})$ is $\mu$-strongly convex and $L$-smooth. Consider the unlearning-training pair $(\mathcal{U}, \mathcal{A})$ in Algorithm 2 using gradient descent during unlearning.*

*Then, $(\mathcal{U}, \mathcal{A})$ satisfies $(q, q\varepsilon)$-approximate unlearning with empirical loss, over $\mathcal{S}_r$ with worst-case $\mathcal{S}_f$, at most $\alpha_{\mathrm{emp}}$ in expectation over the randomness of the algorithm with time complexity:*

*Training:* $\mathcal{O}\left(nd\log\left(\frac{d}{\alpha_{\mathrm{emp}}\varepsilon}\left\|\boldsymbol{\theta}_0 - \boldsymbol{\theta}_{\mathcal{S}_r}^\star\right\|^2\right)\right)$, *Unlearning:* $\mathcal{O}\left(nd\log\left(1 + \frac{d}{\alpha_{\mathrm{emp}}\varepsilon}\frac{f}{n}\mathcal{E}(\mathcal{S}_r)\right)\right)$,

*ignoring dependencies on $L, \mu$. The space complexity is $\mathcal{O}(d)$ during training and unlearning. For $\alpha_{\mathrm{emp}} \leq \alpha$, the out-of-distribution risk $\mathcal{L}_{\mathrm{OOD}}(\mathcal{U}, \mathcal{A})$ is at most $\alpha$, if $n - f = \Omega(\frac{1}{\alpha})$, with time complexity:*

*Training:* $\mathcal{O}\left(nd\log\left(\frac{d}{\alpha\varepsilon}\mathbb{E}_{\mathcal{S}_r}\left\|\boldsymbol{\theta}_0 - \boldsymbol{\theta}_{\mathcal{S}_r}^\star\right\|^2\right)\right)$, *Unlearning:* $\mathcal{O}\left(nd\log\left(1 + \frac{d}{\alpha\varepsilon}\frac{f}{n}\mathcal{E}_{n-f}(\mathcal{D})\right)\right)$.

*Proof.* Let $q > 1, \varepsilon, \alpha, \alpha_{\mathrm{emp}}, \alpha' > 0, 0 \leq f < n$, and $\mathcal{S}_r \in \mathcal{Z}^{n-f}, \mathcal{S} := \mathcal{S}_r \cup \mathcal{S}_f \in \mathcal{Z}^n$. Assume that the loss function is $\mu$-strongly convex and $L$-smooth at any data point, and that $\varepsilon \leq d$. Denote by $\boldsymbol{\theta}_{\mathcal{S}_r}^\star$ and $\boldsymbol{\theta}_{\mathcal{S}}^\star$ the minimizers of the empirical loss functions $\mathcal{L}(\cdot\,; \mathcal{S}_r)$ and $\mathcal{L}(\cdot\,; \mathcal{S})$, respectively. These exist and are well-defined by strong convexity of the loss function. Also, following Algorithm 2, denote by $\boldsymbol{\theta}_{\mathcal{S}}^{\mathcal{A}_f}$ and $\boldsymbol{\theta}_{\mathcal{S}_r}^{\mathcal{A}}$ the model obtained using $\mathcal{A}_f$ over the training set $\mathcal{S}$ and obtained using $\mathcal{A}_0$ (i.e., empty forget set $f = 0$) over the training set $\mathcal{S}_r$, respectively. Moreover, denote by $\boldsymbol{\theta}^{\mathcal{U}}$ the model obtained after using $\mathcal{U}$ over the training set $\mathcal{S}_r$ before Gaussian noise addition.

Recall that the computational cost of reaching the empirical risk minimizer up to squared error $\alpha_{\mathrm{emp}} > 0$, starting with an initialization error $\Delta > 0$, with gradient on smooth strongly convex problems is $\mathcal{O}(nd\frac{L}{\mu}\log(\frac{\Delta}{\alpha_{\mathrm{emp}}}))$ (Nesterov et al., 2018). Therefore, letting $\Delta := \left\|\boldsymbol{\theta}_0 - \boldsymbol{\theta}_{\mathcal{S}_r}^\star\right\|^2$, at the computational cost of $\mathcal{O}(nd\frac{L}{\mu}\log(\frac{Ld\Delta}{\alpha_{\mathrm{emp}}\varepsilon}))$ during training and $\max_{\substack{\mathcal{S}_f \subset \mathcal{S} \\ |\mathcal{S}_f| \leq f}} \mathcal{O}(nd\frac{L}{\mu}\log(\frac{Ld\left\|\boldsymbol{\theta}_{\mathcal{S}}^{\mathcal{A}_f} - \boldsymbol{\theta}_{\mathcal{S}_r}^\star\right\|^2}{\alpha_{\mathrm{emp}}\varepsilon}))$ during unlearning since we initialize at $\boldsymbol{\theta}_{\mathcal{S}}^{\mathcal{A}_f}$, we have by definition

$$\max_{\substack{\mathcal{S}_f \subset \mathcal{S} \\ |\mathcal{S}_f| \leq f}} \left\|\boldsymbol{\theta}^{\mathcal{U}} - \boldsymbol{\theta}_{\mathcal{S}_r}^\star\right\|^2 \leq \frac{\alpha_{\mathrm{emp}}\varepsilon}{4Ld}, \quad \left\|\boldsymbol{\theta}_{\mathcal{S}_r}^{\mathcal{A}} - \boldsymbol{\theta}_{\mathcal{S}_r}^\star\right\|^2 \leq \frac{\alpha_{\mathrm{emp}}\varepsilon}{4Ld}. \tag{31}$$

Also, recalling the result of Lemma 8 and using strong convexity, we have

$$\max_{\substack{\mathcal{S}_f \subset \mathcal{S} \\ |\mathcal{S}_f| \leq f}} \left\|\boldsymbol{\theta}_{\mathcal{S}}^{\mathcal{A}_f} - \boldsymbol{\theta}_{\mathcal{S}_r}^\star\right\|^2 \leq \frac{2}{\mu} \max_{\substack{\mathcal{S}_f \subset \mathcal{S} \\ |\mathcal{S}_f| \leq f}} \left(\mathcal{L}(\boldsymbol{\theta}_{\mathcal{S}}^{\mathcal{A}_f}; \mathcal{S}_r) - \mathcal{L}_{\star, \mathcal{S}_r}\right) \leq \frac{2}{\mu}\left(\frac{45f}{\mu n}\frac{1}{|\mathcal{S}_r|}\sum_{\mathbf{z} \in \mathcal{S}_r}\left\|\nabla\ell(\boldsymbol{\theta}_{\mathcal{S}_r}^\star; \mathbf{z})\right\|^2 + \alpha'\right).$$

$$\tag{32}$$

Also, from Lemma 8, the computational complexity for the statement above is $\mathcal{O}(nd \log\left(\frac{\Delta}{\alpha'}\right))$ (to be added during training), as the per-iteration cost involves computing $d$ times a trimmed mean over $n$ real numbers, each of which can be done in worst-case linear time and constant space with variations of the median-of-medians (Blum et al., 1973; Lai and Wood, 1988). Thus, the computational complexity of unlearning is upper bounded by

$$\mathcal{O}\left(nd\frac{L}{\mu}\log\left(\frac{Ld\left(\frac{f}{\mu n}\frac{1}{|\mathcal{S}_r|}\sum_{\mathbf{z}\in\mathcal{S}_r}\left\|\nabla\ell(\boldsymbol{\theta}^\star_{\mathcal{S}_r};\mathbf{z})\right\|^2+\alpha'\right)}{\mu\alpha_{\mathrm{emp}}\varepsilon}\right)\right). \tag{33}$$

**Unlearning analysis.**   Our goal here is to show that $\mathcal{U}(\mathcal{S}_f,\mathcal{A}_f(\mathcal{S}))$ and $\mathcal{U}(\varnothing,\mathcal{A}_0(\mathcal{S}_r))$ are near-indistinguishable in the sense of Definition 1. To do so, we bound the distance between $\boldsymbol{\theta}^{\mathcal{A}}_{\mathcal{S}_r}$ and $\boldsymbol{\theta}^{\mathcal{U}}$, and infer the unlearning guarantee via the Rényi divergence bound of the Gaussian mechanism.

Now, using inequalities (31) and the triangle inequality, we obtain

$$\left\|\boldsymbol{\theta}^{\mathcal{U}}-\boldsymbol{\theta}^{\mathcal{A}}_{\mathcal{S}_r}\right\|^2\leq 2\left\|\boldsymbol{\theta}^{\mathcal{U}}-\boldsymbol{\theta}^\star_{\mathcal{S}_r}\right\|^2+2\left\|\boldsymbol{\theta}^{\mathcal{A}}_{\mathcal{S}_r}-\boldsymbol{\theta}^\star_{\mathcal{S}_r}\right\|^2\leq\frac{\alpha_{\mathrm{emp}}\varepsilon}{Ld}. \tag{34}$$

Recall that $\mathcal{U}(\mathcal{S}_f,\mathcal{A}_f(\mathcal{S})) := \boldsymbol{\theta}^{\mathcal{U}}+\mathcal{N}(0,\frac{\alpha_{\mathrm{emp}}}{2Ld}\mathbf{I}_d)$ and $\mathcal{U}(\varnothing,\mathcal{A}(\mathcal{S}_r)) := \boldsymbol{\theta}^{\mathcal{A}}_{\mathcal{S}\setminus\mathcal{S}_f}+\mathcal{N}(0,\frac{\alpha_{\mathrm{emp}}}{Ld}\mathbf{I}_d)$. Thus, using the Rényi divergence expression between multivariate Gaussians (e.g., see (Gil et al., 2013)) we conclude that $(\mathcal{U},\mathcal{A})$ satisfies $(q,q\varepsilon)$-approximate unlearning:

$$D_q(\mathcal{U}(\mathcal{S}_f,\mathcal{A}_f(\mathcal{S}))\,\|\,\mathcal{U}(\varnothing,\mathcal{A}(\mathcal{S}_r))) = \frac{q}{2\cdot\frac{\alpha_{\mathrm{emp}}}{2Ld}}\left\|\boldsymbol{\theta}^{\mathcal{U}}-\boldsymbol{\theta}^{\mathcal{A}}_{\mathcal{S}_r}\right\|^2\leq q\varepsilon. \tag{35}$$

**Utility analysis.**   We now analyze the empirical and population loss of the model $\mathcal{U}(\mathcal{S}_f,\mathcal{A}_f(\mathcal{S}))$.

Recall that the loss function is $L$-smooth, and that the retain set $\mathcal{S}_r$ is fixed. Therefore, using inequalities (31), Jensen's inequality and taking expectations over the randomness of the additive Gaussian noise $\mathcal{N}(0,\frac{\alpha_{\mathrm{emp}}}{2Ld}\mathbf{I}_d)$, we can bound the empirical loss:

$$\mathbb{E}_{\mathcal{U}}[\max_{\substack{\mathcal{S}_f\in\mathcal{Z}^*\\|\mathcal{S}_f|\leq f}}\mathcal{L}(\mathcal{U}(\mathcal{S}_f,\mathcal{A}_f(\mathcal{S}));\mathcal{S}_r)]-\mathcal{L}_{\star,\mathcal{S}_r}\leq\frac{L}{2}\mathbb{E}_{\mathcal{U}}[\max_{\substack{\mathcal{S}_f\in\mathcal{Z}^*\\|\mathcal{S}_f|\leq f}}\left\|\mathcal{U}(\mathcal{S}_f,\mathcal{A}_f(\mathcal{S}))-\boldsymbol{\theta}^\star_{\mathcal{S}_r}\right\|^2]$$

$$=\frac{L}{2}\mathbb{E}_{\mathbf{X}\sim\mathcal{N}(0,\frac{\alpha_{\mathrm{emp}}}{2Ld}\mathbf{I}_d)}[\max_{\substack{\mathcal{S}_f\in\mathcal{Z}^*\\|\mathcal{S}_f|\leq f}}\left\|\boldsymbol{\theta}_{\mathcal{U}}-\boldsymbol{\theta}^\star_{\mathcal{S}_r}+\mathbf{X}\right\|^2]$$

$$\leq L\max_{\substack{\mathcal{S}_f\in\mathcal{Z}^*\\|\mathcal{S}_f|\leq f}}\left\|\boldsymbol{\theta}^{\mathcal{U}}-\boldsymbol{\theta}^\star_{\mathcal{S}_r}\right\|^2+L\,\mathbb{E}_{\mathbf{X}\sim\mathcal{N}(0,\frac{\alpha_{\mathrm{emp}}}{2Ld}\mathbf{I}_d)}\left\|\mathbf{X}\right\|^2$$

$$=L\max_{\substack{\mathcal{S}_f\in\mathcal{Z}^*\\|\mathcal{S}_f|\leq f}}\left\|\boldsymbol{\theta}^{\mathcal{U}}-\boldsymbol{\theta}^\star_{\mathcal{S}_r}\right\|^2+Ld\frac{\alpha_{\mathrm{emp}}}{2Ld}\leq L\frac{\alpha_{\mathrm{emp}}\varepsilon}{4Ld}+\frac{\alpha_{\mathrm{emp}}}{2}\leq\alpha_{\mathrm{emp}}, \tag{36}$$

after using the assumption that $\varepsilon\leq d$ for the last inequality. Thus, the expected empirical risk error is at most $\alpha_{\mathrm{emp}}$ with the following computational complexities before and during unlearning respectively:

$$\mathcal{O}\left(nd\frac{L}{\mu}\max\left\{\log\left(\frac{\Delta}{\alpha'}\right),\log\left(\frac{Ld\Delta}{\alpha_{\mathrm{emp}}\varepsilon}\right)\right\}\right),\ \mathcal{O}\left(nd\frac{L}{\mu}\log\left(\frac{Ld\left(\frac{f}{\mu n}\frac{1}{|\mathcal{S}_r|}\sum_{\mathbf{z}\in\mathcal{S}_r}\left\|\nabla\ell(\boldsymbol{\theta}^\star_{\mathcal{S}_r};\mathbf{z})\right\|^2+\alpha'\right)}{\mu\alpha_{\mathrm{emp}}\varepsilon}\right)\right). \tag{37}$$

Finally, we set $\alpha'=\frac{\alpha_{\mathrm{emp}}\mu\varepsilon}{Ld}$ to conclude the first statement of the theorem.

In fact, we can obtain the guarantee (36) in expectation over $\mathcal{S}_r\sim\mathcal{D}^{n-f}$, at the cost of the expectation of the runtimes above. Indeed, taking expectations over the training set in the standard convergence guarantee of gradient descent for smooth strongly convex problems (e.g., (Nesterov et al., 2018, Theorem 2.1.15) implies that expected error $\alpha$ with expected initialization error $\mathbb{E}_{\mathcal{S}_r}[\Delta]$ can be achieved in $\mathcal{O}(nd\frac{L}{\mu}\log\frac{\mathbb{E}_{\mathcal{S}_r}[\Delta]}{\alpha})$ time. That is, we have

$$\mathbb{E}_{\mathcal{S}_r\sim\mathcal{D}^{n-f}}\left[\max_{\substack{\mathcal{S}_f\in\mathcal{Z}^*\\|\mathcal{S}_f|\leq f}}\mathcal{L}(\mathcal{U}(\mathcal{S}_f,\mathcal{A}_f(\mathcal{S}_r\cup\mathcal{S}_f));\mathcal{S}_r)-\mathcal{L}_{\star,\mathcal{S}_r}\right]\leq\alpha, \tag{38}$$

with the following training and unlearning time respectively:

$$\mathcal{O}\left(nd\log\left(\frac{d}{\alpha\varepsilon}\,\mathbb{E}_{\mathcal{S}_r\sim\mathcal{D}^{n-f}}\left\|\boldsymbol{\theta}_0-\boldsymbol{\theta}_{\mathcal{S}_r}^\star\right\|^2\right)\right),$$

$$\mathcal{O}\left(nd\log\left(1+\frac{d}{\alpha\varepsilon}\frac{f}{n}\,\mathbb{E}_{\mathcal{S}_r\sim\mathcal{D}^{n-f}}\frac{1}{|\mathcal{S}_r|}\sum_{\mathbf{z}\in\mathcal{S}_r}\left\|\nabla\ell(\boldsymbol{\theta}_{\mathcal{S}_r}^\star;\mathbf{z})\right\|^2\right)\right).$$

In turn, we can plug the bound (38) in the generalization bound of Proposition 1. As a result, we have

$$\mathcal{L}_{\mathrm{OOD}}(\mathcal{U},\mathcal{A})\leq\frac{L}{\mu}\,\mathbb{E}_{\mathcal{S}_r\sim\mathcal{D}^{n-f}}\big[\max_{\substack{\mathcal{S}_f\in\mathcal{Z}^\star\\|\mathcal{S}_f|\leq f}}\mathcal{L}(\mathcal{U}(\mathcal{S}_f,\mathcal{A}_f(\mathcal{S}_r\cup\mathcal{S}_f));\mathcal{S}_r)-\mathcal{L}_{\star,\mathcal{S}_r}\big]+\frac{L}{2\mu^2}\frac{\mathbb{E}_{\mathbf{z}\sim\mathcal{D}}\left\|\nabla\ell(\boldsymbol{\theta}^\star;\mathbf{z})\right\|^2}{n-f}$$

$$\leq\frac{L}{\mu}\alpha+\frac{L}{2\mu^2}\frac{\mathbb{E}_{\mathbf{z}\sim\mathcal{D}}\left\|\nabla\ell(\boldsymbol{\theta}^\star;\mathbf{z})\right\|^2}{n-f}.$$

Therefore, ignoring dependencies in $L,\mu$, the in-distribution population risk is at most $\alpha$ (the factor $\frac{L}{\mu}$ in the first term above can be removed at the cost of a logarithmic overhead in the time complexity) when $n-f=\Omega(\frac{1}{\alpha})$. This concludes the proof. $\qquad\square$

## F    ADDITIONAL DETAILS AND RESULTS FOR TABLES 1 AND 2 AND FIGURE 1

**Tables 1 and 2.**    We recall that Table 1 presents a summary of the in-distribution deletion capacities (the larger, the better), for error bound $\alpha>0$ and computation budget $T>0$, under approximate unlearning for strongly convex tasks, with smoothness and Lipschitz assumptions. In particular, we note that the utility deletion capacity of the Newton step algorithm is directly deduced from (Sekhari et al., 2021, Theorem 3), with the following adaption from $(\varepsilon_{\mathrm{DP}},\delta)$-unlearning (Sekhari et al., 2021, Definition 2) to our $(q,q\varepsilon)$-unlearning formalism for all $q>1$: $\varepsilon\leq\frac{\varepsilon_{\mathrm{DP}}^2}{16\log(1/\delta)}$ assuming that $\varepsilon_{\mathrm{DP}}\leq\log(1/\delta)$ using (Mironov, 2017, Proposition 3). This adaptation is valid given that the Gaussian mechanism employed (Guo et al., 2020; Sekhari et al., 2021) satisfies Rényi differential privacy (Mironov, 2017, Corollary 3). The same adaptation is conducted for the differential privacy method (Huang and Canonne, 2023). Besides, we note that the last two reported computational capacities mean that no sample can be unlearned unless $T$ exceeds the proven time complexity of these algorithms (Bassily et al., 2014; Sekhari et al., 2021). Finally, the reported lower bound is a direct adaptation of (Lai et al., 2016, Observation 1.4).
In both tables 1 and 2, the computational deletion capacities are obtained by computing the maximum deleted samples $f$ such that the sum of the training and unlearning time complexities is smaller than $T$. We believe that comparing the sum of both unlearning and training times is important. For the sake of the argument, a hypothetical (inefficient) training procedure that computes and stores the empirical risk minimizers on every subset of the full dataset would only require constant-time unlearning, which could be misleading if we only compare unlearning times. It is worth noting here that the unlearning time complexity of the work of Sekhari et al. (2021) is independent of $n$, although at least quadratic in $d$.

**Figure 1.**    We recall that Figure 1 presents a numerical validation of our theoretical results on a linear regression task with synthetic data under a fixed approximate unlearning budget, with in-distribution and out-of-distribution data. Specifically, the full data features are generated from a $d$-dimensional Gaussian $\mathcal{N}(0,\mathbf{I}_d), d=100$, and the labels are generated from the features and a random true underlying model, also from a Gaussian $\mathcal{N}(0,\mathbf{I}_d)$, with a Gaussian response. The in-distribution forget data consists of $f=20$ data points sampled at random from the $10,000$ full training samples. Moreover, we set the unlearning budget for the in-distribution scenario to $\varepsilon=1$. We use the DP-SGD implementation of Opacus (Yousefpour et al., 2021), and convert to group differential privacy, the group being of size $f$, with standard conversion bounds (Vadhan, 2017, Lemma 2.2), and run the optimizer until convergence or for 100 epochs (usually $10\times$ more than Algorithm 1) with a fine-tuned learning rate. For the out-of-distribution scenario, the forget data is obtained by shifting labels with a fixed offset set to $10^3$, the total number of training samples being $1,000$. Moreover, we set the unlearning budget for the in-distribution scenario to $\varepsilon=10$. The learning rates for Algorithm 1 and 2 are set following standard theoretical convergence rates (Nesterov et al., 2018) for strongly convex

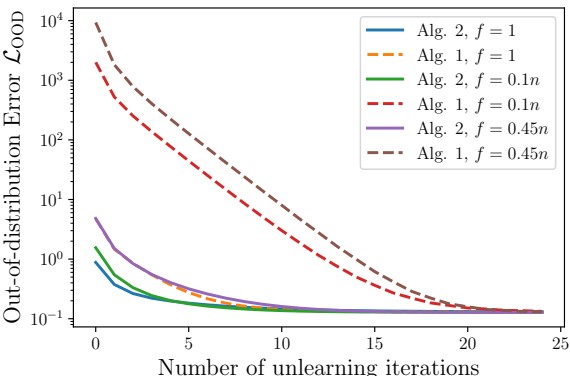

Figure 2: Out-of-distribution error $\mathcal{L}_{\text{OOD}}$ versus number of *unlearning* iterations for Algorithms 1 and 2, using gradient descent as optimizer, with $f \in \{1, 0.1n, 0.45n\}$ forget data out of $16,512$ samples of the *California Housing* dataset. The per-iteration cost is the same for both algorithms. The unlearning time of Algorithm 1 (non-robust) can be $10\times$ slower than Algorithm 2.

tasks, and Theorem 3. The initialization error can be estimated without knowing the distance to the minimizer, since the loss is known to be non-negative in this case, and the strong convexity constant is estimated empirically (see Remark 6 for details).

**Empirical validation on real data.**    We extend the empirical validation in Figure 1 in the out-of-distribution scenario to the *California Housing* dataset (Pace and Barry, 1997), a standard regression benchmark. The dataset contains $20,640$ samples, each representing a district in California, with features describing demographic and geographic information. We conduct the same experiment as in Figure 1b, described in details in the previous paragraph, and show the results in Figure 2. The conclusions drawn from the synthetic data experiment continue to hold here: when unlearning out-of-distribution data, existing unlearning algorithms encompassed by Algorithm 1 (with gradient descent, which is very similar to the algorithms of Neel et al. (2021) and Chourasia and Shah (2023)) are slower than Algorithm 2, which we recall is much less sensitive to out-of-distribution samples by design. It is also worth noting that Algorithm 1 is even slower as the forget data fraction increases, as predicted by the theory, and the same holds for Algorithm 2 although at a lower rate.

