# OpenReview forum: "The Utility and Complexity of In- and Out-of-Distribution Machine Unlearning"
_ICLR.cc/2025/Conference — ICLR 2025 Poster_

### Official Review · Reviewer_BBwt · 2024-11-03

**Soundness:** 3
**Presentation:** 2
**Contribution:** 3
**Rating:** 5
**Confidence:** 3

**Summary:**

This paper presents a comprehensive framework for approximating the trade-off between deletion utility and complexity, addressing two complementary deletion scenarios: when the forget set is in distribution and when the forget set is out of distribution. The author's algorithm defines that the optimal values of unlearning algorithms and retraining algorithms can be bounded, allowing for the derivation of the utility and time complexity of gradient descent algorithms based on this bound to achieve a balance between utility and complexity. However, the author does not provide a proof for this bound.

**Strengths:**

1. Exploring unlearning algorithm with theoretical guarantees is very meaningful, as the paper states that existing heuristic algorithms lack formal guarantees, rendering it unclear when they comply with regulatory standards.

2. Analyzing unlearning when the forgotten data can deviate arbitrarily from the test distribution have certain significance in some potential cases.

**Weaknesses:**

1. **Theoretically**
   - **Lack of Necessary Proof:** The author directly defines that the outputs (models) of the unlearning and retraining algorithms satisfy (Equation 12), $\left\|\theta^{\mathcal{U}}-\boldsymbol{\theta} _ {\mathcal{S} \backslash \mathcal{S} _ f}^{\star}\right\|^2 \leq \frac{\alpha \varepsilon}{4 L d},$ but lacks a crucial proof for Equation 12.

Specifically, The author defines (Equation 12), and proves that the utility objectives $\mathcal{L}\left(\mathcal{U}\left(\mathcal{S} _ f, \mathcal{A}(\mathcal{S})\right) ; \mathcal{S} \backslash \mathcal{S} _ f\right)-\mathcal{L} _ {\mathcal{S} \backslash \mathcal{S} _ f}^{\star} \leq \alpha$ (Equation 16) based on Equation 12. They then state that within $T=\mathcal{O}\left(n d \log \left(1+\frac{d}{\alpha \varepsilon}\left(\frac{R f}{n}\right)^2\right)\right)$time, it's possible to guarantee that the optimization achieves a difference in Equation 16 that is less than $\alpha$. However, just because the optimization leads to a result less than $\alpha$ in Equation 16, it doesn’t mean the unlearning model $\theta^{\mathcal{U}}$ and the minimum values of  retraining algorithms satisfy $\left\|\theta^{\mathcal{U}}-\boldsymbol{\theta} _ {\mathcal{S} \backslash \mathcal{S} _ f}^{\star}\right\|^2 \leq \frac{\alpha \varepsilon}{4 L d}$.  In other words, at time $T$, the model we found satisfies Equation 16, but it does not necessarily satisfy Equation 12.

For instance, consider the $L$-smoothness function $\mathcal{L}(\theta _ 1,\theta _ 2,\theta _ 3) = \theta _ 1^2 + \theta _ 2^2 + \frac{1}{{10}^6} \theta _ 3^2$ which has $L=2$ and dimension $d=3$, with the optimal value occurring at $(\theta _ 1,\theta _ 2,\theta _ 3)=(0,0,0)$. Let $\epsilon=2400, \alpha=0.01$. According to the author’s definition in Equation (12), it follows that $\left\|\theta^{\mathcal{U}}-\boldsymbol{\theta} _ {\mathcal{S} \backslash \mathcal{S} _ f}^{\star}\right\|^2 \leq \frac{\alpha \varepsilon}{4 L d} \leq 1$  and we can derive that $\mathcal{L}\left(\mathcal{U}\left(\mathcal{S} _ f, \mathcal{A}(\mathcal{S})\right) ; \mathcal{S} \backslash \mathcal{S} _ f\right)-\mathcal{L} _ {\mathcal{S} \backslash \mathcal{S} _ f}^{\star} \leq 0.01$. When unlearning algorithm optimizes to a loss difference of less than 0.01 in $T$ time, we still cannot guarantee $\left\|\theta^{\mathcal{U}}-\boldsymbol{\theta} _ {\mathcal{S} \backslash \mathcal{S} _ f}^{\star}\right\|^2 \leq 1$ .A counterexample illustrates this: taking $(\theta _ 1,\theta _ 2,\theta _ 3)=(0,0,100)$,  we find $\mathcal{L}(\theta _ 1,\theta _ 2,\theta _ 3)-\mathcal{L}(0,0,0) = 0.01$, but $||(\theta _ 1,\theta _ 2,\theta _ 3)-(0,0,0)||^2=(100)^2 \gg1$.  Hence, the author should firstly prove bound  $\left\|\theta^{\mathcal{U}}-\boldsymbol{\theta} _ {\mathcal{S} \backslash \mathcal{S} _ f}^{\star}\right\|^2$ similarly to [Ref 1] in order to proceed with the subsequent analysis.
   - For $\mathcal{U}\left(\mathcal{S} _ f, \mathcal{A}(\mathcal{S})\right):=\boldsymbol{\theta}^{\mathcal{U}}+\mathcal{N}\left(0, \frac{\alpha}{2 L d} \mathbf{I} _ d\right)$ and $\mathcal{U}\left(\varnothing, \mathcal{A}\left(\mathcal{S} \backslash \mathcal{S} _ f\right)\right):=\theta _ {\mathcal{S} \backslash \mathcal{S} _ f}^{\mathcal{A}}+\mathcal{N}\left(0, \frac{\alpha}{L d} \mathbf{I} _ d\right)$，please check Equation 15, $\mathrm{D} _ q\left(\mathcal{U}\left(\mathcal{S} _ f, \mathcal{A}(\mathcal{S})\right) \| \mathcal{U}\left(\varnothing, \mathcal{A}\left(\mathcal{S} \backslash \mathcal{S} _ f\right)\right)\right)=\frac{q}{2 \cdot \frac{\alpha}{2 L d}}\left\|\theta^{\mathcal{u}}-\boldsymbol{\theta} _ {\mathcal{S}}^{\mathcal{A}} \backslash \mathcal{S} _ f\right\|^2$, is consistent with the Rényi divergence expression between multivariate Gaussians given by: $D _ q\left(f _ i \| f _ j\right)= \frac{q}{2}\left(\boldsymbol{\mu} _ i-\boldsymbol{\mu} _ j\right)^{\prime}\left[\left(\Sigma _ q\right)^*\right]^{-1}\left(\boldsymbol{\mu} _ i-\boldsymbol{\mu} _ j\right)  -\frac{1}{2(q-1)} \ln \frac{\left|\left(\Sigma _ q\right)^*\right|}{\left|\Sigma _ i\right|^{1-q}\left|\Sigma _ j\right|^q},\ \  \left(\Sigma _ q\right)^*=q \Sigma _ j+(1-q) \Sigma _ i$. The paper did not provide the derivation process, and the result I obtained based on the expression above is inconsistent with the authors.
   - For a gradient method, assumptions such as a unique empirical loss minimizer and strong convexity limit its applicability.
2. **Experiments:** The paper lacks experimental evaluation, only providing a comparison of differential privacy in Figure 1. The author should compare their work with relevant unlearning research. While it's understandable that cutting-edge certified unlearning approaches, as a gradient method, it should at least be conduct experiment for existing gradient methods, such as [Ref 1]. Additionally, the author should report runtime and other metrics, such as privacy budget.
3. **Clarity：** The author should further explain in what privacy contexts data forgetting may encounter out-of-distribution issues to highlight the significance of this research. Considering that all users may submit deletion requests, this means we cannot guarantee that every piece of data to be removed from the model is out-of-distribution and may be far smaller than the in-distribution data. This uncertainty could complicate the design of forgetting mechanisms.
4. **Notation:** The author's notation and symbolic system are very confusing and unclear. The definition of $\alpha$ is confusing: in #203 it refers to worst-case deletion error, in #217 it denotes approximation error, in #234 it is used for the training procedure to approximate the empirical risk minimizer up to squared distance (which seems to conflict with the notation in #220), and in #256 it represents precision.  Please ensure a consistent definition.  Additionally, is $S _ f$ an unlearning request or the forget set as stated in #119? If $S _ f$ represents the forget set, then Equation 3 implies that your unlearning algorithm only needs to utilize the forget set, which conflicts with your algorithm. Furthermore, the paper contains many undefined symbols, such as $\theta _ S^A,\theta _ S^{A _ f}$.

**Questions:**

1. In Algorithm 1, it is required that the unlearning algorithm approximates the retraining risk minimizer on $\mathcal{S} \backslash \mathcal{S}_f$  to a squared distance of $\frac{\pi \in}{4 L \mathcal{d}}$. This requirement is unreasonable because we do not know when the unlearning algorithm approaches $\frac{\pi \in}{4 L \mathcal{d}}$. We have no way of knowing whether the output of the algorithm can even be bounded by this value in relation to the retraining risk minimizer.

2. Considering that there is already a definition of efficiency, it is not wise to extend deletion capacity to computational efficiency. Specifically, deletion capacity was initially intended to understand the relationship between the maximum number of samples that can be deleted and generalization error. The author additionally defines computational (efficiency) deletion capacity, which refers to the maximum number of samples that can be removed within a fixed time overhead. I do not recommend such a generalization, as there are already related metrics, making it a redundant definition. Otherwise, we could generalize deletion capacity to all unlearning metrics, such as efficacy deletion capacity (the maximum number of samples that can be deleted while maintaining good remaining accuracy) and fidelity deletion capacity (the maximum number of samples that can be deleted while maintaining good forget accuracy).

[Ref 1] Descent-to-Delete: Gradient-Based Methods for Machine Unlearning. Seth Neel, Aaron Roth, Saeed Sharifi-Malvajerdi. Proceedings of the 32nd International Conference on Algorithmic Learning Theory, PMLR, 2021.

---

> ### Author Response · Authors · 2024-11-16
> **Rebuttal to Reviewer BBwt (Part 1)**
>
> We thank the reviewer for their thoughtful feedback, and we address their comments below.
>
> >The author directly defines that the outputs (models) of the unlearning and retraining algorithms satisfy (Equation 12), but lacks a crucial proof for Equation 12...
>
> We appreciate the reviewer’s comment and provide clarification below. Theorem 1 analyzes the general framework of (or the meta-algorithm) Algorithm 1, allowing us to abstract away from specific optimization algorithms by assuming that both training and unlearning procedures return approximate empirical minimizers with predefined bounds. This setup, therefore, enables us to use these bounds directly in our proof without further derivation.
>
>
> Subsequently, in Corollary 2, we specialize the result of Theorem 1 to using gradient descent as an optimizer. This is very similar to the algorithm analyzed by Neel et al. (2021) as we explain in Section 4, although the authors did not consider population risk as we do via Proposition 1, and focused on constrained optimization.
> It is in the proof of Corollary 2 (lines 911-913) that we actually use the fact that gradient descent guarantees the approximation error on the empirical risk minimizers (during training and unlearning) within the stated time budget following standard convergence analyses, e.g., Theorem 2.1.15 in Nesterov et al. (2018), which is what the reviewer referred to as proving Equation (12).
>
>
>
> >[...] please check Equation 15 [...] is consistent with the Rényi divergence expression between multivariate Gaussians...
>
> We thank the reviewer for bringing up this point, and will clarify these steps of the proof in the paper as follows.
> The formula we use for the Rényi divergence between multivariate Gaussians (lines 860-863) is indeed a special case of the one from (Gil et al. 2013) and recalled by the reviewer.
> To see this, one should notice that both Gaussians at hand (line 860) have the same covariance matrix, in which case we recall that the formula of Rényi divergences of order $q$ for Gaussians $\mathcal{N}(\mathbf{\mu}, \mathbf{\Sigma}), \mathcal{N}(\mu', \mathbf{\Sigma})$ for arbitrary vectors $\mathbf{\mu}, \mathbf{\mu'} \in \mathbb{R}^d$ and symmetric positive definite matrix $\mathbf{\Sigma} \in \mathbb{R}^{d \times d}$  simplifies to $\frac{q}{2} (\mathbf{\mu} - \mathbf{\mu}')^\top \mathbf{\Sigma}^{-1} (\mathbf{\mu} - \mathbf{\mu}')$.
> Substituting the centers and covariance of these Gaussians with those in the proof of Theorem 1 (line 860) directly yields the equality step in Equation (15).
>
>
> >For a gradient method, assumptions such as a unique empirical loss minimizer and strong convexity limit its applicability.
>
> We acknowledge that assumptions such as strong convexity, which are standard for theoretical works like ours, can limit applicability. Yet, we would like to recall that Theorem 1 does not require convexity, and could in fact be applied to certain non-convex tasks for which approximate empirical risk minimizers exist, such as principal component analysis and matrix completion (see lines 250-252).
>
> >The paper lacks experimental evaluation, only providing a comparison of differential privacy in Figure 1...
>
> We acknowledge that our main contribution is theoretical, and that the objective of our experiments is only to verify the theoretical insights we develop.
> These are (i) the separation of unlearning with differential privacy (Figure 1.a) and (ii) the improved complexity cost of out-of-distribution unlearning with Algorithm 2 (Figure 1.b).
> We also remark that Algorithm 1 with Gradient Descent used as optimizer (implemented in Figure 1) is very similar to the algorithm analyzed in the reference pointed to by the reviewer (Neel et al., 2021), with the minor difference that we do not project models.
> Finally, we would like to recall that related algorithms to our analysis have already been empirically validated in practical unlearning scenarios, e.g., (Guo et al. 2020). We will clarify this in the paper.
>
> >The author should further explain in what privacy contexts data forgetting may encounter out-of-distribution issues to highlight the significance of this research.
>
> We thank the reviewer for bringing up this question, and will clarify this point in the paper as follows.
> The out-of-distribution scenario can arise whenever the forget data deviates from the true distribution of the remaining data, e.g., when the data is collected from heterogeneous sources; this is in fact natural for unlearning settings, since data points are owned by different individuals who can request erasure.
> The extreme out-of-distribution scenario, also supported in theory, corresponds to some of the forget data being corrupt,  has been shown empirically to cause several certified unlearning procedures to be slow (Marchant et al., 2022), which is exactly what our theory prevents; the unlearning time complexity in Theorem 3 is independent of the forget data.

---

> ### Author Response · Authors · 2024-11-16
> **Rebuttal to Reviewer BBwt (Part 2)**
>
> >The author's notation and symbolic system are very confusing and unclear...
>
> We appreciate the feedback on notation. In the revised manuscript, we will ensure consistent use of terminology by adopting unified terms for quantities like $\alpha$ (error) and $\mathcal{S}_f$ (forget set).
>
>
>
> >In Algorithm 1, it is required that the unlearning algorithm approximates the retraining risk minimizer[...] This requirement is unreasonable...
>
> We understand the concern regarding Algorithm 1’s precision requirements.
> There are two practical scenarios where we can compute an upper bound on the number of optimization iterations needed to reach a certain precision (in terms of squared distance to the empirical risk minimizer). Consider the optimizer to be gradient descent here for clarity, and denote the empirical loss $\mathcal{L} \colon \mathbb{R}^d \to \mathbb{R}$, and the corresponding empirical risk minimizer $\mathbf{\theta}_\star \in \mathbb{R}^d$.
>
> The first scenario is when the loss function is non-negative (or some global lower bound is known); this is quite common in machine learning, e.g., quadratic loss, cross-entropy loss, hinge loss, etc...
> In this case, we know that for any initial model $\theta_0 \in \mathbb{R}^d$, we have
>
> $\|\|\theta_0 - \theta_\star\|\|^2 \leq \frac{2}{\mu}(\mathcal{L}(\theta_0) - \mathcal{L}(\theta_\star)) \leq \frac{2}{\mu}\mathcal{L}(\mathbf{\theta}_0),$
>
> where the first inequality is due to strong convexity, and the second to the loss being non-negative.
> Therefore, knowing only the loss at the initial model, and (a lower bound on) the strong convexity parameter, e.g., $\ell_2$-regularization factor, we have a computable upper bound on the initialization error.
> The upper bound on the number of iterations follows directly from standard first-order convergence analyses, e.g., see Theorem 2.1.15 of Nesterov et al. (2018), and is computable knowing the aforementioned bound on the initialization error, and the smoothness and strong convexity constants.
>
> The second scenario is when the parameter space is bounded, and in which case we use the projected variant of gradient descent, analyzed in (Neel et al., 2021) for empirical risk minimization.
> There, we know that the initialization error is bounded by the diameter of the parameter space, which is computable.
> A computable upper bound on the number of iterations needed follows with a similar argument as the first scenario above.
>
>
> >Considering that there is already a definition of efficiency, it is not wise to extend deletion capacity to computational efficiency...
>
> The main reason why we extended the formalism of Sekhari et al. (2021) to computational deletion capacity (Definition 3), is to be able to formally compare unlearning-training pairs in terms of complexity; we only view it as a tool and not a contribution in itself.
> We believe computational deletion capacity to be quite natural; we are interested in the trade-off between utility, complexity and unlearning, and the main notions quantifying each for our setting are, respectively, population risk, time complexity, and bound on model indistinguishability.

---

> > ### Comment · Reviewer_BBwt · 2024-11-22
> >
> > - The authors did not address the core of my concern. Equation (12) is used to derive Equation (16) via the triangle inequality, which suggests that the reasoning leading to Equation (16) cannot be reversed to apply back to Equation (12). The authors should check this to determine whether there is a circular argument involved, as this affects the conclusions of the entire paper. In other words, the iterations of the unlearning algorithm should be derived based on Equation (12), not Equation (16). In my opinion, the stopping (or convergence) time $T$ for this SGD-based unlearning algorithm should be derived from Equation (12) and Theorem 2.1.15 in Nesterov et al. (2018), rather than relying on Equation (16). As an example to demonstrate the proof of $T$, shouldn't the proof be based on $||\theta^\mathcal{U} - \boldsymbol{\theta} _ {S \backslash S _ f}^{\star}||^2 \leq (1 - \frac{2h \mu L}{\mu + L})^T ||\theta_ {S }^{\star} - \boldsymbol{\theta} _ {S \backslash s _ f}^{\star}||^2$ and $\left|\theta^{\mathcal{U}} - \boldsymbol{\theta} _ {S \backslash s _ f}^{\star}\right|^2 \leq \frac{\alpha \varepsilon}{4 L d}$? That is, by setting $ (1 - \frac{2h \mu L}{\mu + L})^T |\theta_ {S }^{\star} - \boldsymbol{\theta} _ {S \backslash S _ f}^{\star}|^2 \leq \frac{\alpha \varepsilon}{4 L d}$, the stopping time $T$ of the unlearning algorithm based on SGD algorithm in Theorem 2.1.15 in Nesterov et al. (2018). Otherwise, the counterexample I provided remains a valid concern.
> >
> > - This leads to a subsequent issue: the result for the convergence time requires knowledge of the unlearning algorithm, such as SGD (Theorem 2.1.15 in Nesterov et al., 2018). To obtain the convergence analysis of these algorithms, certain assumptions (e.g., strong convexity, smoothness) must be made. This means that the conclusion of Theorem 1 needs to specify the particular algorithm (e.g., SGD) for it to be valid. Otherwise, stating that Theorem 1 does not require non-convexity lacks clarity, unless the author can prove that $||\theta _ {S }^{\star} - \boldsymbol{\theta} _ {S \backslash s _ f}^{\star}||$  holds without any assumptions of strong convexity or smoothness.
> >
> > - I am also confused because the authors include convexity and smoothness in Line 912, but omit them in Line 920 in Corollary 2. The authors should avoid misleading descriptions and clearly state the assumptions required for each theorem or corollary.
> >
> > - The two Gaussian distributions, as presented incorrectly in Lines 859-860, do not share the same covariance matrix, contrary to the response's claim. This mistake should be corrected to avoid potential misleading conclusions.
> >
> > The authors' response did not address my concerns. The paper still has the following flaws, making it unable to be accepted: (1) a proof error that undermines the result of the entire paper, (2) weak experimental support, and (3) poor presentation quality, especially regarding the notation. Given these issues, I  do not find it appropriate to raise my score.

---

> ### Author Response · Authors · 2024-11-23
> **Response to Reviewer BBwt (Part 1)**
>
> We thank the reviewer for engaging with our response. We address their remaining concerns below.
>
> ## **Clarifications on Theorem 1**
>
> There is a misunderstanding regarding the statement of Theorem 1.
> We first respond to the latest comments of the reviewer regarding Theorem 1, since it is the major concern, before clarifying Theorem 1 and its proof right after.
>
>
> > Reviewer: Equation (12) is used to derive Equation (16) via the triangle inequality, which suggests that the reasoning leading to Equation (16) cannot be reversed to apply back to Equation (12). The authors should check this to determine whether there is a circular argument involved, as this affects the conclusions of the entire paper.
>
> **The inequalities established in Equation (12) in the paper (see our sketch of proof below)  follow directly from the design of the algorithm.** There is no circular argument involved, since the remaining parts of the proof (including Equation (16)) simply follow from the aforementioned inequalities.
>
> > Reviewer: the result for the convergence time requires knowledge of the unlearning algorithm, such as SGD (Theorem 2.1.15 in Nesterov et al., 2018). To obtain the convergence analysis of these algorithms, certain assumptions (e.g., strong convexity, smoothness) must be made. This means that the conclusion of Theorem 1 needs to specify the particular algorithm (e.g., SGD) for it to be valid.
> Otherwise, stating that Theorem 1 does not require non-convexity lacks clarity, unless the author can prove that
> $|| \boldsymbol{\theta}^\star_{\mathcal{S}} - \boldsymbol{\theta}^\star_{\mathcal{S} \setminus \mathcal{S}_f} ||^2$ holds without any assumptions of strong convexity or smoothness.
>
> **Theorem 1 intentionally abstracts away from any specific training or unlearning optimizer.** We only require access to an approximate global risk minimizer, which allows flexibility in the choice of optimizer. This framework does not necessitate strong convexity. The latter is only imposed in Corollary 2, where we analyze the behavior of a specific optimization algorithm (Gradient Descent for training and unlearning). We will ensure that this distinction is more explicitly stated in the revised paper to avoid any potential confusion.
>
>
> > Reviewer: I am also confused because the authors include convexity and smoothness in Line 912, but omit them in Line 920 in Corollary 2. The authors should avoid misleading descriptions and clearly state the assumptions required for each theorem or corollary.
>
> We assume that the reviewer refers to *ignoring dependencies* on the strong convexity and smoothness constants $\mu, L$ in the time complexities in the proof of Corollary 2 (line 920).
> It is clearly stated in Corollary 2 that we ignore such dependencies (line 265).
> Regarding the assumptions used in the proof, we clearly restate these in the beginning of the proof (line 907).
>
> > Reviewer: The two Gaussian distributions, as presented incorrectly in Lines 859-860, do not share the same covariance matrix, contrary to the response's claim. This mistake should be corrected to avoid potential misleading conclusions.
>
> Thanks for pointing out the typo in the second covariance in line 859, which we will correct. There is a constant 2 missing in the denominator of the multiplicative factor of the covariance. The covariance should clearly be the same for both Gaussians in line 859, since Algorithm 1 always injects the same noise magnitude regardless of the forget data.
>
> We now clarify Theorem 1, starting with notation below.
> ### Setup and Notation
> For any dataset $\mathcal{S}$, we denote by $\boldsymbol{\theta}^\star_{\mathcal{S}}$ the global minimizer of the empirical loss $\mathcal{L}(\cdot\~; \mathcal{S})$, which we assume to be unique.
> In particular, for any forget set $\mathcal{S_f} \subset \mathcal{S}$, we denote by $\boldsymbol{\theta}^\star_{\mathcal{S} \setminus \mathcal{S}_f }$ the unique global minimizer of the empirical loss $\mathcal{L}(\cdot~; \mathcal{S} \setminus \mathcal{S}_f )$.
>
> Consider an arbitrary optimization procedure $\mathcal{A}$ (for training or unlearning) which outputs $\boldsymbol{\theta_{\mathcal{S}}}^{\mathcal{A}} \in \mathbb{R}^d$ when given dataset $\mathcal{S}$ and initial model $\boldsymbol{\theta_0} \in \mathbb{R}^d$.
> For every $\alpha_{\mathrm{precision}}, \Delta_{\mathrm{initial}}>0$, we denote by $T_{\mathcal{A}}(\alpha_{\mathrm{precision}}, \Delta_{\mathrm{initial}})$ the computational complexity required by $\mathcal{A}$ to guarantee
> *on any dataset $\mathcal{S}$* that, given the initialization error
> $||\boldsymbol{\theta_0} - \boldsymbol{\theta_{\mathcal{S}}}^{\star}||^2 \leq \Delta_{\mathrm{initial}}$, its output
> $\boldsymbol{\theta_{\mathcal{S}}}^{\mathcal{A}}$ satisfies
> $$|| \boldsymbol{\theta_{\mathcal{S}}}^{\mathcal{A}} - \boldsymbol{\theta_{\mathcal{S}}}^\star ||^2 \leq \alpha_{\mathrm{precision}}.$$
> We emphasize that the latter should hold for any dataset, including any subset of the full training set.

---

> ### Author Response · Authors · 2024-11-23
> **Response to Reviewer BBwt (Part 2)**
>
> Given this notation, we now restate Theorem 1 for clarity.
>
> **Theorem 1.**
>         Let $\varepsilon, \alpha, \Delta>0, 0 \leq f < n$, and $q>1$.
>     Assume that, for every $\mathbf{z} \in \mathcal{S}$, the loss $\ell(\cdot\~;\mathbf{z})$ is $L$-smooth, and $\varepsilon \leq d$.
>     Assume that for every $\mathcal{S_f} \subset \mathcal{S}, |{\mathcal{S_f}}| \leq f,$ the empirical loss over $\mathcal{S} \setminus \mathcal{S_f}$ has a unique minimizer.
>     Recall the notation above and consider the unlearning-training pair $(\mathcal{U}, \mathcal{A})$ in Algorithm 1, with the initialization error of $\mathcal{A}$ on set $\mathcal{S}$ being at most $\Delta$.
>
> Then, $(\mathcal{U}, \mathcal{A})$ satisfies *$(q, q\varepsilon)$-approximate unlearning* with empirical loss, over worst-case $\mathcal{S} \setminus \mathcal{S_f}$, at most $\alpha$ in expectation over the randomness of the algorithm, with time complexity:
>
> - $\text{Training:}\~\~ T_\mathcal{A}{\left(\frac{\alpha\varepsilon}{4Ld}, \Delta\right)}$
>
> - $\text{Unlearning:}\~\~ T_\mathcal{U}{\bigg(\frac{\alpha\varepsilon}{2Ld},\~ \frac{\alpha\varepsilon}{4Ld} + 2 \max_{\substack{\mathcal{S_f} \subset \mathcal{S}: \\ |{\mathcal{S_f}}| \leq f}} ||{\boldsymbol{\theta_{\mathcal{S}}}^\star - \boldsymbol{\theta}_{\mathcal{S} \setminus \mathcal{S_f}}^\star}||^2\bigg)}.$
>
> **Sketch of proof.**
> The proof is decomposed in three parts: (i) proof of the time complexity bounds and preliminary error bounds (ii) proof of the approximate unlearning claim, (iii) proof of the utility claim (empirical loss at most $\alpha$).
>
>
> First, by design of Algorithm 1, we know that the output $\boldsymbol{\theta_{\mathcal{S}}}^{\mathcal{A}}$ of the training procedure satisfies
> $$|| \boldsymbol{\theta_{\mathcal{S}}}^{\mathcal{A}} - \boldsymbol{\theta_{\mathcal{S}}}^\star ||^2 \leq \frac{\alpha \varepsilon}{4Ld}.$$
> Also, since we assume the initialization error during training to be at most $\Delta$,
> the time complexity of training is at most $T_\mathcal{A}(\frac{\alpha \varepsilon}{4Ld}, \Delta)$.
>
> Similarly, by design of Algorithm 1, we know that the output $\boldsymbol{\theta_{\mathcal{S} \setminus \mathcal{S_f}}}^{\mathcal{A}}$ of the training procedure when run on the retain data, i.e., retraining from scratch, satisfies
> $$|| \boldsymbol{\theta_{\mathcal{S} \setminus \mathcal{S_f}}}^{\mathcal{A}} - \boldsymbol{\theta_{\mathcal{S} \setminus \mathcal{S_f}}}^\star ||^2 \leq \frac{\alpha \varepsilon}{4Ld}.$$
> Here, we do not take into account the computational complexity required since we do not actually run the algorithm on the retain set, i.e., this is simply retraining from scratch.
>
>
> For unlearning, by design of Algorithm 1, we know that the output $\boldsymbol{\theta}^{\mathcal{U}}$ of the unlearning procedure satisfies
> $$|| \boldsymbol{\theta}^{\mathcal{U}} - \boldsymbol{\theta_{\mathcal{S} \setminus \mathcal{S_f}}}^\star ||^2 \leq \frac{\alpha \varepsilon}{4Ld}.$$
> Now, since the unlearning procedure is initialized at $\boldsymbol{\theta_{\mathcal{S}}}^{\mathcal{A}}$ the output of the training procedure, the initialization error is $\Delta(\mathcal{S_f}) := || \boldsymbol{\theta_{\mathcal{S}}}^{\mathcal{A}} - \boldsymbol{\theta_{\mathcal{S} \setminus \mathcal{S_f}}}^\star ||^2$.
> Also, since we want the last inequality to hold for any choice of forget set $\mathcal{S_f} \subset \mathcal{S}$,
> the time complexity required is $T_\mathcal{U}(\frac{\alpha \varepsilon}{4Ld}, \max_{\substack{\mathcal{S_f} \subset \mathcal{S}:\\ |\mathcal{S_f}| \leq f}}\Delta(\mathcal{S_f}))$.
> In fact, we can upper bound $\Delta(\mathcal{S_f})$ using the two first inequalities to get inequality (13) in the paper (line 846), and subsequently we establish that the computational complexity required to obtain
> $$\max_{\substack{\mathcal{S_f} \subset \mathcal{S}\\ |\mathcal{S_f}| \leq f}} || \boldsymbol{\theta}^{\mathcal{U}} - \boldsymbol{\theta_{\mathcal{S} \setminus \mathcal{S_f}}}^\star ||^2 \leq \frac{\alpha \varepsilon}{4Ld},$$
> is at most $T_\mathcal{U}(\tfrac{\alpha\varepsilon}{4Ld}, \tfrac{\alpha \varepsilon}{2Ld} + 2 \max_{\substack{\mathcal{S_f} \subset \mathcal{S} \\ |{\mathcal{S_f}}| \leq f}} ||{\boldsymbol{\theta_{\mathcal{S}}}^\star - \boldsymbol{\theta_{\mathcal{S} \setminus \mathcal{S_f}}}^\star}||^2)$.
> This concludes the first part of the proof. We continue the sketch of proof in the next window.

---

> ### Author Response · Authors · 2024-11-23
> **Response to Reviewer BBwt (Part 3)**
>
> The second part of the proof (lines 852-863) shows the approximate unlearning claim.
> The main step is to bound the distance between the output $\boldsymbol{\theta}^\mathcal{U}$ of the unlearning phase, and the output $\boldsymbol{\theta_{\mathcal{S} \setminus \mathcal{S_f}}}^\mathcal{A}$ of the training procedure on the retain data (retraining from scratch) directly using the second and third inequalities in this sketch of proof to obtain:
> $$  ||{\boldsymbol{\theta}^\mathcal{U} - \boldsymbol{\theta_{\mathcal{S} \setminus \mathcal{S_f}}}^\mathcal{A}}||^2 \leq \frac{\alpha \varepsilon}{Ld}.$$
> Finally, since Algorithm 1 outputs $\boldsymbol{\theta}^\mathcal{U}$, or $\boldsymbol{\theta}_{\mathcal{S} \setminus \mathcal{S}_f}^\mathcal{A}$ when retraining from scratch (i.e., training on the retain data without unlearning requests), with additive Gaussian noise $\mathcal{N}(0, \frac{\alpha}{2Ld} \mathbf{I}_d)$, a standard Rényi divergence bound between two Gaussians (with the same covariance) establishes the approximate unlearning claim.
> This concludes the second part of the proof.
>
> The third and final part of the proof (lines 864-888) shows that the expected empirical loss (over the retain set) is at most $\alpha$ (inequality (16) in the paper).
> To show this, we leverage the smoothness of the loss function, and directly use the fourth inequality in this sketch of proof, which bounds the distance between $\boldsymbol{\theta}^\mathcal{U}$ and $\boldsymbol{\theta}^\star_{\mathcal{S} \setminus \mathcal{S}_f}$ for any choice of forget set.
> This concludes the proof.
>
> ## **Other Concerns**
>
> - **Experimental Validation.**
> Our main contributions are theoretical: (i) we show a tight unlearning-utility-complexity trade-off, (ii) we initiate the rigorous study of out-of-distribution unlearning and propose a new algorithm with strong guarantees there.
> *Yet, we have conducted experiments to illustrate key insights from our theoretical analysis:* (i) we show the separation with differential privacy in the in-distribution case, and (ii) the improvement of Algorithm 2 over Algorithm 1, a standard gradient-based unlearning algorithm (very similar to (Neel et al. 2021)), in the out-of-distribution case.
>
> - **Notation.**
> We have taken into account the reviewer's remarks on how to refer to $\alpha$ the bound on the empirical risk, and $\mathcal{S}_f$ the forget set consistently. We will revise the occurrences pointed out by the reviewer accordingly.

---

### Official Review · Reviewer_FXKP · 2024-11-03

**Soundness:** 4
**Presentation:** 4
**Contribution:** 4
**Rating:** 8
**Confidence:** 3

**Summary:**

This paper provides a theoretical analysis of the utility and complexity trade-offs in approximate machine unlearning, with in-distribution and out-of-distribution cases. For the former one, the authors uncover the dimension independency with the help of an optimisation procedure with output perturbation. For the latter one, which is more challenging and under explored, the authors propose a robust gradient descent variant with satisfying time and space complexity, and corresponding statistical guarantee.

**Strengths:**

The paper is very well-written. The problem addressed in the paper is important and interesting, with solid theoretical insights on both in-distribution and out-of-distribution cases. The discussion is comprehensive and the comparisons with the differential privacy and the illustrations are insightful to me.

**Weaknesses:**

None.

**Questions:**

None.

---

> ### Author Response · Authors · 2024-11-16
> **Rebuttal to Reviewer FXKP**
>
> We appreciate the positive feedback of the reviewer on our contributions!

---

> > ### Author Response · Authors · 2024-11-26
> > **Revision Update**
> >
> > Thank you for your feedback. We have further refined the manuscript, improving clarity and adding experiments to strengthen our contributions. We hope these updates effectively address your concerns and highlight the significance of our work.
> > We would greatly appreciate it if you could consider revisiting your score, noting that at this conference, an "8" corresponds to "accept."

---

### Official Review · Reviewer_Lmpv · 2024-11-03

**Soundness:** 4
**Presentation:** 3
**Contribution:** 3
**Rating:** 6
**Confidence:** 3

**Summary:**

This paper studies the problem of machine unlearning, the removal of certain data points from a model after unlearning, for both in and out of distribution data, with a focus on the deletion capacity of unlearning algorithms. First, they present a general theorem for analyzing the utility deletion capacity of an unlearning algorithm, the number of deletions an unlearning algorithm can tolerate while maintaining some error rate. Next, they present an algorithm (Algorithm 1) for unlearning in-distribution data using a noisy ERM approximation and analyze the time and space complexity of their algorithm. They demonstrate that the time complexity of Algorithm 1 can become unbounded for even a single out-of-distribution data removal, and propose a new algorithm for unlearning out-of-distribution data using gradient descent with coordinate-wise trimmed mean gradients.

**Strengths:**

The theoretical study of out-of-distribution deletions in unlearning is valuable and lacking in the unlearning literature.

Proposition 1 gives a generic framework for assessing the very important tradeoff between utility and deletion capacity which I think could be very informative to the community.

**Weaknesses:**

I think this paper would be improved by a comparison of the utility and computational deletion capacity between the Noisy-GD algorithm from Chourasia and Shah [2023]. The algorithms are very similar and Chourasia and Shah [2023] also have a detailed discussion of the tradeoffs between utility, deletion capacity, and computation time of their algorithm.

For the unlearning algorithms in this paper, the learning and unlearning time are both $\tilde{O}(nd)$. Although some time is saved during unlearning, the goal of unlearning is to unlearn using significantly less time than retraining from scratch.

**Questions:**

For Table 1, wouldn’t it be more fair to just compare unlearning times, rather than the sum of both learning and unlearning times? Especially since the goal of unlearning is primarily to process deletion requests quickly at the time of unlearning, and one would have to unlearn many more times than one would have to learn. For example, the unlearning time of Sekhari et al [2021] is independent of n, even though the learning time depends on n, but this distinction isn’t clear in Table 1. Perhaps separate columns for learning and unlearning times would make the comparison more clear

Suggestions

In line 203, I would suggest using a different variable (other than alpha), such as epsilon, because alpha has been used to represent the bound on the generalization error, but in line 203, it is being used to represent a bound on the empirical error.

---

> ### Author Response · Authors · 2024-11-16
> **Rebuttal to Reviewer Lmpv**
>
> We thank the reviewer for their thoughtful feedback, and we address their comments below.
>
> >I think this paper would be improved by a comparison of the utility and computational deletion capacity between the Noisy-GD algorithm from Chourasia and Shah [2023]. The algorithms are very similar and Chourasia and Shah [2023] also have a detailed discussion of the tradeoffs between utility, deletion capacity, and computation time of their algorithm.
>
>
> We thank the reviewer for bringing up this point, and we will add a comparison with the analysis of Chourasia and Shah (2023) as the reviewers suggests.
> As a remark, we recall that Chourasia and Shah (2023) focused on empirical risk minimization instead of generalization, that is why we had not included a formal comparison initially.
>
> >For the unlearning algorithms in this paper, the learning and unlearning time are both $\widetilde{\mathcal{O}}{(nd)}$. Although some time is saved during unlearning, the goal of unlearning is to unlearn using significantly less time than retraining from scratch.
>
> We acknowledge that both learning and unlearning time are $\widetilde{\mathcal{O}}{(nd)}$ in our work. However, there is an improvement compared to retraining from scratch, since its unlearning time complexity grows with the initialization error (which can be large since the initial model is arbitrary for retraining from scratch), which is not the case for the algorithm we analyze whose unlearning time complexity grows only with the fraction of the forget data (line 269). Moreover, for non-strongly convex tasks, the time complexity gap can be super-logarithmic, since the dependence on the initialization error can be polynomial following standard convergence analyses (Nesterov et al., 2018).
> We will clarify this in the paper.
>
> >For Table 1, wouldn’t it be more fair to just compare unlearning times, rather than the sum of both learning and unlearning times? Especially since the goal of unlearning is primarily to process deletion requests quickly at the time of unlearning, and one would have to unlearn many more times than one would have to learn. For example, the unlearning time of Sekhari et al [2021] is independent of n, even though the learning time depends on n, but this distinction isn’t clear in Table 1. Perhaps separate columns for learning and unlearning times would make the comparison more clear
>
>
> While comparing unlearning time only could indeed be interesting for certain use cases, we believe that comparing the sum of unlearning and training times to be fair in the general case.
> For the sake of the argument, a hypothetical (inefficient) training procedure that computes and stores the empirical risk minimizers on every subset of the full dataset would only require constant-time unlearning, which could be misleading if we only compare unlearning times.
> However, we acknowledge that the unlearning time complexity of the work of Sekhari et al. (2021) being independent of $n$, although at least quadratic in $d$, is worth highlighting, which we will do in the paper.

---

> > ### Author Response · Authors · 2024-11-26
> > **Revision Update**
> >
> > Thank you for your feedback. We have further refined the manuscript, improving clarity and adding experiments to strengthen our contributions. We hope these updates effectively address your concerns and highlight the significance of our work.
> > We would greatly appreciate it if you could consider revisiting your score, noting that at this conference, an "8" corresponds to "accept."

---

### Official Review · Reviewer_Nr4d · 2024-11-04

**Soundness:** 3
**Presentation:** 2
**Contribution:** 3
**Rating:** 6
**Confidence:** 4

**Summary:**

The paper presents a theoretical analysis of the utility and complexity trade-offs in approximate machine unlearning with the deletion capacity under fixed error bound and computational budgets. The authors consider both in-distribution and out-of-distribution unlearning scenarios and propose a new robust and efficient variant of the gradient descent method for the latter situation.

**Strengths:**

1. The study of the out-of-distribution scenario is very meaningful.
2. The insight of deletion capacity under fixed computational budgets while maintaining utility is novel.
3. The theoretical analysis is rigorous.
4. The paper is well-organized.

**Weaknesses:**

1. Adding a detailed explanation of out-of-distribution in Abstract would be better.
2. The reviewer is quite confused about Definition 2.
   - $\mathcal Z^*$ is a space (stated on line 127) rather than "the set of training sets". Please clarify the definition and usage of $\mathcal Z^*$
   - Does $\mathcal L*$ represent the minimum population risk on the training set or retain set? Simply defining $L*:=\min\mathcal L(\theta)$ (without the dataset) is not clear. Please specify over which dataset $\mathcal L^*$ is defined
   - Should $\mathcal L_{\mathrm {ID}}$ and $\mathcal L_{\mathrm{OOD}}$ be bounded by a certain value?
3. The authors should add descriptions of gradient descent in Algorithms 1 and 2.
4. Lack of explanation for the initialization error $\Delta$.
5. Although this work focuses more on the theoretical part, it is still better to provide some preliminary results on some datasets under practical scenarios.
6. Some typos, e.g., missing bold for $\theta$ in Table 1.

**Questions:**

What is "procedure 2" in Algorithm 2? Please either define "procedure 2" explicitly in Algorithm 2, or remove this reference if it's not needed.

---

> ### Author Response · Authors · 2024-11-16
> **Rebuttal to Reviewer Nr4d**
>
> We thank the reviewer for their thoughtful feedback, and we address their comments below.
>
> >Adding a detailed explanation of out-of-distribution in Abstract would be better.
>
> We thank the reviewer for bringing this up, and we will add a detailed explanation of out-of-distribution unlearning in the abstract
>
> >The reviewer is quite confused about Definition 2...
>
> We will clarify Definition 2 in the paper as follows.
> First, we recall that $\mathcal{L}_\star$ denotes the minimum value of $\mathcal{L}$, which had been defined in Equation (1) as the population risk.
> Second, it is implicitly required that the in- and out-of-distribution losses are lower bounded so that the minimum values of these functions are well-defined.
> Finally, by definition of $\mathcal{Z}^*$, any $\mathcal{S}$ is in $\mathcal{Z}^*$ if and only if there exists an integer $k \geq 0$ such that $\mathcal{S} \in \mathcal{Z}^k$ where we recall that $\mathcal{Z}$ is the data space (line 110), i.e., $\mathcal{Z}^*$ is the set of tuples on the data space $\mathcal{Z}$.
> We will clarify all these points in the paper.
>
>
> >The authors should add descriptions of gradient descent in Algorithms 1 and 2.
>
> We will recall gradient descent in the paper, since it is indeed considered in Corollary 2.
> Note that Algorithm 1 does not necessarily use gradient descent, it is a meta-algorithm where any optimizer can be used, and the same holds for the unlearning phase of Algorithm 2.
> We will explain this in the paper.
>
> >Lack of explanation for the initialization error .
>
>
> The initialization error refers to the squared Euclidean distance between the initial model and the empirical risk minimizer (line 911-912) as is standard in convex optimization. We will clarify this in the paper.
>
> >Although this work focuses more on the theoretical part, it is still better to provide some preliminary results on some datasets under practical scenarios.
>
> We acknowledge that our main contribution is theoretical, and that the objective of our experiments is only to verify the theoretical insights we develop.
> These are (i) the separation of unlearning with differential privacy (Figure 1.a) and (ii) the improved complexity cost of out-of-distribution unlearning with Algorithm 2 (Figure 1.b).
> We would also like to recall that related algorithms to our analysis have already been empirically validated in practical unlearning scenarios, e.g., (Guo et al. 2020).
>
> >What is "procedure 2" in Algorithm 2? Please either define "procedure 2" explicitly in Algorithm 2, or remove this reference if it's not needed.
>
> We will remove this reference for clarity, as suggested by the reviewer.

---

> > ### Comment · Reviewer_Nr4d · 2024-11-26
> >
> > During the discussion phase, the authors are allowed to revise their submissions to address concerns that arise. Please upload the latest revised version according to your responses (including the responses to other reviewers).

---

> > > ### Author Response · Authors · 2024-11-26
> > > **Revision Update**
> > >
> > > Thank you for your feedback. We have further refined the manuscript, improving clarity and adding experiments to strengthen our contributions. We hope these updates effectively address your concerns and highlight the significance of our work.
> > > We would greatly appreciate it if you could consider revisiting your score, noting that at this conference, an "8" corresponds to "accept."

---

### Official Review · Reviewer_RJfS · 2024-11-12

**Soundness:** 3
**Presentation:** 4
**Contribution:** 3
**Rating:** 8
**Confidence:** 3

**Summary:**

This work analyses the utility and complexity trade-offs in approximate machine unlearning, focusing on both in-distribution and out-of-distribution scenarios. The authors address the challenge of efficiently removing specific data from trained models. For in-distribution data, they demonstrate that ERM with output perturbation can achieve tight trade-offs between unlearning accuracy and computational complexity, connecting a theoretical gap regarding differential privacy limitations in unlearning. However, they identify that these techniques do not work with out-of-distribution data, where the unlearning process can become less efficient than retraining from scratch. To overcome this, the paper introduces a robust and noisy gradient descent algorithm that maintains utility while improving unlearning efficiency in out-of-distribution contexts.

**Strengths:**

* The paper provides an in-depth theoretical exploration of the utility and complexity trade-offs in approximate machine unlearning, which I liked and found beneficial.
* The authors tackle unlearning in both in-distribution and out-of-distribution (OOD) contexts and make an interesting separation between the scenarios
* A notable contribution is the introduction of a robust and noisy gradient descent algorithm (Algorithm 2) for OOD unlearning.
* The paper addresses and resolves a theoretical question Sekhari et al. (2021) posed regarding dimension-independent utility deletion capacity.
* Overall, it is well-written and straightforward to follow.

**Weaknesses:**

* Strong Assumptions Limit Applicability: The theoretical results rely on assumptions such as strong convexity, smoothness of the loss function, and the existence of a unique global minimum. While these are standard assumptions for theoretical work, their applicability in practice needs to be clarified.
* Limited Experimental Evaluation: The experimental validation is conducted on relatively simple tasks, such as linear regression with synthetic data.
* Assumption of Known Forget Set Size: The algorithms and theoretical guarantees often assume that the size of the forget set is known and relatively small compared to the total dataset size.

**Questions:**

* Am I correct in stating that unlearning is not generally possible for non-convex models without a unique minimiser? Can the authors comment on relaxing their assumption to general non-convex models?
* Could access to a global minimiser be relaxed with a weaker assumption, like the optima within a bounded set of known diameters?

---

> ### Author Response · Authors · 2024-11-16
> **Rebuttal to Reviewer RJfs**
>
> We thank the reviewer for their thoughtful feedback, and we address their comments below.
>
>
> >Strong Assumptions Limit Applicability: The theoretical results rely on assumptions such as strong convexity, smoothness of the loss function, and the existence of a unique global minimum. While these are standard assumptions for theoretical work, their applicability in practice needs to be clarified.
>
> We acknowledge that assumptions such as strong convexity are standard for theoretical works like ours but can limit applicability.
> Yet, we would like to recall that Theorem 1 does not require convexity, and could in fact be applied to certain non-convex tasks for which approximate empirical risk minimizers exist, such as principal component analysis and matrix completion (see lines 250-252).
> Besides, the strongly convex case itself is of practical interest, e.g., for unlearning the last layer of a deep classifier pre-trained on public data (Guo et al. 2020).
>
> >Limited Experimental Evaluation: The experimental validation is conducted on relatively simple tasks, such as linear regression with synthetic data.
>
> We acknowledge that our main contribution is theoretical, and that the objective of our experiments is only to verify the theoretical insights we develop.
> These are (i) the separation of unlearning with differential privacy (Figure 1.a) and (ii) the improved complexity cost of out-of-distribution unlearning with Algorithm 2 (Figure 1.b).
> We would also like to recall that algorithms related to our analysis have been validated empirically in practical unlearning scenarios, e.g., (Guo et al. 2020).
>
> >Assumption of Known Forget Set Size: The algorithms and theoretical guarantees often assume that the size of the forget set is known and relatively small compared to the total dataset size.
>
> This is a very good remark.
> We recall that only an upper bound on the size of the forget set should be given to our out-of-distribution unlearning algorithm, which is arguably acceptable for practical cases, e.g., where we would know that no more than say $5$% of the data would be deleted at a time.
> Indeed, Theorem 3 still holds with $f$ replaced by the upper bound given on the forget data size (see lines 413-416).
>
> >Am I correct in stating that unlearning is not generally possible for non-convex models without a unique minimiser? Can the authors comment on relaxing their assumption to general non-convex models?
> Could access to a global minimiser be relaxed with a weaker assumption, like the optima within a bounded set of known diameters?
>
> The reviewer is right in that our theory does not currently tackle non-convex tasks with multiple global minima.
> There are a few ways in which our theory can be extended, which offers an exciting direction of future research.
> First, if the set of minima is bounded, we can hope to achieve an error proportional to the diameter of the set of minima with the same algorithm; this is because the level of added noise would need to be large enough to make the most distant minima indistinguishable.
> Second, for certain overparametrized non-convex tasks, adding regularization could induce implicit bias, so that in reality not all minima are reachable, and the noise to be added (and hence, the final error) can be much smaller.
> We will add this discussion to the paper.

---

> > ### Author Response · Authors · 2024-11-26
> > **Revision Update**
> >
> > Thank you for your feedback. We have further refined the manuscript, improving clarity and adding experiments to strengthen our contributions. We hope these updates effectively address your concerns and highlight the significance of our work.
> > We would greatly appreciate it if you could consider revisiting your score, noting that at this conference, an "8" corresponds to "accept."

---

### Author Response · Authors · 2024-11-26
**Revision Update**

We sincerely thank all reviewers for their constructive feedback and thoughtful engagement with our paper, and for highlighting our contributions. In response to the feedback, we have uploaded a revised version addressing the comments and suggestions. Specifically:

**Improved Clarity and Additional Discussion.** We have enhanced several sections and notation for readability following reviewers' suggestions. We have also included additional comparison with Neel et al. (2021) and Chourasia and Shah (2023) in Section 4, as well as practical implementation details in Remark 6.

**Expanded Empirical Validation:** We have incorporated additional experiments to support our theoretical contributions in the out-of-distribution scenario. We have reconducted our out-of-distribution unlearning experiment on a real dataset (California Housing) as detailed in Appendix F. These new results align with our initial findings and validate our theoretical results, demonstrating that the robust training procedure in Algorithm 2 enables faster unlearning compared to standard unlearning procedures encompassed by Algorithm 1, which are very similar to the algorithms of Neel et al. (2021) and Chourasia and Shah (2023).

---

> ### Author Response · Authors · 2024-12-02
>
> Dear Reviewers,
>
> Thank you again for your valuable feedback. We kindly ask for any further comments or questions before the discussion phase ends so we can address them effectively.

---

### Meta-Review · Area_Chair_3wCk · 2024-12-23

**Metareview:**

This paper derives a theoretical analysis of machine unlearning, focusing on data deletion of in-distribution and out-of-distribution samples. The main results show that empirical risk minimization with output perturbation achieves favourable trade-offs for in-distribution data. The paper also introduces a noisy gradient descent variant that balances complexity and utility for out-of-distribution data.

The reviewers were favorable towards the work and its contributions overall. Reviewer FXKP was very positive, yet their review was brief. Reviewer RJfS emphasized the analysis of in-distribution versus out-of-distribution contexts and appreciates the noisy gradient descent of OOD unlearning. Reviewer BBwt -- the most negative reviewer -- raised concerns about theoretical results and experiments, which the authors addressed through an extensive rebuttal. Reviewer Nr4d requested clarity improvements to notation and definitions, which the authors also addressed in the rebuttal.

The paper merits acceptance due to its theoretical contributions to machine unlearning. The OOD vs. in-distribution angle is interesting and new, and the paper provides meaningful algorithms and theoretical guarantees. Concerns surrounding notation and proofs were assuaged during the rebuttal. I encourage the authors to implement the promised changes and address all of the reviewers comments in the final version of the manuscript.

**Additional Comments On Reviewer Discussion:**

The authors addressed most of the reviewers' concerns during the rebuttal, and reviewers stood by their scores in the discussion phase.

---

### Decision · Program_Chairs · 2025-01-22

Accept (Poster)